# Aged bone matrix-derived extracellular vesicles as a messenger for calcification paradox

Zhen-Xing Wang [1,2], Zhong-Wei Luo[1,2], Fu-Xing-Zi Li[3], Jia Cao [1,2], Shan-Shan Rao[2,4], Yi-Wei Liu[1,2], Yi-Yi Wang[1,2], Guo-Qiang Zhu[1,2], Jiang-Shan Gong[1,2], Jing-Tao Zou[1,2], Qiang Wang[5], Yi-Juan Tan[1,2], Yan Zhang[2], Yin Hu[2], You-You Li[1,2], Hao Yin[1,2], Xiao-Kai Wang[6], Ze-Hui He[1,2], Lu Ren[3], Zheng-Zhao Liu [2,7,8,9,10], Xiong-Ke Hu[2], Ling-Qing Yuan [3], Ran Xu[3], Chun-Yuan Chen [1,2✉] & Hui Xie [1,2,7,8,9,10✉]

Adipocyte differentiation of bone marrow mesenchymal stem/stromal cells (BMSCs) instead of osteoblast formation contributes to age- and menopause-related marrow adiposity and osteoporosis. Vascular calcification often occurs with osteoporosis, a contradictory association called "calcification paradox". Here we show that extracellular vesicles derived from aged bone matrix (AB-EVs) during bone resorption favor BMSC adipogenesis rather than osteogenesis and augment calcification of vascular smooth muscle cells. Intravenous or intramedullary injection of AB-EVs promotes bone-fat imbalance and exacerbates Vitamin D3 (VD3)-induced vascular calcification in young or old mice. Alendronate (ALE), a bone resorption inhibitor, down-regulates AB-EVs release and attenuates aging- and ovariectomy-induced bone-fat imbalance. In the VD3-treated aged mice, ALE suppresses the ovariectomy-induced aggravation of vascular calcification. MiR-483-5p and miR-2861 are enriched in AB-EVs and essential for the AB-EVs-induced bone-fat imbalance and exacerbation of vascular calcification. Our study uncovers the role of AB-EVs as a messenger for calcification paradox by transferring miR-483-5p and miR-2861.

[1] Department of Orthopedics, Xiangya Hospital, Central South University, Changsha, Hunan, China. [2] Movement System Injury and Repair Research Center, Xiangya Hospital, Central South University, Changsha, Hunan, China. [3] The Second Xiangya Hospital, Central South University, Changsha, Hunan, China. [4] Xiangya Nursing School, Central South University, Changsha, Hunan, China. [5] Department of Laboratory Medicine, Affiliated Zhejiang Hospital, Zhejiang University School of Medicine, Hangzhou, Zhejiang, China. [6] Department of Emergency Medicine, Xiangya Hospital, Central South University, Changsha, Hunan, China. [7] Department of Sports Medicine, Xiangya Hospital, Central South University, Changsha, Hunan, China. [8] National Clinical Research Center for Geriatric Disorders, Xiangya Hospital, Central South University, Changsha, Hunan, China. [9] Hunan Key Laboratory of Organ Injury, Aging and Regenerative Medicine, Changsha, Hunan, China. [10] Hunan Key Laboratory of Bone Joint Degeneration and Injury, Changsha, Hunan, China. ✉email: chency19@csu.edu.cn; huixie@csu.edu.cn

O steoporosis is a bone disease characterized by low bone mass and poor bone quality due to an imbalanced bone metabolism and often occurs in aged people, especially in postmenopausal women[1–3]. Unbalanced differentiation of bone marrow mesenchymal stem/stromal cells (BMSCs) into adipocytes at the expense of osteoblasts contributes importantly to age- and menopause-associated marrow adiposity and osteoporosis[4]. Vascular calcification, a pathological process sharing many common features with bone formation, is often accompanied by decreased bone mass or disturbed bone remodeling[5–7]. This contradictory association is called "calcification paradox"[5–7]. However, the mechanism underlying the calcification paradox is not fully understood.

Exosomes are 40–150 nm extracellular vesicles (EVs) derived from multivesicular bodies and play critical roles in intercellular communication by transferring bioactive components, such as proteins, messenger RNAs (mRNAs), microRNAs (miRNAs), and genomic DNA to recipient cells[8–11]. Bone matrix vesicles are nanosized EVs located within the matrix of bone and provide the initial site for bone mineralization[12]. Similarities in size, morphology, and components (proteins and lipids) between bone matrix vesicles and exosomes suggest that bone matrix vesicles are anchored exosomes secreted by bone cells[12]. Since the term "exosomes" in the vast majority of the literatures generally refers to EVs sharing certain biochemical or biophysical features of exosomes without demonstrating their intracellular origin, here we chose to use the generic term "EVs" instead of "exosomes". Currently, the cell types mainly responsible for the production of bone matrix EVs (B-EVs) remain unclear. Evidence is also lacking on whether B-EVs are involved in age- and menopause-associated BMSC fate switching and bone-fat imbalance.

Like bone cells, vascular smooth muscle cells (VSMCs) can also produce matrix EVs-like structures[13]. Matrix EVs from the mineralized VSMCs contain fewer matrix Gla protein (a mineralization inhibitor) and have a higher ability to promote mineral nucleation than matrix vesicles secreted under physiological condition[13], suggesting the involvement of matrix EVs in vascular calcification. Studies have indicated that multiple growth factors deposited in the bone matrix can be released into bone marrow during osteoclastic bone resorption and then participate in the regulation of bone homeostasis[14,15]. These reports, together with the evidence of the relatively or absolutely increased bone resorption in senile or postmenopausal osteoporosis[1,16], suggest that B-EVs may be released into bone marrow and then into circulation to induce calcification paradox during skeletal aging and menopause.

In this study, we uncover the role of EVs from the aged bone matrix (AB-EVs) as a messenger for the calcification paradox by transferring specific miRNAs to BMSCs and VSMCs. First, we isolated B-EVs from the aged and young humans or rats and evaluated the exosome identity of theses vesicles. We then investigated the cell origin of B-EVs and compared the in vitro effects of AB-EVs and EVs from young bone (YB-EVs) on osteogenic and adipogenic differentiation of BMSCs as well as on calcification of VSMCs. In vivo, we tested whether AB-EVs could promote bone-fat imbalance and exacerbate Vitamin D3 (VD3)-induced vascular calcification in young or aged mice. Moreover, we evaluated whether alendronate (ALE), a bone resorption inhibitor widely used for osteoporosis treatment[17], could decrease AB-EVs release and attenuate bone-fat imbalance and VD3-induced vascular calcification in aged ovariectomized (OVX) mice. Meanwhile, the candidate miRNAs that may mediate AB-EVs-induced marrow fat accumulation and calcification paradox were screened based on miRNA expression profiling, and their mediator roles were confirmed in vitro and in vivo.

## Results

**Identification and characterization of B-EVs.** YB-EVs and AB-EVs were respectively isolated from young and aged human bone specimens. As viewed under the transmission electron microscope (TEM), YB-EVs and AB-EVs exhibited cup-like morphologies (Fig. 1a). Nanoparticle Tracking Analysis (NTA) revealed that YB-EVs and AB-EVs had mean diameters of 109.1 ± 38.5 nm and 101.9 ± 34.6 nm, respectively (Fig. 1b), similar to that previously reported exosomes[9,18,19]. Flow cytometric analysis showed that a vast majority (>97%) of the isolated YB-EVs and AB-EVs expressed the exosomal markers including CD63, CD81, and TSG101 (Fig. 1c), which further indicates the exosome identity of these vesicles. Exosomes are specifically enriched with many molecules from their parent cells and also express marker proteins specific for their parent cells[20,21]. Flow cytometric analysis was conducted to assess the protein expression of SOST (a glycoprotein protein mainly produced by osteocytes[22,23]), type I collagen (COL I, a osteoblast marker protein[24]), and type X collagen (COL X, a phenotypic marker of hypertrophic chondrocytes[25]) in YB-EVs, AB-EVs, osteocytes-derived EVs (OCY-EVs), osteoblasts-derived EVs (OB-EVs), and hypertrophic chondrocytes-derived EVs (HYPC-EVs). The results showed that SOST protein was expressed in YB-EVs, AB-EVs, OCY-EVs, and HYPC-EVs, but not in OB-EVs (Fig. 1d). COL I and COL X were expressed in the vast majority of OB-EVs and HYPC-EVs, respectively, whereas OCY-EVs were negative for this protein (Fig. 1d). A very small proportion (<10%) of YB-EVs and AB-EVs were positive for COL I and COL X (Fig. 1d). These results indicate that the majority of B-EVs are released from osteocytes, but not from osteoblasts or hypertrophic chondrocytes.

The tetraspanin CD63 is preferentially enriched in the membrane of intracellular endosomal intraluminal vesicles (ILVs, the precursors of exosomes) and extracellular exosomes, but not in other types of EVs[26]. To label the osteocyte-specific ILVs/exosomes in bone and vasculature, we generated Cd63[em(loxp-mCherry-loxp-eGFP)3] mice harboring conditional Cd63 alleles in which the stop codon in exon 8 at the 3′ UTR of Cd63 gene was replaced with a knock-in mCherry reporter gene flanked by two LoxP sites, and followed by an eGFP reporter gene, which would be activated if the loxp-mCherry-loxp sequence was excised (Fig. 1e). Then, we crossed the reporter mice with the transgenic mice expressing improved Cre recombinase under the control of the dentin matrix protein 1 (Dmp1) promoter to generate Dmp1[iCre]; Cd63[em(loxp-mCherry-loxp-eGFP)3] mice, whose eGFP could be transcribed in osteocytes due to the deletion of the floxed mCherry by iCre recombinase (Fig. 1e, f). The fluorescent signals of mCherry and eGFP proteins in the bone and vessel sections of Dmp1[iCre]; Cd63[em(loxp-mCherry-loxp-eGFP)3] mice, Cd63[em(loxp-mCherry-loxp-eGFP)3] mice, and Dmp1[iCre] mice were assessed. Meanwhile, immunostaining was performed to determine whether the eGFP-labeled exosomes could be stained by SOST protein. As evidenced by Fig. 1g, abundant mCherry red fluorescence was observed in cells (osteocytes) within the bone matrix of cortical bone of Cd63[em(loxp-mCherry-loxp-eGFP)3] mice, and in cells of vascular tissues from Cd63[em(loxp-mCherry-loxp-eGFP)3] mice and Dmp1[iCre]; Cd63[em(loxp-mCherry-loxp-eGFP)3] mice. However, there were only a few signals of mCherry protein in osteocytes of Dmp1[iCre]; Cd63[em(loxp-mCherry-loxp-eGFP)3] mice (Fig. 1g). Abundant dot-like eGFP green signals were at the perinuclear region of osteocytes, within the bone matrix, or detected in the vascular tissues of these mice, and most of these green dots were positive for SOST protein (Fig. 1g), suggesting that most of them are the released OCY-EVs and can enter into the bone matrix and vascular tissues under physiologic condition. Neither mCherry or eGFP signals were observed in the bone and

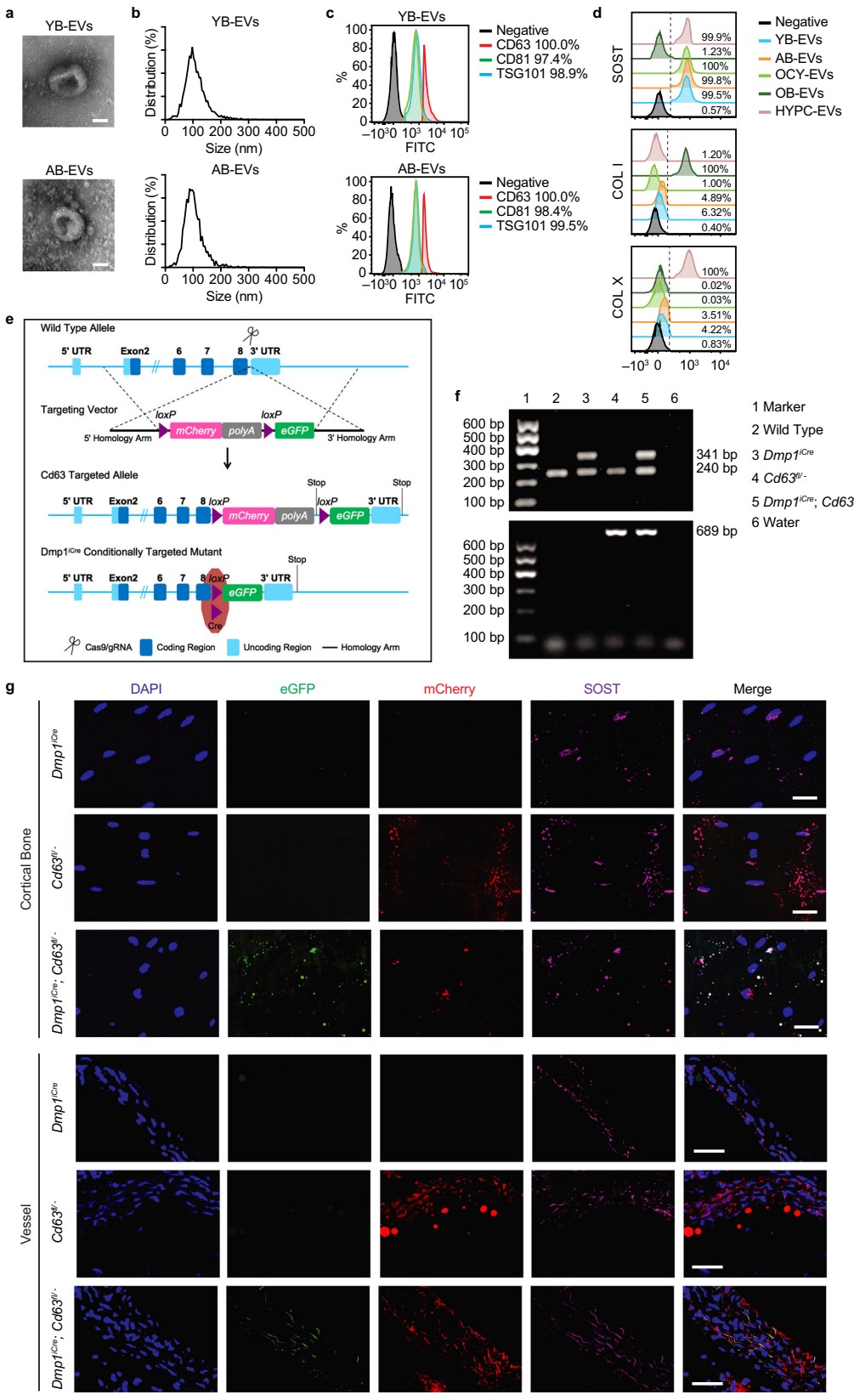

vascular tissues of *Dmp1^iCre* mice (Fig. 1g), indicating that these signals in *Cd63*^em(loxp-mCherry-loxp-eGFP)3 mice and *Dmp1^iCre*; *Cd63*^em(loxp-mCherry-loxp-eGFP)3 mice are not nonspecific-fluorescence. There were dot-like SOST-positive signals in the vascular tissues of *Dmp1^iCre* mice and *Cd63*^em(loxp-mCherry-loxp-eGFP)3 mice (Fig. 1g),

which also suggest that OCY-EVs can be transported from bone to blood vessels under normal circumstance.

Together, these findings indicate that the majority of B-EVs are derived from osteocytes and can enter from bone matrix to blood vessels.

**Fig. 1 Identification and characterization of B-EVs. a–d**, Morphology (**a**), diameter distribution (**b**), exosomal marker analysis (**c**), and expression of SOST, COL I, and COL X (**d**) in the indicated EVs. Scale bar: 50 nm. **e** Schematic diagram of the gene targeting strategy for the generation of $Cd63^{em(loxp-mCherry-loxp-eGFP)3}$ mice by inserting a *mCherry* reporter gene flanked by two *loxP* sites and an *eGFP* reporter gene in the stop codon in exon 8 at the 3' UTR of *Cd63* gene. **f** PCR genotyping of wild-type mice, *Dmp1^iCre^* mice, $Cd63^{em(loxp-mCherry-loxp-eGFP)3}$ mice, and $Dmp1^{iCre}$; $Cd63^{em(loxp-mCherry-loxp-eGFP)3}$ mice using primers for determining the insertion of *iCre* (up) and *eGFP* (down). $n = 3$ biologically independent animals. **g** Localization of eGFP (green) and mCherry (red), and immunofluorescence staining for SOST (purple) in bone and vessel from *Dmp1^iCre^* mice, $Cd63^{em(loxp-mCherry-loxp-eGFP)3}$ mice, and $Dmp1^{iCre}$; $Cd63^{em(loxp-mCherry-loxp-eGFP)3}$ mice. Scale bar: 20 μm (for bone) or 50 μm (for vessel). $n = 3$ biologically independent animals. Experiments in **a–d** were repeated independently three times with similar results. Experiments in **f–g** were repeated independently two times with similar results. The illustrated results represented one of the three or two independent experiments. Source data are provided as a Source Data file.

**AB-EVs favor adipogenesis rather than osteogenesis of BMSCs and augment osteogenic transdifferentiation of VSMCs**. To assess the impacts of AB-EVs and YB-EVs on osteogenic and adipogenic differentiation of BMSCs, we firstly detected whether the red lipophilic dye DiI-labeled AB-EVs and YB-EVs could be internalized by mouse BMSCs. As shown in Supplementary Fig. 1a, large numbers of red fluorescent signals were presented at the perinuclear region of BMSCs treated with AB-EVs or YB-EVs for 3 h, indicating the successful uptake of these EVs by BMSCs. Quantitative real-time PCR (qRT-PCR) analysis (Supplementary Fig. 1b) and Alizarin Red S (ARS) staining (Fig. 2a, b), respectively, showed that YB-EVs, but not AB-EVs, markedly promoted the expression of osteogenesis-related genes (*Runx2*, *Bglap*, and *Alpl*) and calcium nodule formation of BMSCs under osteogenic induction. In contrast, AB-EVs, but not YB-EVs, profoundly augmented the expression of adipogenesis-related genes (*Ppary*, *Cebpα*, and *Fabp4*) and lipid droplet formation of BMSCs under adipogenic differentiation, as shown by qRT-PCR analysis (Supplementary Fig. 1c) and Oil Red O (ORO) staining (Fig. 2a, b). Fluorescence microscope showed that both the DiI-labeled AB-EVs and YB-EVs could be taken up by VSMCs (Supplementary Fig. 2a). Interestingly, ARS staining, qRT-PCR analysis, and alkaline phosphatase (ALP) activity assay, respectively, showed that AB-EVs significantly increased calcium deposition (Fig. 2c, d), reduced the expression of SMC markers (including smooth muscle 22α (SM22α) and smooth muscle α-actin (SMαA/αSMA)[27,28] (Fig. 2e), enhanced the expression of *RUNX2* (a transcription factor essential for osteogenesis and VSMC calcification[27,29]) and *COL1A1* (a gene that encodes the major component of extracellular matrix[30]) (Fig. 2f), and augmented ALP activity (a critical marker of osteoblast-like change of VSMCs[30]) (Fig. 2g) in human VSMCs undergoing osteogenic induction, indicating that AB-EVs promote the switching of VSMC phenotype to osteogenic phenotype. However, YB-EVs did not induce significant changes of these parameters in the differentiated VSMCs (Fig. 2c–g), all of which were inconsistent with their remarkable stimulatory effects on the osteogenic differentiation of BMSCs. Cell counting kit-8 (CCK-8) assay indicated that AB-EVs did not reduce the survival or growth of VSMCs (Supplementary Fig. 2b), suggesting that the calcium deposition in the AB-EVs-treated VSMCs is not due to the induction of cell death and intracellular calcium ion release. Together, these findings indicate that with the aging of bone donor, B-EVs lose the capacity to stimulate BMSC osteogenesis and their actions become pro-adipogenic on BMSCs and pro-osteogenic on VSMCs.

Consistent with the effects of B-EVs, ARS and ORO staining revealed that OCY-EVs from young donor-derived bone (YB-OCY-EVs) markedly augmented osteogenesis of BMSCs, but OCY-EVs from aged donor-derived bone (AB-OCY-EVs) significantly enhanced BMSC adipogenesis and VSMC mineralization (Fig. 2h, i), which further support the osteocyte origin of B-EVs.

To mimic B-EVs release from bone matrix, the osteoclast progenitor RAW264.7 cells were subjected to osteoclastic induction and cultured with or without bone slices from the aged or young humans. The conditioned media (CM) from osteoclasts receiving different treatments were then harvested to assess their impacts on osteogenic and adipogenic differentiation of BMSCs and calcification of VSMCs. Consistent with the effects of YB-EVs on BMSC differentiation, ARS and ORO staining revealed that the bone-resorption CM from osteoclasts with young bone slices (OC^YB^-CM) remarkably enhanced the ability of BMSCs to form calcium nodules under osteogenic induction, but did not induce obvious effect on adipogenesis of BMSCs compared with CM from osteoclasts without bone slices (OC-CM) (Fig. 2j, k). On the contrary, the bone-resorption CM from osteoclasts with aged bone slices (OC^AB^-CM) had no notable influence on BMSC osteogenesis, but induced a significant increase in adipogenic differentiation of BMSCs (Fig. 2j, k). EVs were isolated from OC^YB^-CM (EVs in OC^YB^-CM) and OC^AB^-CM (EVs in OC^AB^-CM) and their effects on BMSC differentiation were evaluated. Similar to CM from which they were derived, EVs in OC^YB^-CM and OC^AB^-CM, respectively, were able to profoundly augment osteogenic and adipogenic differentiation of BMSCs, whereas the EVs-depleted OC^AB^-CM exhibited a significantly impaired ability to stimulate adipogenesis of BMSCs (Fig. 2j, k), indicating an essential role of EVs in the OC^AB^-CM-induced promotion of adipogenesis of BMSCs. Figure 2l, m show the extents of VSMC calcification in different groups as visualized by ARS staining. Neither OC^YB^-CM nor EVs in OC^YB^-CM induced notable effects on VSMC calcification, whereas OC^AB^-CM and EVs in OC^AB^-CM, but not the exosomes-depleted OC^AB^-CM, remarkably enhanced the calcification of VSMCs. These results suggest that B-EVs can be released from the bone matrix during bone resorption to regulate BMSC differentiation fate and VSMC calcification. The positive effects of OC^AB^-CM and EVs in OC^AB^-CM on BMSC adipogenesis and VSMC calcification, together with the similar action of AB-EVs directly isolated from the aged bone matrix, prompt us to investigate whether AB-EVs are involved in the induction of bone marrow adiposity and calcification paradox in vivo.

**AB-EVs promote bone-fat imbalance in young and aged mice.** We next asked whether the intramedullary injection of AB-EVs one time per two weeks for one month could induce bone loss and marrow fat accumulation in young (3-month-old) mice and aggravate bone-fat imbalance in aged (15-month-old) mice (Fig. 3a). The effects of YB-EVs on bone were also assessed. Considering animal samples are easily attainable than human specimens and a larger quantity of B-EVs can be obtained from rat bone than that from an equal number of mouse bone, AB-EVs and YB-EVs were respectively isolated from bone specimens from 18-month-old and 2-month-old SD rats. The aged donor rats exhibited much higher serum levels of blood urea nitrogen (BUN) and creatinine (CREA) compared with the young donor rats (Supplementary Fig. 3a, b), indicating that aging induced the reduction of kidney function in the rats. Similar to that from human bones, the rats-derived AB-EVs also significantly

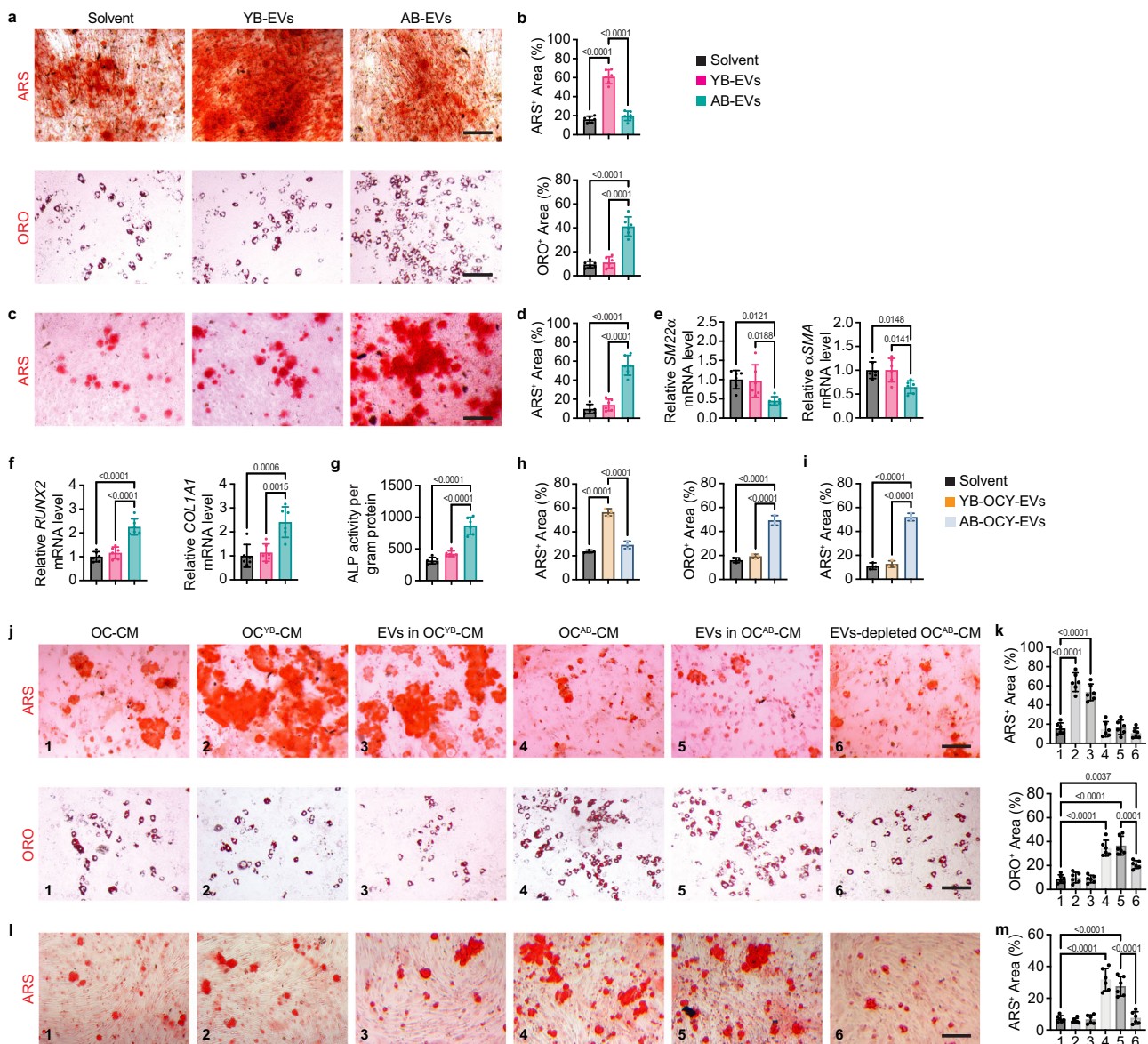

**Fig. 2 AB-EVs favor adipogenesis rather than osteogenesis of BMSCs and augment osteogenic transdifferentiation of VSMCs. a,b**, ARS or ORO staining of BMSCs treated with solvent, YB-EVs, or AB-EVs under osteogenic or adipogenic induction (**a**) and quantification of the percentages of ARS+ (red) and ORO+ (red) areas (**b**). Scale bar: 50 µm. $n = 6$ biologically independent cells per group. **c–g**, ARS staining (**c**), quantification of the percentage of ARS+ areas (red; **d**), qRT-PCR analysis of *SM22α* and *αSMA* (**e**), *RUNX2* and *COL1A1* (**f**) expression, and ALP activity (**g**) in VSMCs treated with solvent, YB-EVs, or AB-EVs under osteogenic induction. Scale bar: 50 µm. $n = 6$ biologically independent cells per group. **h** Quantification of the percentages of ARS+ and ORO+ areas in BMSCs treated with solvent, YB-OCY-EVs, or AB-OCY-EVs under osteogenic or adipogenic induction. $n = 3$ biologically independent cells per group. **i** Quantification of the percentages of ARS+ areas in VSMCs treated with solvent, YB-OCY-EVs, or AB-OCY-EVs under osteogenic induction. $n = 3$ biologically independent cells per group. **j–k**, ARS and ORO staining (**j**) and quantification of the percentages of ARS+ (red) and ORO+ (red) areas (**k**) in BMSCs with different treatments under osteogenic or adipogenic induction. OC: osteoclasts; CM: conditioned media. Scale bar: 50 µm. $n = 6$ biologically independent cells per group. **l–m**, ARS staining (**l**) and quantification of the percentage of ARS+ areas (red; **m**) in VSMCs with different treatments under osteogenic induction. Scale bar: 50 µm. $n = 6$ biologically independent per group. Experiments in **a–d** and **h–m** were repeated independently three times with similar results. The illustrated results represented one of the three independent experiments. Experiments in **e–g** were performed with six biological replicates per group without independent repetition. Data were presented as mean ± SD. Statistical significance was determined by one-way ANOVA with Bonferroni post hoc test. Source data are provided as a Source Data file.

stimulated adipogenesis rather than osteogenesis of BMSCs and increased VSMC mineralization, while YB-EVs from rats markedly enhanced calcium nodule formation of BMSCs and had no distinct effect on BMSC adipogenesis and VSMC calcification, as indicated by ARS and ORO staining of the differentiated BMSCs (Supplementary Fig. 3c, d) and ARS staining of the differentiated VSMCs (Supplementary Fig. 3e, f). As shown in Fig. 3b, c, the AB-EVs-treated young and aged mice had the lowest trabecular

bone mass phenotypes compared with the other two age-matched groups, as revealed by the microcomputed tomography (µCT)-reconstructed images of femurs, the much lower trabecular bone volume fraction (Tb. BV/TV), trabecular number (Tb. N), or/and trabecular thickness (Tb. Th), as well as the higher trabecular separation (Tb. Sp). In contrast, YB-EVs remarkably enhanced Tb. BV/TV and Tb. Th in young mice, and markedly increased Tb. Th and decreased Tb. Sp in aged mice (Fig. 3b, c). Both YB-

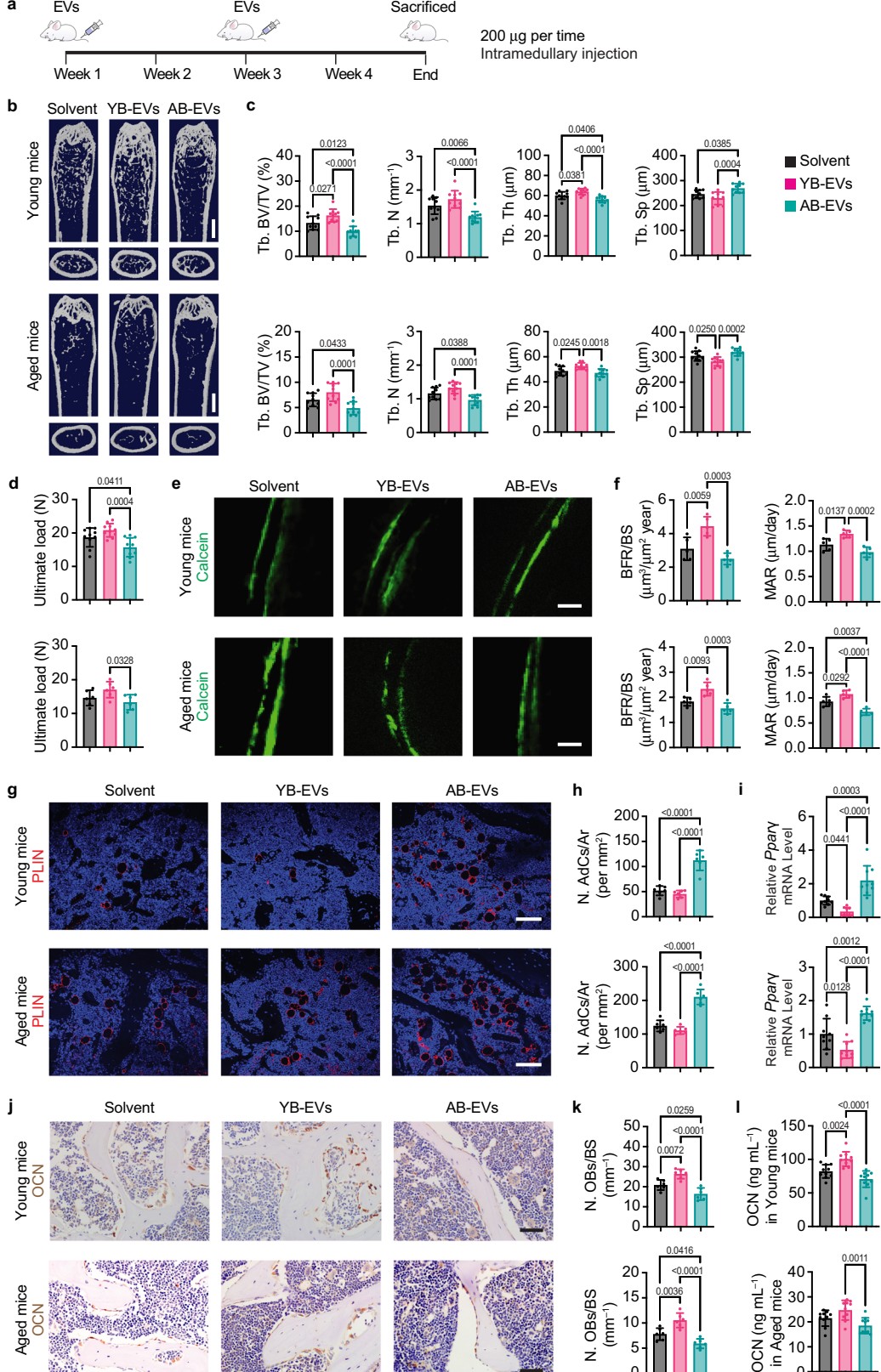

EVs and AB-EVs did not cause statistically significant differences of cortical bone area fraction (Ct. Ar/Tt. Ar) and cortical thickness (Ct. Th) in young and aged mice (Supplementary Fig. 4a, b). Three-point bending test revealed that YB-EVs induced a trend of increase of femur ultimate load value in both young and aged mice, whereas AB-EVs caused a significant and tend of decrease

of this parameter in young and aged mice, respectively (Fig. 3d), indicating the potential positive effect of YB-EVs and definite negative effect of AB-EVs on bone strength.

Calcein double-labeling showed that YB-EVs treatment induced much higher levels of bone formation rate per bone surface (BFR/BS) and mineral apposition rate (MAR) values in

**Fig. 3 AB-EVs promote bone-fat imbalance in young and aged mice. a** Schematic diagram of the experimental design for assessing the effects of YB-EVs and AB-EVs on bone phenotypes in young and aged mice. **b** µCT-reconstructed images of femurs. Scale bars: 1 mm. **c** Quantification of Tb. BV/TV, Tb. N, Tb. Th, and Tb. Sp. $n = 10$ biologically independent animals per group. **d** Ultimate load values of femurs. $n = 10$ biologically independent animals for young mice. $n = 6$ biologically independent animals for aged mice. **e–f,** Calcein (green) double labeling of trabecular bones (**e**) and quantification of BFR/BS and MAR (**f**). Scale bar: 25 µm. $n = 5$ biologically independent animals per group. **g–h,** PLIN immunofluorescence staining images of femur sections (**g**) and quantification of the number of PLIN$^+$ (red) adipocytes in bone marrow (**h**). Scale bar: 100 µm. $n = 6$ biologically independent animals per group. **i** qRT-PCR for $Ppar\gamma$ expression in femurs. $n = 9$ biologically independent animals per group. **j–k,** OCN immunohistochemical staining images (**j**) and the number of OCN-stained (brown) osteoblasts (N. OBs) on the trabecular bone surface (BS) (**k**). Scale bar: 50 µm. $n = 6$ biologically independent animals per group. **l** ELISA for serum OCN. $n = 10$ biologically independent animals per group. All experiments were performed with at least five biological replicates per group without independent repetition. Data were presented as mean ± SD. Statistical significance was determined by one-way ANOVA with Bonferroni post hoc test. Source data are provided as a Source Data file.

both young and aged mice, whereas the AB-EVs-treated young and aged mice exhibited trend of or significant lower levels of these indicators compared with the age-matched mice treated with solvent or YB-EVs (Fig. 3e, f). Immunofluorescence staining for perilipin (PLIN) demonstrated that only a small number of adipocytes were observed in the bone marrow of the solvent-treated young mice, while marrow adipocytes accumulated in the aged solvent-treated mice (Fig. 3g, h). Treatment with AB-EVs, but not YB-EVs, significantly increased the number of adipocytes in the bone marrow of young and aged mice (Fig. 3g, h), consistent with the pro-adipogenic effect of AB-EVs on BMSCs. In aged mice, YB-EVs induced a trend of decrease in marrow adipocyte number (Fig. 3g, h), which may be due to that YB-EVs favor osteogenesis rather than adipogenesis of BMSCs. qRT-PCR analysis for the gene expression of adipogenic transcription factor $Ppar\gamma$ demonstrated the remarkable stimulatory effect of AB-EVs and the significant inhibitory effect of YB-EVs on adipogenesis (Fig. 3i).

Immunohistochemical staining for osteocalcin (OCN) revealed that YB-EVs significantly enhanced osteoblast number on trabecular bones of both young and aged mice (Fig. 3j, k). In contrast, AB-EVs caused a marked decrease in osteoblast number compared with the solvent-treated group (Fig. 3j, k), even if AB-EVs did not show any negative effect on osteogenic differentiation of BMSCs in vitro. This may be attributable to the augmentation of BMSC adipogenesis at the expense of osteogenesis after AB-EVs treatment. Enzyme-linked immunosorbent assay (ELISA) for serum OCN further confirmed the remarkable positive effect of YB-EVs on osteogenic activity, whereas the AB-EVs-treated young and mice showed a trend of decrease in serum OCN compared with the solvent-treated control mice (Fig. 3l). These findings indicate that B-EVs contribute to the bone formation during the growth period, but facilitate bone marrow fat accumulation rather than bone formation during skeletal aging.

Tartrate-resistant acid phosphatase (TRAP) staining (Supplementary Fig. 4c, d) and ELISA for the bone resorption marker C-terminal telopeptides of type I collagen (CTX-I) in serum (Supplementary Fig. 4e) indicated that neither AB-EVs nor YB-EVs notably affected osteoclast formation and activity in both young and aged mice. Consistently, the result in vitro also demonstrated that both AB-EVs and YB-EVs did not trigger marked effects on osteoclast formation of RAW264.7 cells (Supplementary Fig. 4f, g).

The spleen samples from the above-described young mice receiving solvent, YB-EVs, or AB-EVs treatment were obtained and photographed. As shown in Supplementary Fig. 5a, b, the YB-EVs- or AB-EVs-treated mice showed comparable spleen sizes and weights compared to the solvent-treated control mice. Hematoxylin and eosin (H&E) staining revealed that treatment with YB-EVs or AB-EVs did not induce obvious histopathological changes such as inflammatory cell infiltration and lymph node

hyperplasia in the mouse spleen tissues (Supplementary Fig. 5c). There were also no significant differences in the percentages of lymphocytes and neutrophils in white blood cells among the solvent-, YB-EVs-, or AB-EVs-treated mice (Supplementary Fig. 5d). Together, these findings indicate that the rats-derived YB-EVs and AB-EVs do not induce notable immune and inflammatory responses in mice after intravenous injection.

**AB-EVs exacerbate VD3-induced vascular calcification.** We then established experimental models with acute vascular calcification in 3-month-old mice by intraperitoneal injection with VD3 for continuous four days[31,32] and compared the effects of AB-EVs and YB-EVs on vascular calcification after two times intravenous injections on days 1 and 3 during the period of VD3 treatment (Fig. 4a). qRT-PCR analysis revealed that treatment with AB-EVs, but not YB-EVs, induced prominent reductions of mRNA levels of SMC markers including $Sm22\alpha$ and $\alpha Sma$ in abdominal aortas of these mice (Fig. 4b), indicating the loss of vascular smooth muscle phenotype after AB-EVs administration. Von Kossa and ARS staining showed the presence of small Von Kossa- or ARS-stained areas in abdominal aortas of the mice treated with solvent or YB-EVs, whereas the aortas from the AB-EVs-treated mice had large calcium deposition lesion areas stained with high intensity of Von Kossa or ARS dye (Fig. 4c–e). The significant increase of calcium deposition in abdominal aortas of the AB-EVs-treated mice was further confirmed by vascular calcium content analysis (Fig. 4f). Immunofluorescence staining for the osteogenic factor RUNX2 and qRT-PCR analysis for the osteogenic marker $Alpl$ revealed that the mice treated with AB-EVs, but not YB-EVs, displayed markedly increased levels of RUNX2 protein and $Alpl$ mRNA in abdominal aortas than that treated with solvent (Fig. 4g–i), indicating that AB-EVs induce the transition of the cells within the vessels into an osteogenic phenotype.

We also established chronic vascular calcification experiment models in 3-month-old female mice by freely feeding the mice with diet containing 0.25% adenine for 4 weeks[33,34] and explored the effects of AB-EVs and YB-EVs on vascular calcification after intravenous injection at the first day of weeks 1 and 3 during the feeding period (Fig. 4j). Consistent with that observed in the mouse models of VD3-induced acute vascular calcification, treatment with AB-EVs, but not YB-EVs, resulted in significantly decreased mRNA levels of $Sm22\alpha$ and $\alpha Sma$ (Fig. 4k), as well as profoundly increased calcium deposition lesion areas (Fig. 4l–n), vascular calcium content (Fig. 4o), RUNX2 protein expression (Fig. 4p, q), and $Alpl$ gene expression (Fig. 4r) in abdominal aortas of the mice with adenine-induced chronic vascular calcification, which further demonstrate the positive role of AB-EVs on vascular calcification.

We then assessed the circulating levels of calcium ions and inorganic phosphate in the VD3-induced acute vascular calcification and adenine-induced chronic vascular calcification mouse

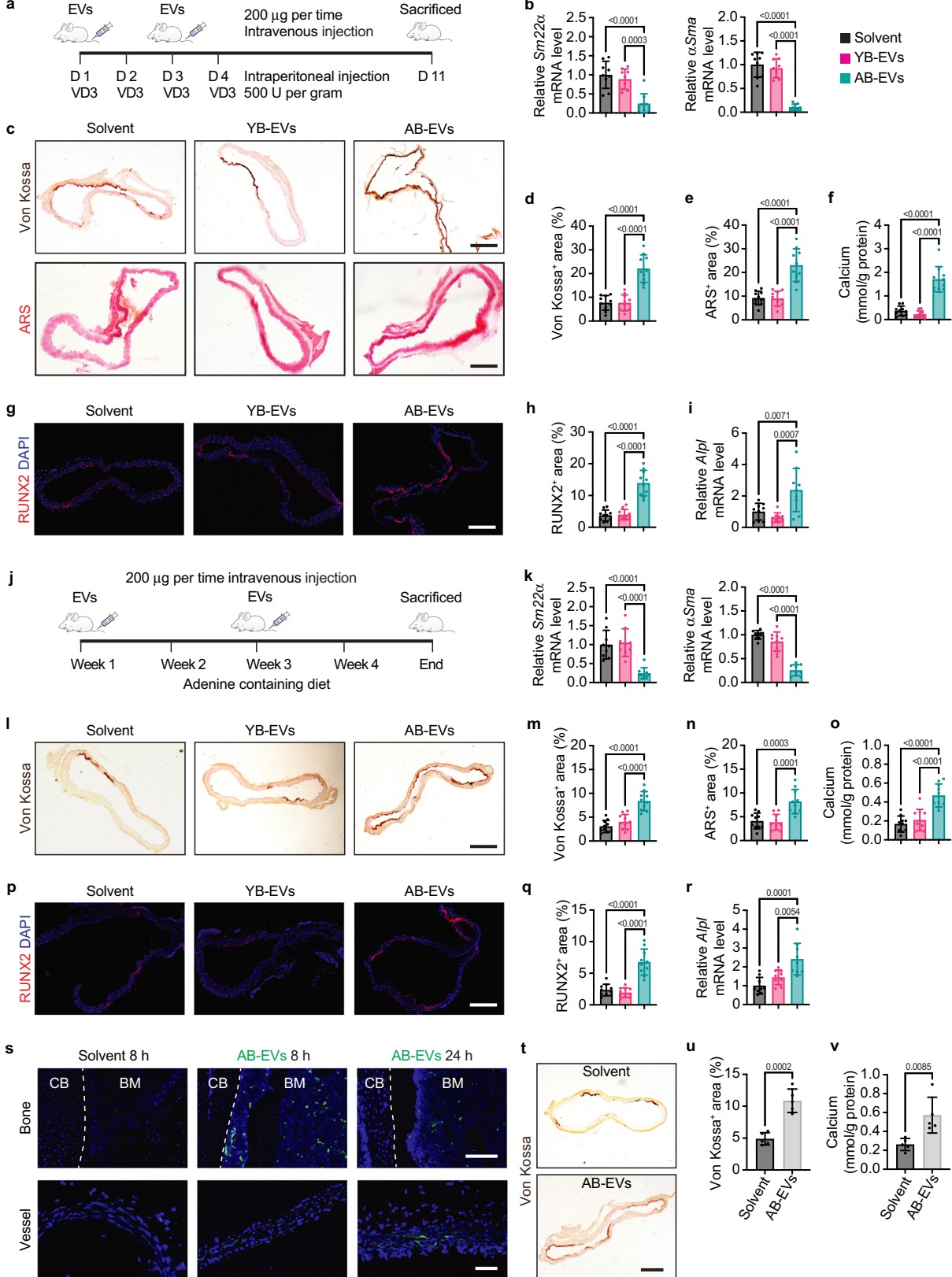

models receiving different treatments. The results showed that YB-EVs induced a significant increase of serum calcium ion in the mice with VD3-induced acute vascular calcification and a trend of increase of serum inorganic phosphate in both the acute and chronic mouse models of vascular calcification (Supplementary Fig. 6a, b). However, treatment with AB-EVs resulted in marked increases of both serum calcium ions and inorganic phosphate in these two models of vascular calcification, and the effects were much higher than that of YB-EVs (Supplementary Fig. 6a, b). The levels of calcium and phosphorus in AB-EVs and B-Exo were also tested. The results revealed that both calcium and phosphorus could be detected in AB-EVs and YB-EVs, but the

**Fig. 4 AB-EVs exacerbate vascular calcification. a** Experimental design of the VD3-induced acute vascular calcification mouse models treated with solvent, YB-EVs, or AB-EVs by intravenous injection. **b** qRT-PCR analysis of $Sm22\alpha$ and $\alpha Sma$ expression in abdominal aortas of mice in (**a**). $n = 9$ biologically independent animals per group. **c–e**, Von Kossa and ARS staining images (**c**) and quantification of the percentages of Von Kossa$^+$ (brownish black; **d**) and ARS$^+$ (red; **e**) areas. Scale bar: 200 μm. $n = 10$ biologically independent animals per group. **f** Vascular calcium content measurement. $n = 10$ biologically independent animals per group. **g–h**, RUNX2 immunofluorescence staining images (**g**) and quantification of the percentage of RUNX2$^+$ (red) areas (**h**). Scale bar: 200 μm. $n = 10$ biologically independent animals per group. **i** qRT-PCR analysis of $Alpl$ expression. $n = 9$ per group. **j** Experimental design of the adenine-induced chronic vascular calcification mouse models treated with solvent, YB-EVs, or AB-EVs by intravenous injection. **k** qRT-PCR analysis of $Sm22\alpha$ and $\alpha Sma$ expression in abdominal aortas of mice in (**j**). $n = 9$ biologically independent animals per group. **l–n** Von Kossa staining images (**l**) and quantification of the percentages of Von Kossa$^+$ (brownish black; **m**) and ARS$^+$ (red; **n**) areas. Scale bar: 200 μm. $n = 10$ biologically independent animals per group. **o** Vascular calcium content measurement. $n = 10$ biologically independent animals per group. **p–q**, RUNX2 immunofluorescence staining images (**p**) and quantification of the percentage of RUNX2$^+$ (red) areas (**q**). Scale bar: 200 μm. $n = 10$ biologically independent animals per group. **r** qRT-PCR for $Alpl$ expression. $n = 9$ biologically independent animals per group. **s** Fluorescence microscopy analysis of femur and abdominal aorta sections from mice treated with solvent or DiO (green)-labeled AB-EVs by intramedullary injection. CB: cortical bone; BM: bone marrow. Scale bar: 200 μm (for bone) or 50 μm (for vessel). $n = 3$ biologically independent animals per group. **t–v**, Von Kossa staining images (**t**), quantification of the percentage of Von Kossa$^+$ areas (brownish black; **u**), and vascular calcium content measurement (**v**) in abdominal aortas from the VD3-induced acute vascular calcification mouse models receiving solvent or AB-EVs treatment by intramedullary injection. Scale bar: 200 μm. $n = 5$ biologically independent animals per group. Experiment in **s** was repeated independently three times with similar results. The illustrated results represented one of the three independent experiments. The other experiments were performed with at least five biological replicates per group without independent repetition. Data were presented as mean ± SD. Statistical significance was determined by one-way ANOVA with Bonferroni *post hoc* test (**a–t**) or unpaired, two-tailed Student's *t*-test (**u–v**). Source data are provided as a Source Data file.

level of calcium in AB-EVs was much higher than that in YB-EVs (Supplementary Fig. 6c). Thus, the delivery of abundant calcium to blood and the increase of blood phosphorus may also contribute to the positive effect of AB-EVs on vascular calcification.

To simulate the transport of AB-EVs from bone to blood vessels in vivo, AB-EVs were labeled with a green fluorescent dye DiO and administered into mice by intramedullary injection. As shown in Fig. 4s, a lot of green dot-like signals were detected in bone marrow at 8 h after AB-EVs injection and the signals were decreased after injection of AB-EVs for 24 h. In the vessel wall of the abdominal aorta, only a small amount of green fluorescence was observed at 8 h after AB-EVs treatment and the signals were increased at 24 h after AB-EVs injection (Fig. 4s), suggesting that AB-EVs located in the bone can be transported to the blood vessel wall. Subsequently, we assessed whether a single intramedullary injection of AB-EVs could exacerbate VD3-induced vascular calcification in 3-month-old mice (Supplementary Fig. 7a). As evidenced by Von Kossa and ARS staining, vascular calcium content analysis, and immunohistochemical staining for RUNX2, respectively, the ratios of calcium deposition lesion areas (Figs. 4t, u and S7b), vascular calcium content (Fig. 4v), and RUNX2 expression (Supplementary Fig. 7c, d) in the vessel wall of the abdominal aortas were significantly higher in the AB-EVs-treated mice than that in the solvent group. Together, these findings indicate that AB-EVs have the ability to accumulate into the vessel wall to promote vascular calcification.

**ALE downregulates AB-EVs release and attenuates bone-fat imbalance and VD3-induced vascular calcification in aged OVX mice.** To investigate whether the bone resorption inhibitor ALE could inhibit the release of AB-EVs from aged bone, ALE or solvent was added to the cultures of osteoclasts with or without bone slices from an aged volunteer (AB). The CM from osteoclasts treated with solvent (OC-CM), ALE (OC$^{ALE}$-CM), AB (OC$^{AB}$-CM), or AB + ALE (OC$^{AB+ALE}$-CM) were collected to assess the changes of contents of EVs and the effects of these CM on osteogenesis and adipogenesis of BMSCs and calcification of VSMCs. As the concentration of EV proteins is widely used as an indicator of the quantity of EVs[35–37], the contents of EV proteins in these CM were compared. As shown in Fig. 5a, ALE did not significantly affect the production of EVs in osteoclasts cultured without aged bone slices, but notably reduced the concentration

of EVs in the culture supernatant of osteoclasts in the presence of aged bone slices, as indicated by the comparable levels of EV proteins between OC-CM and OC$^{ALE}$-CM groups, and the much lower levels of EV proteins in OC$^{AB+ALE}$-CM group compared with OC$^{AB}$-CM group. This suggests that ALE can successfully inhibit AB-EVs release from the bone matrix during osteoclastic bone resorption. The quantitative data for ARS and ORO staining of the differentiated BMSCs (Fig. 5b, c) and ARS staining of the differentiated VSMCs (Fig. 5d) revealed that OC$^{ALE}$-CM did not induce significant changes in osteogenic and adipogenic differentiation of BMSCs and calcification of VSMCs relative to OC-CM. However, OC$^{AB+ALE}$-CM lost the ability to augment BMSC adipogenesis and VSMC calcification when compared with OC$^{AB}$-CM (Fig. 5b–d), suggesting that ALE blocks the release of AB-EVs from the bone matrix during bone resorption and thereby leads to the decreased ability of OC$^{AB}$-CM to stimulate adipogenesis of BMSCs and calcification of VSMCs.

The animals subjected to ovariectomy have been widely used as experimental models of postmenopausal osteoporosis and studies have shown the markedly increased bone resorption activity in OVX model[1,38]. Thus, we established OVX models in 16-month-old female mice and tested whether the intragastric administration of ALE once a week for three times could attenuate bone-fat imbalance in these aged OVX mice (Supplementary Fig. 8a). The much lower uterus weights in all OVX mice relative to all sham-operated mice confirmed the success of the operation (Supplementary Fig. 8b). ALE did not markedly affect uterus weights in OVX mice (Supplementary Fig. 8b), which excluded the regulatory effect of ALE on estrogen production. μCT-reconstructed images of femur samples and quantitative trabecular bone microstructural parameters revealed that ALE caused a trend of increase in trabecular bone mass in Sham mice and profoundly reversed the OVX-induced reduction of trabecular bone mass in OVX mice (Fig. 5e, f). Treatment with ALE did not induce any notable effects on cortical bone parameters including Ct. Ar/Tt. Ar and Ct. Th in both Sham and OVX mice (Supplementary Fig. 8c). The profound inhibitory effects of ALE on osteoclast formation and activity were confirmed by TRAP staining of femur sections (Supplementary Fig. 8d, e) and ELISA for serum CTX-I (Supplementary Fig. 8f), respectively. Immunofluorescence staining for PLIN and qRT-PCR analysis for $Ppar\gamma$ indicated that ALE induced trend of decreases of bone marrow adipocyte number and $Ppar\gamma$ expression in Sham mice (Fig. 5g, i). In OVX mice, ALE did not cause a statistically significant reduction of $Ppar\gamma$ expression,

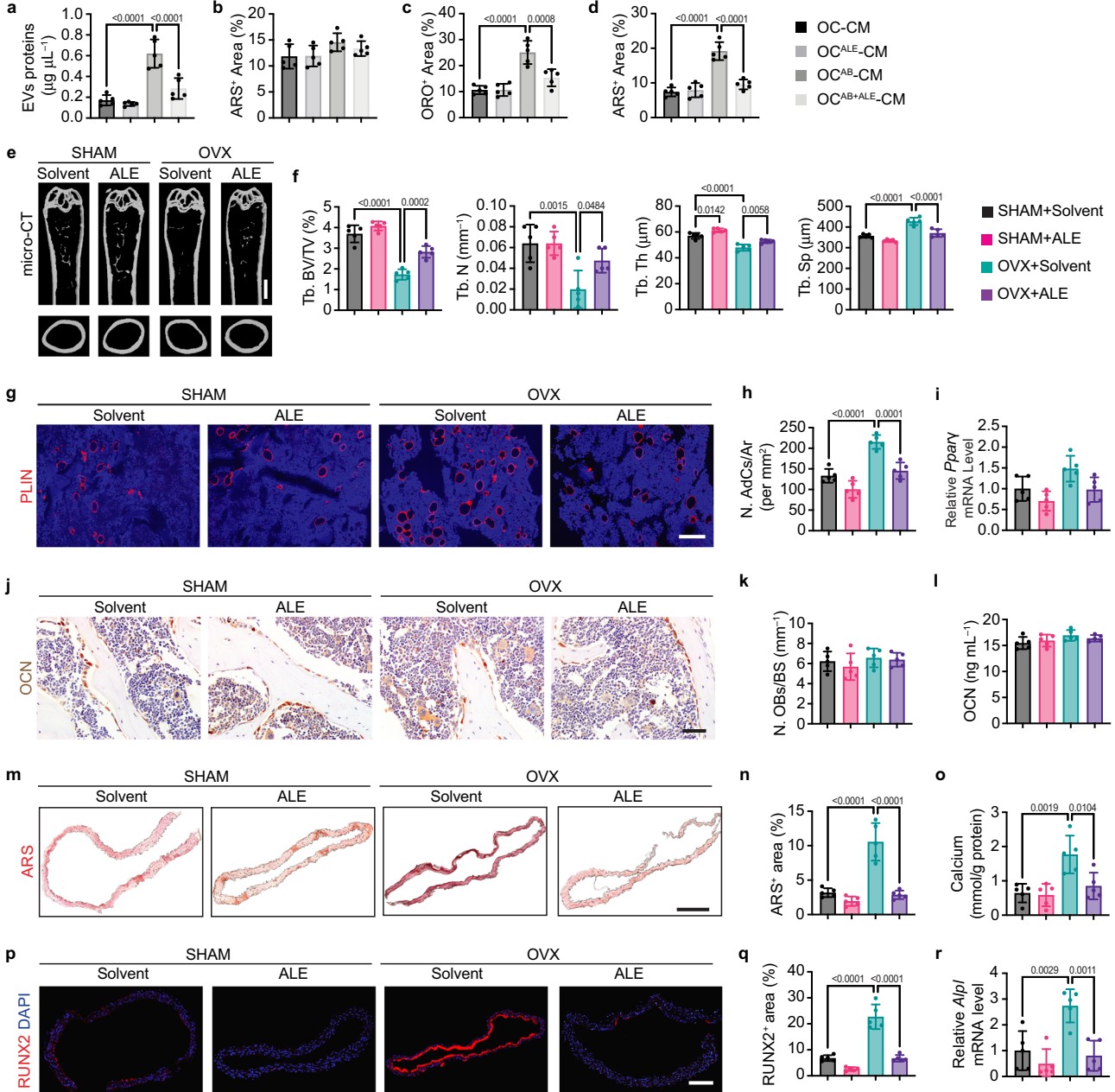

**Fig. 5 ALE down-regulates AB-EVs release and attenuates bone-fat imbalance and VD3-induced vascular calcification in aged OVX mice. a** Total protein contents of EVs isolated from the conditioned media of osteoclasts treated with solvent (OC-CM), ALE (OC$^{ALE}$-CM), AB (OC$^{AB}$-CM), or AB + ALE (OC$^{AB+ALE}$-CM). $n = 5$ biologically independent samples per group. **b–d**, Quantification of the percentages of ARS$^+$ (**b**) and ORO$^+$ (**c**) areas in BMSCs with different treatments under osteogenic or adipogenic induction, or ARS$^+$ areas (**d**) in VSMCs with different treatments under osteogenic induction. $n = 5$ biologically independent cells per group. **e–f** μCT-reconstructed images of femurs from 16-month-old Sham or OVX mice in different groups (**e**) and quantification of Tb. BV/TV, Tb. N, Tb. Th, and Tb. Sp (**f**). Scale bars: 1 mm. $n = 5$ biologically independent animals per group. **g–h** PLIN immunofluorescence staining images of femur sections (**g**) and quantification of the number of PLIN$^+$ (red) adipocytes in bone marrow (**h**). Scale bar: 100 μm. $n = 5$ biologically independent animals per group. **i** qRT-PCR for *Ppary* expression in femurs. $n = 5$ biologically independent animals per group. **j–l** OCN immunostaining images (**j**), quantification of the number of OCN-stained (brown) osteoblasts on BS (**k**), and ELISA for OCN (**l**). Scale bar: 50 μm. $n = 5$ biologically independent animals per group. **m–n** ARS staining images (**m**) and quantification of the percentage of ARS$^+$ areas (red; **n**). Scale bar: 200 μm. $n = 5$ biologically independent animals per group. **o** Vascular calcium content measurement. $n = 5$ biologically independent animals per group. **p–q**, RUNX2 immunostaining images (**p**) and quantification of the percentage of RUNX2$^+$ (red) areas (**q**). Scale bar: 200 μm. $n = 5$ biologically independent animals per group. **r** qRT-PCR for *Alpl* expression. $n = 5$ biologically independent animals per group. Experiments in **b–d** were repeated independently three times with similar results. The illustrated results represented one of the three independent experiments. The other experiments were performed with at least five biological replicates per group without independent repetition. Data were presented as mean ± SD. Statistical significance was determined by two-way ANOVA with Bonferroni *post hoc* test. Source data are provided as a Source Data file.

but significantly blocked the OVX-induced increase of marrow adipocytes (Fig. 5g, i). Although ALE showed a direct inhibitory effect on adipocyte differentiation of BMSCs (Supplementary Fig. 9a, b), the higher sensitivity of OVX mice than Sham mice to the antiadipogenic effect of ALE suggests that the inhibition of bone resorption may also contribute to the ALE-induced decrease of marrow adiposity. ALE did not notably affect osteogenic differentiation of BMSCs in vitro (Supplementary Fig. 9c, d). Consistently, the ALE-treated Sham and OVX mice showed no significant changes in osteoblast number and activity compared with the solvent-treated Sham and OVX mice, respectively, as shown by OCN immunostaining of femur sections (Fig. 5j, k) and ELISA for serum OCN (Fig. 5l).

We then evaluated the impact of ALE on VD3-induced vascular calcification in aged (16-month-old) Sham and OVX mice (Supplementary Fig. 10a). All OVX mice had significant reduced uterus weights compared with the solvent- or ALE-treated Sham mice (Supplementary Fig. 10b). ARS staining of abdominal aorta sections, vascular calcium content analysis, RUNX2 immunofluorescence staining, and qRT-PCR analysis for *Alpl*, respectively, demonstrated that the solvent-treated aged OVX mice displayed much higher ratios of calcium deposition areas (Fig. 5m, n), vascular calcium content (Fig. 5o), and ectopic osteogenic activities (Fig. 5p–r) in the vessel wall of the abdominal aortas than that in the solvent-treated aged Sham mice after intraperitoneal injection of VD3. However, OVX surgery did not exacerbate VD3-induced vascular calcification in 3-month-old young mice, as indicated by vascular calcium content analysis (Supplementary Fig. 11a) and ARS staining (Supplementary Fig. 11b, c), which were consistent with the evidence that YB-EVs released from young bone could not accelerate vascular calcification. ALE not only dramatically abolished the increases of calcium deposition areas, vascular calcium content, RUNX2 protein expression, and *Alpl* mRNA levels in abdominal aortas of the VD3-treated aged OVX mice, but also resulted in trends of decreases in most of these parameters in the VD3-treated aged Sham mice (Fig. 5m–r). Nevertheless, ALE at indicated gradient dosages did not directly inhibit calcification of VSMCs in vitro (Supplementary Fig. 12a, b), suggesting that the negative effect of ALE on vascular calcification is mediated by an indirect mechanism.

OVX also induced remarkable increases in the levels of serum calcium ions and inorganic phosphate in the mice subjected to VD3 administration (Supplementary Fig. 13a, b). ALE did not notably affect the levels of serum calcium ions and inorganic phosphate in the VD3-treated aged Sham mice (Supplementary Fig. 13a, b). In the VD3-treated aged OVX mice, ALE reduced the circulating levels of these two parameters, but the differences did not reach statistically significance (Supplementary Fig. 13a, b). These results suggest that the reductions of serum calcium ions and inorganic phosphate are not the primary mechanism that contributes to the ALE-induced inhibition of vascular calcification in the VD3-treated aged OVX mice.

Together, the above findings demonstrate the protective effects of ALE against bone-fat imbalance and VD3-induced vascular calcification in aged OVX mice, which may be associated with the ALE-induced inhibition of osteoclastic bone resorption and subsequent the reduction of AB-EVs release.

**MiR-483-5p and miR-2861 are enriched in AB-EVs and responsible for the AB-EVs-induced promotion of adipogenesis of BMSCs and calcification of VSMCs.** To explore the involvement of miRNAs in the AB-EVs-induced promotion of adipocyte formation and vascular calcification, the Agilent miRNA array was conducted to compare the miRNA expression profiles in AB-EVs and YB-EVs from mouse bone specimens. Totally, 1881 miRNAs were identified and 46 miRNAs were differentially expressed (absolute fold change ≥1.5; $P < 0.05$) in

AB-EVs and YB-EVs, among which 37 miRNAs were much higher and 9 miRNAs were much lower in AB-EVs compared with YB-EVs (Fig. 6a and Supplementary Data 1). Figure 6b shows the top ten most abundant miRNAs in AB-EVs relative to YB-EVs. miR-483-5p and miR-2861, which have been reported to positively modulate adipogenesis[39,40] and osteogenic transdifferentiation of VSMCs, respectively[41], were the most and second most abundant miRNAs in AB-EVs compared with YB-EVs (Fig. 6b), suggesting that these two miRNAs may be involved in the AB-EVs-induced promotion of BMSC adipogenesis and VSMC calcification.

The expression changes of miR-483-5p and miR-2861 were assessed in B-EVs from bone specimens of the mice at different ages (2-, 12-, and 18-month-old) by qRT-PCR. As shown in Fig. 6c, both miR-483-5p and miR-2861 showed an age-dependent increase of expression in B-EVs. Among different tissues, the bones (femurs and tibias) from 18-month-old aged mice showed the highest extent of up-regulation of these two miRNAs compared with that from 2-month-old young mice (Fig. 6d). The significantly increased expression of these two miRNAs was also observed in vessels (abdominal aortas) and liver tissues from 18-month-old aged mice than that from 2-month-old young mice, but other tissues, including spleen, lungs, kidneys, brain, heart, and muscles did not show significant increases in the levels of miR-483-5p and miR-2861 with aging (Fig. 6d). The enrichment of these two miRNAs was further demonstrated in serum EVs (Ser-EVs) from old people (67- to 73-year-old) compared with those from 27- to 31-year-old young people (Fig. 6e). The liver tissues-derived EVs (Liver-EVs) and Ser-EVs from 18-month-old aged mice (A-Liver-EVs and A-Ser-EVs) also had higher levels of these two miRNAs than those from 2-month-old young mice (Y-Liver-EVs and Y-Ser-EVs), but the extents of up-regulation were higher in A-Ser-EVs than that in A-Liver-EVs (Supplementary Fig. 14a). qRT-PCR analysis revealed that EVs isolated from OC$^{AB}$-CM also exhibited much higher levels of miR-483-5p and miR-2861 compared to EVs from OC-CM (Fig. 6f). These data suggest that aging induces the accumulation of miR-483-5p and miR-2861 in B-EVs and Liver-EVs, which are respectively released from bone matrix and liver tissues into circulation and then deposited in blood vessels. ARS and ORO staining showed that both Y-Liver-EVs and A-Liver-EVs did not induce remarkable effects on BMSC differentiation and VSMC mineralization, whereas Y-Ser-EVs induced a statistically significant increase of calcium nodule formation of BMSCs and A-Ser-EVs markedly increased BMSC adipogenesis and VSMC mineralization (Supplementary Fig. 14b–e), indicating that Liver-EVs are not the main contributor to the Ser-EVs-induced regulation of differentiation of BMSCs and VSMCs.

We then used specific agomiRs or antagomiRs to overexpress or silence miR-483-5p or miR-2861 in AB-EVs. Flow cytometry analysis revealed that incubation with the agomiR-NC-Cy3 indicator for 30 min resulted in more than 90% of AB-EVs positive for Cy3 fluorescent signals, as compared with the solvent-treated AB-EVs (Fig. 6g), suggesting that agomiRs or antagomiRs are able to enter into AB-EVs. ORO staining and qRT-PCR analysis for *Pparγ* revealed that the overexpression of miR-483-5p by agomiR-483-5p further augmented the ability of AB-EVs to stimulate lipid droplet formation and *Pparγ* expression of mouse BMSCs under adipogenesis, whereas the inhibition of this miRNA by antagomiR-483-5p significantly suppressed the pro-adipogenic effects of AB-EVs (Fig. 6h–j). ARS staining and qRT-PCR analysis for *RUNX2* showed that the agomiR-2861-pretreated AB-EVs induced a much higher extent of mineralized nodule formation and a higher level of *RUNX2* expression in human VSMCs compared with the agomiR-NC-pretreated AB-EVs, but the antagomiR-2861-pre-treated AB-EVs

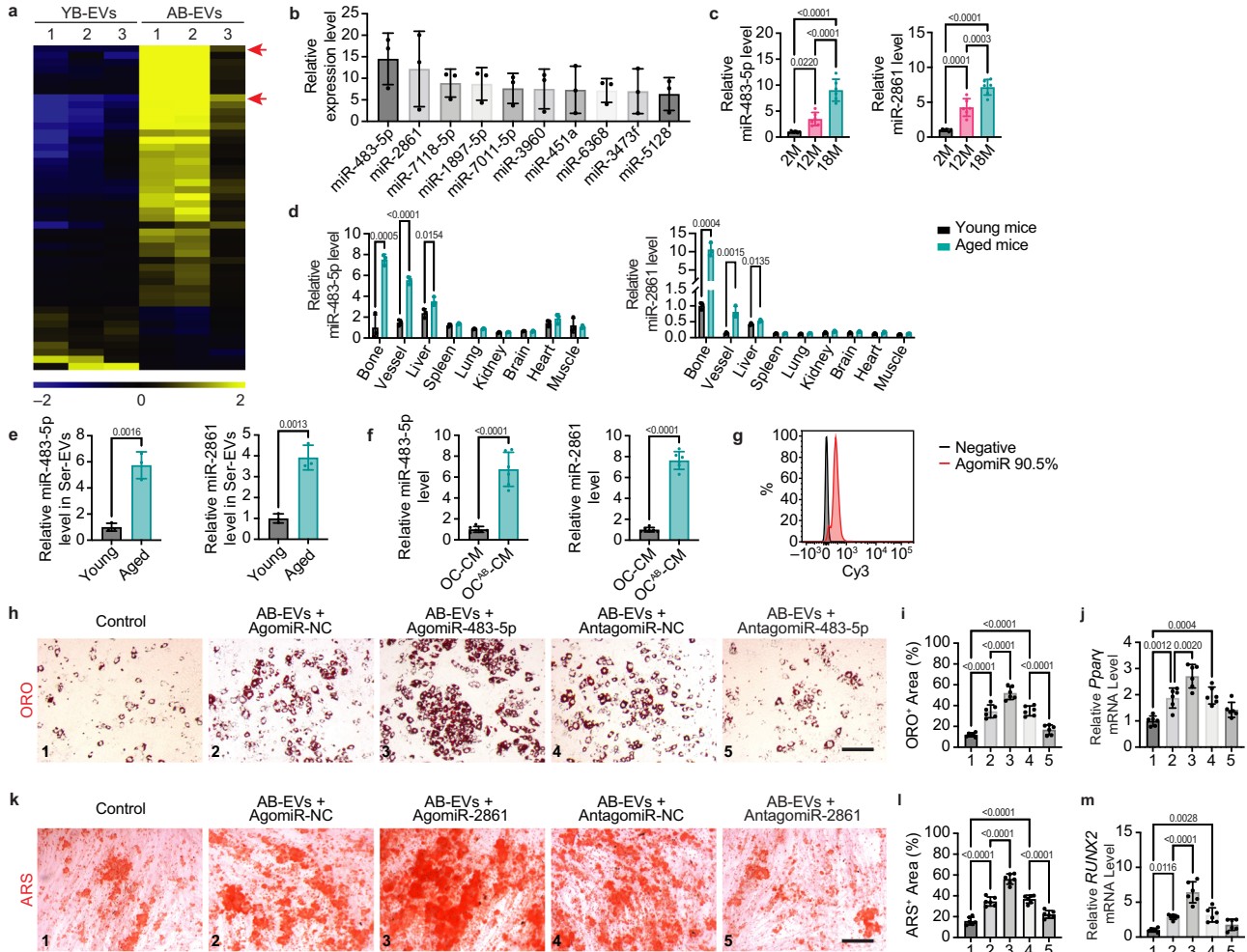

**Fig. 6 miR-483-5p and miR-2861 are enriched in AB-EVs and responsible for the AB-EVs-induced promotion of adipogenesis of BMSCs and calcification of VSMCs. a** Heatmap showing the differentially expressed miRNAs (absolute fold change ≥ 1.5; *P* < 0.05) between AB-EVs and YB-EVs. *n* = 3 biologically independent samples per group. **b** Top ten most abundant miRNAs in AB-EVs relative to YB-EVs. *n* = 3 biologically independent samples per group. **c-f**, qRT-PCR analysis of miR-483-5p and miR-2861 expression in B-EVs from bone specimens of the mice at different ages (**c**; *n* = 6 biologically independent animals per group), in different tissues from 3-month-old or 18-month-old mice (**d**; *n* = 3 biologically independent animals per group), in Ser-EVs from young (27- to 31-year-old) or old (67- to 73-year-old) human donors (**e**; *n* = 3 biologically independent donors per group), and in OC-CM and OC^AB^-CM (**f**; *n* = 6 biologically independent samples per group). **g**, Flow cytometry histograms showing the presence of agomiR-NC-Cy3 indicator in AB-EVs. *n* = 3 biologically independent samples per group. **h–j** ORO staining images (**h**), quantification of the percentage of ORO+ areas (red; **i**), and qRT-PCR for *Pparγ* expression (**j**) in BMSCs with different treatments under adipogenic induction. Scale bar: 50 μm. *n* = 6 biologically independent cells per group. **k–m**, ARS staining images (**k**), quantification of the percentage of ARS+ areas (red; **l**), and qRT-PCR for *RUNX2* expression (**m**) in VSMCs with different treatments under osteogenic induction. Scale bar: 50 μm. *n* = 6 biologically independent cells per group. Experiments in **a–g** were performed with at least three biological replicates per group without independent repetition. The other experiments were repeated independently three times with similar results. The illustrated results represented one of the three independent experiments. Data were presented as mean ± SD. Statistical significance was determined by one-way ANOVA with Bonferroni *post hoc* test (**c**, **i–j**, and **i–m**) or unpaired, two-tailed Student's *t*-test (**d–f**). Source data are provided as a Source Data file.

failed to promote mineralization and *RUNX2* expression of VSMCs (Fig. 6k–m). These findings indicate that miR-483-5p and miR-2861, respectively, act as the mediator of the AB-EVs-induced promotion of adipogenic differentiation of BMSCs and calcification of VSMCs.

**MiR-483-5p and miR-2861 contribute to AB-EVs-induced marrow adiposity and calcification paradox.** We finally determined whether miR-483-5p and miR-2861 were responsible for AB-EVs-induced bone-fat imbalance and vascular calcification, respectively. The 3-month-old young mice were administered with the antagomiR-483-5p- or antagomiR-NC-pre-treated AB-EVs or an equal volume of solvent by intramedullary injection

one time every two weeks for one month. As expected, μCT analysis revealed that the inhibition of miR-483-5p markedly impaired the ability of AB-EVs to induce trabecular bone loss, as indicated by the much higher levels of Tb. BV/TV, Tb. N, and Tb. Th, as well as the much lower level of Tb. Sp in antagomiR-483-5p-pretreated AB-EVs group compared to antagomiR-NC-pretreated AB-EVs group (Fig. 7a, b). No significant changes of Ct. Ar/Tt. Ar and Ct. Th were observed among the mice treated with solvent, antagomiR-NC-pretreated AB-EVs, or antagomiR-483-5p-pretreated AB-EVs (Supplementary Fig. 15a, b). Immunofluorescence staining for PLIN and qRT-PCR analysis for *Pparγ* showed profound increases of marrow adipocyte number and *Pparγ* expression in the bone tissues from mice treated

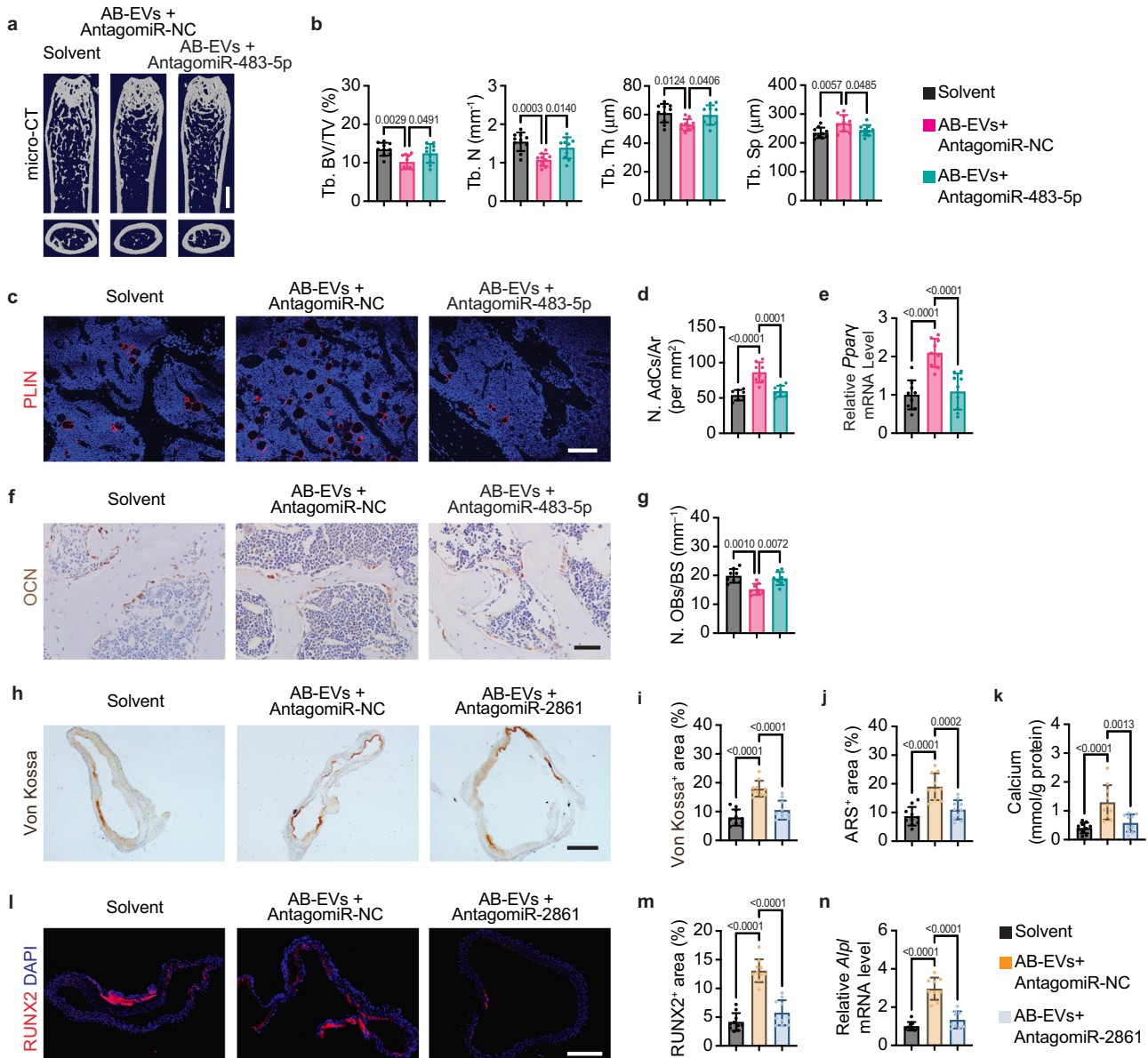

**Fig. 7 miR-483-5p and miR-2861 contribute to AB-EVs-induced marrow adiposity and calcification paradox. a** μCT-reconstructed images of femurs from 3-month-old young mice treated with solvent or AB-EVs pre-treated with antagomiR-NC or antagomiR-483-5p. Scale bars: 1 mm. **b** Quantification of Tb. BV/TV, Tb. N, Tb. Th, and Tb. Sp. $n = 10$ biologically independent animals per group. **c–d** PLIN immunofluorescence staining images (**c**) and quantification of the number of PLIN+ (red) adipocytes in bone marrow (**d**). Scale bar: 100 μm. $n = 8$ biologically independent animals per group. **e** qRT-PCR analysis of *Pparγ* expression. $n = 9$ biologically independent animals per group. **f–g**, OCN immunostaining images (**f**) and quantification of the number of OCN+ (brown) osteoblasts on BS (**g**). Scale bar: 50 μm. $n = 8$ biologically independent animals per group. **h–j** Von Kossa staining images (**h**), quantification of the percentage of Von Kossa+ (brownish black; **i**) and ARS+ (red; **j**) areas in abdominal aortas from the VD3-induced acute vascular calcification mouse models receiving solvent or AB-EVs pretreated with antagomiR-NC or antagomiR-2861. Scale bar: 200 μm. $n = 10$ biologically independent animals per group. **k** Vascular calcium content analysis. $n = 10$ biologically independent animals per group. **l–m**, RUNX2 immunostaining images (**l**) and quantification of the percentage of RUNX2+ areas (red; **m**). Scale bar: 200 μm. $n = 10$ biologically independent animals per group. **n**, qRT-PCR for *Alpl* expression. $n = 9$ biologically independent animals per group. All experiments were performed with at least eight biological replicates per group without independent repetition. Data were presented as mean ± SD. Statistical significance was determined by one-way ANOVA with Bonferroni *post hoc* test. Source data are provided as a Source Data file.

with the antagomiR-NC-pre-treated AB-EVs, but the antagomiR-483-5p-pre-treated AB-EVs failed to stimulate the accumulation of marrow adipocytes and *Pparγ* expression (Fig. 7c, d). Immunohistochemical staining for OCN indicated that the mice treated with the antagomiR-NC-pre-treated AB-EVs exhibited a significant reduction in osteoblast number, but the alteration did not occur in the mice treated with the antagomiR-483-5p-pretreated AB-EVs (Fig. 7f, g). These findings determine the essential role of

miR-483-5p in AB-EVs-induced bone-fat imbalance. The role of miR-2861 in the AB-EVs-induced promotion of vascular calcification was assessed in 3-month-old mice subjected to VD3 treatment. Von Kossa and ARS staining indicated that the inhibition of miR-2861 in AB-EVs by antagomiR-2861 prominently reduced, but did not entirely block, the AB-EVs-induced increases of calcium deposition areas in the mouse abdominal aortas (Fig. 7h, j). Vascular calcium content analysis further

**13**

determined the markedly decreased ability of AB-EVs to increase vascular calcium content after pre-treatment with antagomiR-2861 (Fig. 7k). RUNX2 immunostaining and RT-PCR analysis for *Alpl* revealed that the antagomiR-2861-pre-treated AB-EVs had a profoundly reduced capacity to augment ectopic osteogenic activities in the vessel wall of the abdominal aorta compared to the antagomiR-NC-pretreated AB-EVs (Fig. 7l–n). Inhibition of miR-2861 also significantly decreased, but did not entirely abolish, the ability of AB-EVs to increase serum calcium ions and inorganic phosphate in the mice with VD3-induced vascular calcification, indicating that miR-2861 partially contributes to the AB-EVs-induced increase of serum calcium ions and inorganic phosphate (Supplementary Fig. 16a, b). These results indicate that miR-2861 is a critical mediator of the AB-EVs-induced stimulatory effect on vascular calcification.

Collectively, the findings in our study suggest a new mechanism underlying the age- and menopause-associated calcification paradox (Fig. 8). With skeletal aging and menopause, osteoclastic bone resorption activity is relatively or absolutely increased. During bone resorption, AB-EVs secreted by osteocytes are released from the bone matrix into bone marrow, where AB-EVs deliver miR-483-5p to promote the expression of *PPARγ* and the differentiation of BMSCs into adipocytes rather than osteoblasts, thus leading to bone-fat imbalance and osteoporosis. A portion of AB-EVs are transported into circulation and deposited in blood vessels to stimulate *RUNX2* expression and osteogenic transdifferentiation of VSMCs via transferring miR-2861, thereby causing vascular calcification.

## Discussion

Aging and menopause are associated with high risks of both osteoporosis and vascular calcification[5–7]. The calcification paradox suggests a link between bone and vasculature and the existence of possible mechanisms leading to the coincidence of bone loss and vascular calcification during skeletal aging and menopause. There have been substantial evidences that a bone-vascular axis exists[42–45]. The relative or absolute increase of bone resorption in the elderly or postmenopausal women facilitates the mobilization of bone phosphorus and calcium into the blood, thus leading to abnormal mineral deposition in blood vessels[13]. Inflammation, oxidative stress and multiple biological factors such as fibroblast growth factor 23 (FGF-23), osteoprotegerin (OPG), osteopontin (OPN), fetuin-A, matrix Gla protein, and circulating calcifying cells, also implicate causal connections between bone metabolism and vascular calcification[43,46]. In this study, we found that B-EVs, which were usually embedded in the bone matrix, could be released during osteoclastic bone resorption and exert temporal-dependent regulatory effects on both bone homeostasis and vascular calcification. B-EVs from young bone (YB-EVs) were able to stimulate osteogenesis of BMSCs and enhance bone formation. The aged bone-derived B-EVs (AB-EVs), however, lost the ability to exert pro-osteogenic effect on bone, but could favor adipogenesis of BMSCs and mineralization of VSMCs in vitro and increase bone-fat imbalance as well as VD3- or adenine-induced vascular calcification in vivo. Adequate calcium and phosphorus supply is a prerequisite for both the occurrence of bone mineralization and the development vascular calcification[7,13]. Besides the direct stimulatory effect on VSMC mineralization, we found that AB-EVs contained a higher level of calcium compared with YB-EVs and could increase serum calcium ions and inorganic phosphate in both the acute and chronic mouse models of vascular calcification, suggesting that the direct transport of large amounts of calcium to circulation and the increase of phosphate in the blood may be another important mechanism by which AB-EVs promote vascular calcification. Our findings present an AB-EVs-mediated new mechanism of the calcification paradox and further enrich the knowledge of the bone-vascular axis.

Since their first discovery in growth plate cartilage, researchers have found that the biological role of matrix vesicles is not confined to acting as the sites for induction of mineral deposition, but also involves the modulation of the biological activity of cells within bone[13]. Nahar et al. have shown that matrix EVs isolated from the 4-week-old rats-derived tibial and femoral growth plate cartilages contain vascular endothelial growth factor (VEGF), bone morphogenetic proteins (BMPs), OPN, OCN, osteonectin (ON), and bone sialoprotein (BSP)[47]. Incubation with these matrix EVs increases ALP activity in chondrocytes, suggesting a positive role of cartilages-derived matrix EVs in chondro-osseous differentiation[47]. A recent study by Minamizaki et al. has reported that matrix EVs harvested from the extracellular matrix deposited by the cultured mouse osteoblastic MC3T3-E1 cells can inhibit osteoclast formation through the delivery of miR-125b[48]. In this study, the periosteum- and bone marrow-depleted whole bones, but not the growth plate cartilages only, the cultured chondrocytes, or mineralizing osteoblasts, were used to isolate matrix EVs by the collagenase II digestion method combined with the use of Optiprep™ density gradient ultracentrifugation. The morphology, diameter distribution, and surface marker expression profiles indicated the exosome identity of the isolated

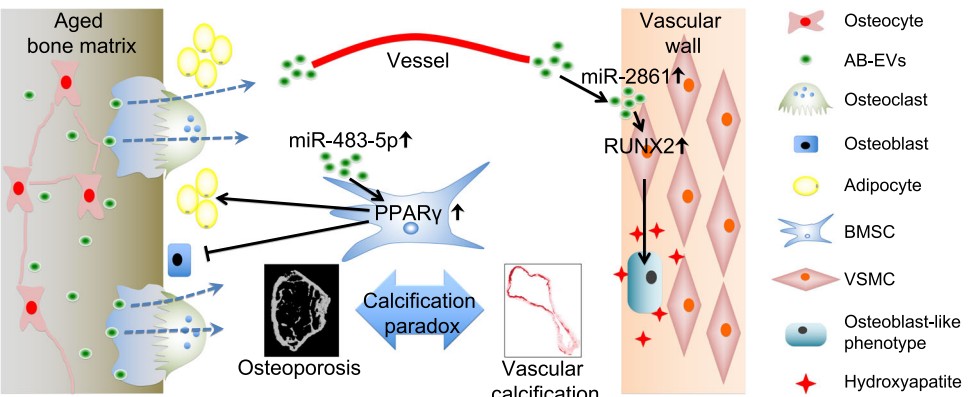

**Fig. 8 Schematic diagram showing the role of AB-EVs as a messenger for calcification paradox by favoring BMSC adipogenesis and VSMC calcification via transferring miR-483-5p and miR-2861.** During bone resorption, AB-EVs secreted by osteocytes are released from bone matrix into bone marrow, where AB-EVs promote *PPARγ* expression and adipogenic differentiation of BMSCs rather than osteogenesis by delivering miR-483-5p, thus leading to bone-fat imbalance and osteoporosis. AB-EVs are also transported into circulation and deposited in blood vessels to stimulate *RUNX2* expression and osteogenic transition of VSMCs via transferring miR-2861, thereby causing vascular calcification.

bone-derived matrix EVs. Since we did not further determine their intracellular origin, we just called them "B-EVs", but not "B-Exo". Similar to the previously reported B-EVs, we found that YB-EVs could exert anabolic effects on bone. More importantly, our study further determined that AB-EVs failed to induce bone benefits, but rather induced marrow fat accumulation and aorta calcification, indicating that the regulatory action of B-EVs on bone is age-dependent and B-EVs can be transported to circulation to affect vascular health.

B-EVs have always been considered to be released from hypertrophic chondrocytes and mineralizing osteoblasts[12,13,47–49]. Interestingly, here we found that the majority of B-EVs expressed SOST protein, an osteocyte-derived negative regulator of bone formation[22,23]. Only a very small proportion of B-EVs expressed the osteoblast marker COL I and the hypertrophic chondrocyte marker COL X. EVs secreted from osteocytes (OCY-EVs) also exhibited abundant expression of SOST and could trigger a B-EVs-like age-dependent regulation of BMSC differentiation and VSMC calcification. Using $Dmp1^{iCre}$; $Cd63^{em(loxp-mCherry-loxp-eGFP)3}$ mice and SOST immunostaining, we found the presence of eGFP- and SOST-double-positive OCY-EVs-like dot signals were at the perinuclear region of osteocytes, within the bone matrix, or in the vascular tissues of mice. These results suggest that osteocytes, the most abundant bone cells (>90%) located inside the mineralized bone matrix[50], are the major parent cells that release B-EVs and can be transported from bone to blood vessels under normal circumstance. Nevertheless, even though $Dmp1$-$Cre$ mice are widely used for gene deletion in osteocytes[51–54], $Dmp1$-$Cre$ has been reported to be able to inevitably target osteoblasts and some cells in other places[55]. DMP1 expression is also found in hypertrophic chondrocytes[56]. Thus, $Dmp1$-$Cre$ mice is not an ideal model to label and trace OCY-EVs. SOST protein is mainly produced by osteocytes[22,23]. However, there are evidences that other cell types such as hypertrophic chondrocytes are also able to produce SOST protein[57,58]. In our study, we did not find out and utilize an ideal Cre mouse model that targets only osteocytes. Although we found that the majority of B-EVs were negative for the characteristic markers that were expressed on EVs from osteoblasts or hypertrophic chondrocytes, our current results could not rule out that a minority of osteoblasts and hypertrophic chondrocytes may also contribute to the generation of B-EVs and play roles in the calcification paradox.

The calcification paradox is not only observed in osteoporosis patients, but also frequently occurs in patients with other age-related disorders, such as chronic kidney disease (CKD)[7]. The decrease of renal phosphorus excretion leads to increased serum phosphorus (hyperphosphatemia), which can combine with blood calcium to generate calcium phosphates and induce the osteogenic transition of VSMCs, resulting in CKD-mineral bone disorder (CKD-MBD) and vascular calcification[7,33]. The adenine-induced CKD model is widely utilized for studying chronic vascular calcification[33,34,59,60]. In our study, we found that AB-EVs could not only exacerbate VD3-induced acute vascular calcification, but were also able to aggravate vascular calcification in mouse models of adenine-induced CKD-related chronic vascular calcification, suggesting that the strategy targeting AB-EVs may be promising in treating vascular calcification in patients with CKD. We found that the donor rats of AB-EVs had much higher serum levels of BUN and CREA compared with the donor rats of YB-EVs, consistent with previous evidence that these two parameters increase with age in humans[61]. Aging contributes to the development of CKD, which in turn increases the risks of both osteoporosis and vascular calcification[62]. CKD can lead to increased bone resorption due to a state of inflammation[62]. Considering that AB-EVs could be released during bone resorption and exert positive effects on bone-fat imbalance and vascular calcification, the impaired renal function during aging may be a

factor leading to the increase of AB-EVs release to the bone marrow and blood, which finally induces bone-fat imbalance and vascular calcification. In our study, we did not further assess the effects of AB-EVs on bone phenotypes in animal models of CKD and test whether the reduction of AB-EVs release or the administration of YB-EVs could induce bone protective effects in the animals with CKD. Given their effects in aged mice and the association between aging and CKD, we hypothesized that the inhibition of release or function of AB-EVs or the supplementation of YB-EVs may provide bone benefits in individuals with CKD, which requires future investigation.

Bisphosphonates are one of the classic anti-osteoporosis drugs due to their inhibitory effects on osteoclastic bone resorption, but they can also hinder osteoclast formation and induce osteoclast apoptosis[63,64]. In recent years, evidences have shown that bisphosphonates have the ability to promote osteogenesis and inhibit adipogenesis of BMSCs[65,66], attenuate marrow adiposity in OVX mice[67], and prevent VSMC mineralization and vascular calcification[68,69]. There are also studies reporting that treatment with bisphosphonates decreases osteoblast activity and bone formation[69,70]. ALE is a second-generation bisphosphonate and commonly used as first-line therapy for osteoporosis by inhibiting bone resorption and increasing bone mass[71,72]. In our study, we found that this bone resorption inhibitor did not obviously affect osteogenic differentiation of BMSCs and mineralization of VSMCs at the current dose, but it could inhibit adipocyte differentiation of BMSCs, mitigate aging- and OVX-induced bone-fat imbalance, and suppress the OVX-induced aggravation of vascular calcification. Previous studies have suggested two possible mechanisms of the bisphosphonates-induced suppression of vascular calcification. One is the inhibition of bone resorption and subsequent the reduction of calcium release[73], and the other is the local effect on vascular wall[69]. Our findings showed that ALE could reduce the release of AB-EVs from bone matrix and almost entirely abolished the positive effects of the culture supernatant from bone-resorbing osteoclasts on adipogenesis of BMSCs and calcification of VSMCs, suggesting that the blockade of AB-EVs release because of the inhibition of bone resorption is another important mechanism by which bisphosphonates protect against bone-fat imbalance and vascular calcification. ALE possesses not only anti-resorptive activity, but also anti-osteoclastogenic and pro-apoptosis effects[63,64]. Thus, besides the direct inhibitory action on bone resorption, the suppression of osteoclast formation and induction of osteoclast apoptotic death by ALE could also cause the decline of AB-EVs release from the bone matrix due to the reduction of osteoclasts to resorb bones, thereby attenuating bone-fat imbalance and vascular calcification. In our study, we did not explore whether other available newer bisphosphonates (such as ibandronate and zoledronate[74]) can induce similar effects as ALE on AB-EVs release, bone-fat imbalance, and vascular calcification. Furthermore, the association between the inhibition of AB-EVs release by ALE and the protective effects of ALE against bone-fat imbalance and vascular calcification remains speculative based on our current evidences. We did not utilize an animal model to trace AB-EVs in vivo after ALE administration. There are multiple ways that bisphosphonates affect bone phenotypes. Future studies are required to find out a strategy that specifically inhibits AB-EVs release without affecting other processes and thereby deeply decipher the role of AB-EVs in the calcification paradox.

MiRNAs are small non-coding RNAs that regulate cell function by down-regulating the expression of target genes[75,76]. MiR-483-5p has been reported to facilitate adipogenesis of mouse pre-adipocyte 3T3-L1 cells by positively regulating the expression of $Ppar\gamma$[39]. It can also promote adipogenic differentiation of human adipose-derived MSCs by directly inhibiting $ERK1$ gene and

subsequently increasing *PPARγ* expression[40]. MiR-2861 is a miRNA that can target histone deacetylase 5 (HDAC5) to promote the expression of *Runx2*, thus stimulating osteoblast differentiation and osteogenic transdifferentiation of VSMCs[41,77]. In this study, we identified that miR-483-5p and miR-2861 were enriched in AB-EVs compared with YB-EVs and required for the AB-EVs-induced positive effects on adipogenesis of BMSCs and osteogenic transdifferentiation VSMCs, respectively. The results in vivo revealed that the knockdown of miR-483-5p and miR-2861 impaired the ability of AB-EVs to induce marrow adiposity and calcification paradox. These findings, along with the evidence of the much higher expression of these two miRNAs in bones, abdominal aortas, and Ser-EVs from aged mice than those from young mice, suggest that AB-EVs act as the carrier to transfer miR-483-5p and miR-2861 from bone matrix to bone marrow and then to the vasculature, where these miRNAs increase BMSC adipogenesis and VSMC mineralization, thereby promoting bone-fat imbalance and vascular calcification. Some other miRNAs may also partially contribute to the regulatory roles of AB-EVs in these processes, because either the antagomiR-483-5p- or antagomiR-2861-pretreated AB-EVs could still induce trend of positive effects on BMSC adipogenic differentiation and VSMC mineralization in vitro, as well as on adipogenesis and vascular calcification in vivo. It should be also noted that the miR-2861-abundant AB-EVs was not sufficient to augment osteogenic differentiation of BMSCs and bone formation like YB-EVs, which may be associated with the accumulation of bone-detrimental molecules, such as the pro-adipogenic miR-483-5p, in AB-EVs. Besides the aged bones, aged vessels, AB-EVs, and A-Ser-EVs, we also found that the aged liver and A-Liver-EVs also showed increased levels of miR-483-5p and miR-2861. However, A-Liver-EVs did not induce similar significant effects as AB-EVs on BMSC adipogenesis and VSMC mineralization, which may be associated with the insufficient levels of these two miRNAs or/and the enrichment of other miRNAs that have different regulatory effects on these processes in A-Liver-EVs. Nevertheless, we could not rule out the contribution of A-Liver-EVs to the increase of miR-483-5p and miR-2861 in A-Ser-EVs. EVs from other non-bone tissues such as Liver-EVs may be also involved in the development of aging-associated bone-fat imbalance and vascular calcification, which still requires future investigation.

There are many other limitations in this study. For assessment of marrow adipocytes, a limitation of our study is that we just performed immunofluorescence staining for the adipocyte marker PLIN, but did not use osmium tetroxide staining combined with μCT, to evaluate the changes of marrow adipocytes. During the assays for comparing the effects of OC-CM, OC^YB-CM, OC^AB-CM, and EVs from OC^YB-CM or OC^AB-CM on differentiation of BMSCs and VSMCs, the components in EVs from simple-cultured osteoclasts (OC-EVs) may be different with EVs from osteoclasts cultured with YB (OC^YB-EVs) or AB (OC^AB-EVs). There may exist some factors in OC^YB-EVs that can promote BMSC osteogenesis, and some molecules in OC^AB-EVs that can promote adipogenic differentiation of BMSCs and calcification of VSMCs, which may contribute to the pro-osteogenic effect of EVs in OC^YB-CM and the positive effects of EVs in OC^AB-CM on BMSC adipogenesis and VSMC mineralization. In other words, the bone-resorbing osteoclasts may also secrete functional EVs to affect bone and vessel phenotypes, which still warrants future investigation. Another limitation in our study is that we did not perform "dose-response" experiments to more carefully evaluate the effects of AB-EVs and YB-EVs on bone and vessel phenotypes in the normal physiology and the pathology of osteoporosis and vascular calcification. Currently, there is no evidence showing the physiological concentrations of AB-EVs and YB-EVs in the bone and vessel tissues. Future studies are

required to determine the physiological concentrations of AB-EVs and YB-EVs using accurate assays and investigate whether there exist dose-dependent responses in the AB-EVs- or YB-EVs-treated mice, which will be beneficial for more deeply deciphering the functional roles of bone-derived EVs in skeletal and vascular aging and for developing strategies to inhibit AB-EVs or utilize YB-EVs for therapeutic uses.

## Methods

**Ethics statement**. Animal experiments were approved by the Animal Ethics Committee and followed the Guidelines for the Care and Use of Laboratory Animals at Xiangya Hospital of Central South University. The collection and use of human bone and blood samples were approved by the Committee of Clinical Ethics at Xiangya Hospital of Central South University and the protocols followed were compliant with the ethical principles of the Helsinki Declaration. Written informed consents were obtained from all human donors. No study participant received compensation.

**Human samples**. The bone tissues were collected during surgical resection in patients (three 27−31 years old young women and three 67−73 years old women with osteoporosis) who underwent open reduction and internal fixation of tibial plateau fracture, or joint replacement due to osteoarthritis, in order to obtain YB-EVs and AB-EVs. Serum samples from these donors were collected to isolate exosomes. Besides osteoarthritis, the aged donors suffered from other diseases such as osteoporosis, thyroid nodule, pulmonary nodule, pneumonia, hyperlipidemia, hyperuricemia, or/and thyroid polypectomy. The young donors had no other diseases. The clinical information on these patients has been detailed in Supplementary Table 1.

**Cell isolation and culture**. BMSCs were isolated from the bone marrow of the femurs and tibias of C57BL/6 mice as described previously[78]. Briefly, the femurs and tibias were dissected under sterile condition and placed in α-MEM (SH30265.01; Hyclone, Logan, USA) with 1% Penicillin-Streptomycin (PS; P1400; Solarbio, Beijing, China). The epiphyses were removed and bone marrow was flushed out with α-MEM + 1% PS. The cells were washed with α-MEM + 1% PS and then cultured in α-MEM with 10% fetal bovine serum (FBS; 10099141; Gibco, Grand Island, USA) and 1% PS. After 24 h of culture, the culture medium was replaced with fresh complete α-MEM medium to remove the nonadherent cells. The adherent cells were cultured for another 3–5 days and passaged until reaching 90% confluence. Osteocytes and osteoblasts were isolated from the marrow-depleted mouse femurs and tibias with the protocol provided by Stern AR et al.[79]. Briefly, the bones (femurs and tibias) dissected under sterile condition were placed in α-MEM + 1% PS. After removing the periosteum, epiphyses, and bone marrow, the bones were cut into small pieces with lengths approximately 1 mm in Hank's balanced salt solution (HBSS; SH30031.02; Hyclone), followed by digestion with warmed collagenase type IA (300 U/mL; C9891; Sigma-Aldrich, St. Louis, MO, USA) for 25 min in α-MEM and subsequent 3 times washing with HBSS. The collagenase type IA-HBSS treatment was repeated again for 2 times. The solution obtained during the processes was aspirated and subjected to cell plating. Cells in the solution were plated in the collagen-coated culture plates and incubated in α-MEM with 5% fetal calf serum (FCS; 26010074; Gibco), 5% FBS, and 1% PS. These cells would be primarily osteoblasts. The bone pieces were then incubated with EDTA solution for 25 min and washed with HBSS, followed by digestion with collagenase type IA for 25 min and subsequent washing 3 times with HBSS. The solution aspirated during the processes was also subjected to cell plating and the obtained cells would be also mainly osteoblasts. After EDTA–HBSS–collagenase type IA–HBSS treatment for another 2 times, the fractions containing abundant osteocytes could be obtained. After mincing the remaining bone pieces, the resulting bone particles and the osteocyte-enriched fractions were seeded into the collagen-coated plates and incubated in α-MEM + 5% FCS + 5% FBS + 1% PS. Human aorta VSMCs (FH1244; FuHeng Biology, Shanghai, China) were incubated in F12K medium (SH30526.01; HyClone) + 10% FBS + 1% PS. RAW264.7 cells (CL-0190; Procell, Wuhan, China) were cultured in high glucose DMEM (11965092; Gibco) + 10% FBS + 1% PS. ATDC5 cells (HTX2427; Otwo Biotech, Shenzhen, China) were cultured in DMEM-F12 medium (C11330500B; Gibco) + 10% FBS + 1% PS. Cells grew at 37 °C and 5% $CO_2$. To obtain hypertrophic chondrocytes, ATDC5 cells were induced in the commercial chondrogenic medium (MUBMX-90042; Cyagen Biosciences, Guangzhou, China) for 21 days. The medium was changed every two days during the induction period.

**Isolation and characterization of EVs**. Bone specimens were obtained from young or old human donors, or from 2- or 18-month-old SD rats. After removing the periosteum and bone marrow, the remaining bones were physically crashed to small pieces using a tissue homogenizer and then subjected to digestion by collagenase II (; Gibco) for 3 h at 37 °C. The digested tissues were filtered through 70-μm nylon mesh membranes (352350; BD Falcon, San Jose, USA) and then centrifuged at $4500 \times g$ for 30 min and $10,000 \times g$ for 30 min at 4 °C. The supernatant was collected for Optiprep™ density

gradient ultracentrifugation to harvest B-EVs. For the collection of OCY-EVs, osteocytes were cultured in the complete α-MEM medium for 7 days. For the harvest of OB-EVs and HYPC-EVs, the osteoblasts and hypertrophic chondrocytes were incubated in their complete culture media for 48 h. The culture supernatants of these cells were harvested and subjected to sequential centrifugation ($300 \times g$ for 10 min, $2000 \times g$ for 30 min, and $10,000 \times g$ for 30 min) at 4 ℃[9], in order to remove dead cells and debris. For the isolation of Ser-EVs, the serum was also subjected sequential centrifugation as described above to obtain the supernatant. For the collection of Liver-EVs, the liver tissues were digested using collagenase II for 3 min at 37 ℃ and filtered with 70-μm nylon mesh membranes. After centrifugation at $4500 \times g$ for 30 min and $10,000 \times g$ for 30 min at 4 ℃, the supernatant was then harvested.

The supernatant obtained during the above processes was then subjected to EVs purification by Optiprep™ density gradient ultracentrifugation. Briefly, the supernatant was firstly filtered using a filter (SLGPR33RB; 0.22 μm pore size; Millipore, Billerica, USA) and centrifuged at $100,000 \times g$ for 3 h to pellet the crude EVs. Subsequently, the EV pellet was resuspended in 15 mL PBS and concentrated to about 1.5 mL in a 100 kDa Amicon Ultra-15 Centrifugal Filter Unit (UFC9100; Millipore) by centrifugation at $4000 \times g$ and 4 ℃. 1.33 mL of the ultrafiltration liquid was mixed with 6.67 mL Optiprep™ solution (60% w/v iodixanol; D1556; Sigma-Aldrich) to produce a 50% w/v iodixanol layer in a 38.5-mL polyallomer Beckman Coulter tube, followed by sequentially adding a discontinuous iodixanol gradient (8 mL of 40% w/v, 8 mL of 20% w/v, and 7 mL of 10% w/v iodixanol) and 2 mL PBS on the top of the 50% w/v iodixanol layer. The gradient was centrifuged for 18 h at $100,000 \times g$ and 4 ℃ using a SW 32 Ti rotor (Beckman Coulter; k-factor 204). 2 mL each of the density gradient fractions were harvested and subjected to NTA with the ZetaView® nanoparticle tracking analyzer (Particle Metrix, Meerbusch, Germany) for determining the EVs-rich fractions. After being diluted by PBS to 30 mL, the EVs-rich fractions were centrifuged for 3 h at $100,000 \times g$ and 4 ℃. The pellets of EVs were collected and resuspended with PBS.

The exosomal protein contents were quantified by BCA protein assay kit (70-PQ0011; Multi Sciences LTD., Hangzhou, China). The morphology and size distribution of EVs were tested by a H-7650 TEM (Hitachi, Tokyo, Japan) and NTA, respectively. The protein expression of exosomal markers, SOST, COL I, and COL X was assessed by flow cytometry. Anti-CD81 (sc-7637; 1:100), anti-CD63 (sc-5275, 1:100), and anti-TSG101 (sc-7964, 1:100) were obtained from Santa Cruz Biotechnology (Santa Cruz, USA). Anti-SOST (AF1589, 1:100), anti-COL I (14695-1-AP, 1:100), and anti-COL X (bs-0554R, 1:100) were purchased from R&D Systems (Minneapolis, USA), ProteinTech (Chicago, USA), and Bioss (Beijing, China), respectively. EVs incubated with secondary antibody (Jackson ImmunoResearch, West Grove, USA) alone served as negative controls.

For assays in vitro, EVs in different groups were used at the concentration of 20 μg/mL. For experiments in vivo, EVs were used at 200 μg (dissolved in 100 μL PBS for intravenous injection; dissolved in 10 μL PBS for intramedullary injection) or 500 μg (dissolved in 10 μL PBS for intramedullary injection) per time for each mouse.

**Preparation of bone-resorption CM.** For the preparation of the CM from young or aged bone-resorbing osteoclasts (OC^YB-CM or OC^AB-CM), the periosteum- and bone marrow-depleted young or aged bones were cut into small slices and equally distributed into several culture plates. The osteoclast progenitor RAW264.7 cells ($1 \times 10^5$ per well) were seeded onto the bone slices and cultured in the osteoclastic induction medium (high glucose DMEM + 10% FBS + 100 ng/mL RANKL). The cells cultured in the RANKL-containing osteoclastic induction medium without bone slices served as the control group. The medium was changed every other day. After 5 to 6 days of induction, many osteoclasts were formed and located on the surface of the bone slices. Osteoclasts began to resorb bone during this period. At days 6, the medium was replaced with a fresh osteoclastic induction medium (high glucose DMEM + EVs-depleted 10% FBS + 100 ng/mL RANKL) and the cells were incubated for another 2 days. Then, the OC^YB-CM, OC^AB-CM, or CM from the simple-cultured osteoclasts (OC-CM) were collected. If the newly formed osteoclasts were less after 5 to 6 days of induction, the cells were subjected to induction for a total of 8 days, followed by incubation in fresh osteoclastic induction medium (high glucose DMEM + EVs-depleted 10% FBS + 100 ng/mL RANKL) for additional 2 days to collect the CM. The obtained OC^YB-CM, OC^AB-CM, and OC-CM were stored at −80 ℃ before use or subjected to EVs purification by Optiprep™ density gradient ultracentrifugation. The isolated EVs and EVs-depleted CM were stored at −80 ℃ before use. For downstream experiments, the concentrations of CM from different groups, EVs in CM, and EVs-depleted CM were used at the concentrations of 20 μg/mL. RANKL was purchased from Peprotech (315-11C; Rocky Hill, USA).

**Exosome uptake assay.** B-Exo were labeled with DiI fluorescent dye (40726ES10; YEASEN Biotech, Shanghai, China) following the manufacturer's protocol. After removing the redundant dye, B-EVs were added to the cultures of BMSCs or VSMCs and incubated at 37 ℃ for 3 h. After discarding the culture supernatant and washing the cells with PBS, the cells were fixed in 4% Paraformaldehyde (PFA) for 15 min and incubated with DAPI (D1306; Invitrogen, Carlsbad, USA) to stain nuclei. The uptake of the red DiI-labeled B-EVs by these cells was determined by a Zeiss ApoTome fluorescence microscope (Jena, Germany).

**Osteogenic or adipogenic induction of BMSCs.** BMSCs (osteogenic induction: $1 \times 10^5$ cells per well; adipogenic induction: $2 \times 10^5$ cells per well) were seeded in 48-well plates and cultured for 24 h in complete medium, which was then replaced by osteogenic or adipogenic medium (MUBMD-90021 or MUBMD-90031; Cyagen Biosciences) supplemented with solvent, YB-EVs, AB-EVs, YB-OCY-EVs, AB-OCY-EVs, OC-CM, OC^YB-CM, OC^AB-CM, EVs from OC^YB-CM or OC^AB-CM, EVs-depleted OC^AB-CM, ALE (10 μM; 129318-43-0; Aladdin, Shanghai, China), OC^ALE-CM, OC^AB+ALE-CM, Y-Liver-EVs, A-Liver-EVs, Y-Ser-EVs, A-Ser-EVs, or AB-EVs pretreated with agomiR-483-5p, antagomiR-483-5p, agomiR-NC, or antagomiR-NC. The differentiation medium was changed every two days. For analyzing the expression of osteogenic or adipogenic genes, the cells were collected at 2 days after induction and processed for qRT-PCR. For detecting the formation of mineralized nodules or lipid droplets, the cells were stained with ARS solution (G1452; Solarbio) at 7 days after osteogenic induction or ORO solution (G1262; Solarbio) at 15 days after adipogenic induction. The percentages of ARS-positive (ARS^+) and ORO^+ areas were measured using Image-Pro Plus 6 software.

**Osteogenic induction of VSMCs.** VSMCs ($5 \times 10^4$ cells per well) were plated into 24-well plates and grew to 70% confluence in complete F12K medium, which was then changed to osteogenic medium (MUBMD-90021; Cyagen Biosciences) supplemented with solvent, YB-EVs, AB-EVs, YB-OCY-EVs, AB-OCY-EVs, OC-CM, OC^YB-CM, OC^AB-CM, EVs from OC^YB-CM or OC^AB-CM, EVs-depleted OC^AB-CM, ALE (0.1−10 μM), OC^ALE-CM, OC^AB+ALE-CM, Y-Liver-EVs, A-Liver-EVs, Y-Ser-EVs, A-Ser-EVs, or AB-EVs pre-treated with agomiR-483-5p, antagomiR-483-5p, agomiR-NC, or antagomiR-NC. The expression levels of SM22α, αSMA, RUNX2, and COL1A1 expression was assessed at 2 days after induction. ALP activity and mineralized nodule formation, respectively, were detected at 3 and 15 days after induction.

**Osteoclastic induction of RAW264.7 cells.** RAW264.7 cells ($1 \times 10^4$ cells per well) were seeded in 48-well plates and cultured in complete DMEM for 24 h, followed by replacing the medium with complete DMEM added with RANKL (100 ng/mL) + solvent, RANKL + YB-EVs, or RANKL + AB-EVs. After induction for 8 days, the cells were subjected to TRAP staining with a commercial kit (387A; Sigma-Aldrich) for the detection of osteoclasts (>three nuclei).

**CCK-8 assay.** VSMCs ($5 \times 10^3$ cells per well) were plated into 96-well plates and incubated in complete medium added with solvent or AB-EVs. 1, 2, 3 and 4 days later, the cells were cultured for additional 3 h in fresh complete medium with CCK-8 reagent (C008-3; 7Sea Biotech, Shanghai, China). Four wells without cells were also added with CCK-8 solution and served as the blank group. The absorbance at 450 nm was measured with a microplate reader (Varioskan LUX Multimode, Thermo Fisher Scientific, Waltham, USA).

**MiRNA array.** RNA from AB-EVs and YB-EVs was extracted using ExoQuick® Exosome Isolation and RNA Purification Kit (EQ806TC-1; System Biosciences, Mountain View, USA) and then subjected to RNA integration assessment by an Agilent Bioanalyzer 2100 (Agilent technologies, Palo Alto, USA). After RNA labeling and array hybridization, Agilent Microarray Scanner was used to scan the slides with default settings. Raw data were processed with AgiMicroRna R package. Differential expression analysis between AB-EVs and YB-EVs was performed and the genes with absolute fold change ≥1.5 and $P < 0.05$ were considered as differentially expressed.

**AgomiR or antagomiR transfection.** AgomiRs and antagomiRs were purchased from Ribobio Biotechnology (Guangzhou, China). AB-EVs were transfected with agomiR-483-5p (miR40004782-4-5), agomiR-2861 (miR40013803-4-5), agomiR-NC (miR4N0000001-4-5), agomiR-NC-Cy3 (miR04102-4-5), antagomiR-483-5p (miR30004782-4-5), antagomiR-2861 (miR30013803-4-5), or antagomiR-NC (miR3N0000001-4-5) at the concentration of 200 nM for 30 min at 37 ℃. The redundant agomiRs and antagomiRs were removed by centrifugation at $4,000 \times g$ for 5 min in a 100 kDa Amicon Ultra-4 Centrifugal Filter Unit (Millipore). The internalization of agomiR-NC-Cy3 by AB-EVs was assessed by flow cytometry. AB-EVs treated with other agomiRs and antagomiRs were used for subsequent experiments.

**qRT-PCR.** Total RNA was extracted with RNAiso Plus kit (9109; TaKaRa, Dalian, China) and subjected to cDNA synthesis with the GoScript™ Reverse Transcription System (PRA5000; Promega, Madison, USA) or miRNA First Strand cDNA Synthesis kit (B532451; Sangon Biotech, Shanghai, China). Then, qRT-PCR was performed with GoTaq® qPCR Master Mix (A6002; Promega) on FTC-3000 V1.0.3.44 real-time PCR system. GAPDH or U6 small nuclear RNA served as the reference genes for normalization. The primers were as follows: mouse-Runx2: forward, 5′-GACTGTGGTTACC GTCATGGC-3′, and reverse, 5′-ACTTGGTTTTTCATAAC AGCGGA-3′; mouse-Bglap: forward, 5′-CTGACCTCACAGATCCCAAGC-3′, and reverse, 5′-TGGTCTGA TAGCTCGTCACAAG-3′; mouse-Alpl: forward, 5′-CCAACTCTTTTGTGCC AGAG A-3′, and reverse, 5′-GGCTACATTGGTGTTGAGCTTTT-3′; mouse-Pparγ: forward, 5′-TCGCTGATGCACTGCCTATG-3′, and reverse, 5′-GAGAGGTCCACAGAGCTG

ATT-3′; *mouse-Cebpα*: forward, 5′-CAAGAACAGCAACGAGTACCG-3′, and reverse, 5′-GTCACTG GTCAACTCCAGCAC-3′; *mouse-Fabp4*: forward, 5′-AAGGTGAAGA GCATCATAACCCT-3′, and reverse, 5′-TCACGCCTTTCATAACACATTCC-3′; *mouse-Sm22α*: forward, 5′-ACTG CCTAGGCGGCCTTTA-3′, and reverse, 5′-ATGCC GTAGGATGGACCCTT-3′; *mouse-αSma*: forward, 5′-GTACCACCATGTACCCAG GC-3′, and reverse, 5′-GCTGGAAGGTAGACAGC GAA-3′; *mouse-Gapdh*: forward, 5′-AGGT CGGTGTGAACGGATTTG-3′, and reverse, 5′-TGTAGAC CATGTAGTTGAGGT CA-3′; *human-SM22α*: forward, 5′-GAGGAATTGATGGAAACCAC CG-3′, and reverse, 5′-CTCATGCCATAGGAAGGACCC-3′; *human-αSMA*: forward, 5′-CTTC AGCTTTCAGCTTCCCTGA-3′, and reverse, 5′-CAGAGCCCAGAGCCA TTGT-3′; *human-RUNX2*: forward, 5′-TGGTTACTGTCATGGCGGGTA-3′, and reverse, 5′-TC TCAGATCGTT GAACCTTGCTA-3′; *human-COL1A1*: forward, 5′-CTCCCCAGC CACAAAGAGTC-3′, and reverse, 5′-CCGTTCTGTACGCAGGTGAT-3′; *human-GAPDH*: forward, 5′-TCGACAGTCA GCCGCATCT-3′, and reverse, 5′-AGTTAAAA GCAGCCCTGGTGA-3′; miR-483-5p: 5′-CTC CCTTCTCTTCTCCCGTCTT-3′; miR-2861: 5′-CCGCCCGCCGCCAGGCCCC-3′; U6: 5′-GCGCGTCGTGAAGCGTTC-3′.

**Animals and treatments**. 3-, 15- and 16-month-old male or female C57BL/6J mice and male SD rats of 2- or 18-month-old were used and housed in barrier facilities on a 12 h light/dark cycle at temperature 18–22 °C and humidity 50-60%. 2- and 18-month-old male SD rats were used for the isolation of YB-EVs and AB-EVs, respectively. The serum samples of the donor rats were collected for assessing BUN and CREA on automated instruments in the Department of Clinical Laboratory in Xiangya Hospital of Central South University. Calcium ions and inorganic phosphate in YB-EVs and AB-EVs were examined with the commercial kits purchased from Nanjing Jiancheng Bioengineering Institute (C004-2-1 and C006-1-1; Nanjing, China). To assess the effects of AB-EVs and YB-EVs on bone-fat balance and the role of miR-483-5p in the AB-EVs-induced promotion of marrow adiposity, 3- or 15-month-old aged male mice were treated with 200 µg of YB-EVs, AB-EVs, antagomiR-483-5p- or antagomiR-NC-pre-treated AB-EVs, or an equal volume of solvent (PBS; 100 µL) by intramedullary injection one time every two weeks for one month. Then, the mice were killed after collecting the blood specimens. Spleen tissues were obtained, weighed, and processed for H&E staining using reagents from Solarbio. Whole blood samples were collected for evaluating the changes of lymphocytes and neutrophils in white blood cells. Serum samples were collected for assessing the concentrations of OCN and CTX-I using the commercial ELISA kits from Elabscience (E-EL-M0864c and E-EL-M3023; Wuhan, China). The femora were obtained for µCT analysis, biomechanical test, PLIN immunofluorescence staining, OCN immunohistochemical staining, and TRAP staining.

To evaluate the impacts of YB-EVs and AB-EVs on acute vascular calcification and the role of miR-2861 in the AB-EVs-induced aggravation of vascular calcification, 3-month-old male mice were intraperitoneally injected with VD3 (500 U/g/d) for continuous 4 days, followed by treatment with 200 µg of AB-EVs, YB-EVs, antagomiR-2861- or antagomiR-NC-pre-treated AB-EVs, or an equal volume of PBS by intravenous injection (100 µL per mice) at days 1 and 3, or treatment with 500 µg of AB-EVs or YB-EVs, or an equal volume of PBS (10 µL per mice) by intramedullary injection at day 1 only. To evaluate the impacts of YB-EVs and AB-EVs on chronic vascular calcification, 3-month-old female mice were fed a diet containing 0.25% adenine (Sigma-Aldrich) for 4 weeks to induce CKD and vascular calcification. During the feeding period, the mice were treated with 200 µg of YB-EVs or AB-EVs, or an equal volume of solvent (100 µL per mice) by intravenous injection at the first day of weeks 1 and 3. 11 days later, the abdominal aortas of the mice with acute or chronic vascular calcification were collected for qRT-PCR analysis for *Sm22α* and *αSma* expression, Von Kossa staining, ARS staining, vascular calcium content analysis, immunostaining for RUNX2, or/and qRT-PCR analysis for *Alpl* expression. Serum samples were collected for testing calcium ions and inorganic phosphate using the kits from Nanjing Jiancheng Bioengineering Institute (C004-2-1 and C006-1-1).

To investigate the effects of ALE on bone-fat balance, 16-month-old female mice were intragastrically administered with ALE (600 µg/kg/d) or solvent before and after bilateral OVX or sham operation once a week for three weeks (at 7 days before surgery; at 1 and 7 days after surgery). To detect the influences of ALE on vascular calcification, the 3- or 16-month-old mice subjected to OVX or sham operation were given ALE or solvent as described above and additionally treated with VD3 or solvent for continuous four days by intraperitoneal injection during the second week after OVX surgery. 18 days after OVX or sham operation, the mice were killed and their uteri were obtained and weighted. The serum, femora, and abdominal aortas were also collected and subjected to downstream analyses as described above.

*Cd63*em(loxp-mCherry-loxp-eGFP)3 mice were generated using the CRISPR/Cas9-mediated knock-in system by Cyagen Biosciences Inc. (Guangzhou, China). Briefly, the *loxp-cc-mCherry-polyA-loxp-cc-eGFP* cassette was inserted into *Cd63* gene at the site of stop codon in exon 8 to generate the homologous recombination donor vector, which was microinjected into the fertilized eggs of C57BL/6J mice combined with Cas9 mRNA and guide RNA (gRNA). The fertilized eggs were then transplanted into female C57BL/6J mice to produce chimeric mice. The positive F0 generation mice were identified by long fragment PCR and then cross-breeding with wild-type C57BL/6J mice to generate F1 mice, which were subjected to genotyping using PCR and DNA sequencing. The primers for determining the insertion of *loxP*-flanked *mCherry* gene

were as follow: P1: 5′-TGAGCAAGGGCGAGGAGGATAACA-3′; P2: 5′-CATGCCG CCGGTGGAGTGG-3′. The primers P1 and P2 will amplify a 689-bp product if the *loxP*-flanked *mCherry* was inserted, and no products can be amplified in the case of no insertion. *Cd63*em(loxp-mCherry-loxp-eGFP)3 mice were crossed with *Dmp1*iCre mice (Jackson Laboratory) to obtain *Dmp1*iCre; *Cd63*em(loxp-mCherry-loxp-eGFP)3 mice. The femora and abdominal aortas of these three types of mice were harvested and processed for the detection of mCherry and eGFP signals as well as SOST expression with a fluorescence microscope. Anti-eGFP and anti-SOST were obtained from Abcam (ab290; 1:10, 000; Cambridge, USA) and R&D Systems (AF1589; 1:100; Minneapolis, USA), respectively. The primers for the *iCre* transgene were as below: Dmp1-iCre-FRT-F: CACGTCCTCTCACTTCTCACG; Dmp1-iCre-FRT-R: CTTTGA CAGTGT CTTATCCAATAGCC. A 341-bp PCR product will be generated in mice hemizygous for the *Dmp1*iCre allele and a 240-bp PCR product will be yielded in the wild-type mice.

**µCT analysis**. The 4% PFA-fixed femora were subjected to µCT scanning by vivaCT80 scanner (SCANCO Medical AG, Bruettisellen, Switzerland) with a voltage of 50 kV, a current of 400 µA and a resolution of 18 µm per pixel. microCT V6.1 was used to collect the raw data. CT Analyser 1.11.0.0, µCTVol 2.2.0.0, and Dataviewer 1.4.3 were used for trabecular bone reconstruction and visualization in the distal femurs, respectively. The region for trabecular bone analysis was 5% of femoral length from 0.1 mm above the distal femoral growth plate, in order to analyze Tb. BV/TV, Tb. N, Tb. Th, and Tb. Sp. For cortical bone, the region for analysis was 5% of femoral length in the femoral mid-diaphysis for measuring Ct. Ar/Tt. Ar and Ct. Th.

**Biomechanical test**. Bone strength was evaluated by three-point bending test using a mechanical testing machine (Instron 3343; Instron, Canton, USA) as describe previously[80]. Briefly, the femur samples were placed on the lower supporting bars with two fulcrums spaced 8 mm. A vertical compression load with a constant rate of 5 mm/min was applied until fracture happened. The load-deformation curves were generated and used for calculating the ultimate load value (N).

**Histological analyses and immunostaining**. The femora and abdominal aortas were fixed for 48 h with 4% PFA, decalcified for one week in 18% EDTA (for bone only) and embedded in paraffin after dehydration. 5 µm-thick sections were made and processed for PLIN immunostaining using the antibody from Sigma-Aldrich (P1873; 1:300), immunohistochemical staining for OCN using the antibody from Servicebio (ab93876; 1:200; Wuhan, China), and TRAP staining with a kit from Sigma-Aldrich (387A). 5 µm-thick aortic sections were obtained and stained with Von Kossa reagent (G3282; Solarbio), ARS reagent (G1452; Solarbio), or RUNX2 antibody (12556; 1:800; Cell Signaling Technology, Danvers, USA). Secondary antibodies were purchased from Cell Signaling Technology (7074; 1:200) or Jackson Immuno Research (711-545-152 or 711-585-152; 1:400). Images were obtained under an optical microscope (Olympus CX31, Tokyo, Japan) or a Zeiss ApoTome fluorescence microscope. The number of adipocytes per square millimeter of marrow tissue (N. AdCs/Ar/mm$^2$), the numbers of osteoblasts and osteoclasts per millimeter of bone surface (N. OBs/BS/mm and N. OCs/BS/mm), and the percentages of Von Kossa$^+$, ARS$^+$, and RUNX2$^+$ areas for each aortic section were quantified.

**Histomorphometric analysis**. For double calcein labeling, 0.1% calcein (10 mg/kg; Sigma-Aldrich) was administered into the mice in different groups by intraperitoneal injection at 10 days and 3 days before sacrifice. The femora were obtained, fixed with 4% PFA for 48 h, dehydrated in ethanol, and then embedded in methyl methacrylate. 50-µm-thick bone sections were obtained and the signals were detected under a Zeiss ApoTome fluorescence microscope. The images were analyzed using the Image-Pro Plus 6 software to measure BFR/BS and MAR values.

**Vascular calcium content analysis**. The abdominal aortas were decalcified with 0.6 N HCl at 4 °C for 48 h. After testing the protein concentration, the calcium content in the supernatant was assessed using a commercial kit from Nanjing Jiancheng Bioengineering Institute (C004-2-1). The vascular calcium content was normalized to the concentration of protein.

**EV tracing**. To explore whether AB-EVs could be transported from bone to blood vessel wall after intramedullary injection, 500 µg AB-EVs were labeled with DiO dye following the manufacturer's instruction. Then, the DiO-labeled AB-EVs (dissolved in 10 µL PBS) or PBS only were administered into the bone marrow cavity of mice. 8 or 24 h later, the femora and abdominal aortas were collected and fixed in 4% PFA for 24 h. After dehydration and quick freezing, the tissues were embedded in OCT compound (4853; Tissue-Tek, Sakura Finetek, Torrance, USA) and cut into 10 µm-thick sections. After being stained with DAPI, the sections were washed and then detected with a fluorescence microscope (Zeiss Apotome).

**Statistical analysis**. Data were represented as mean ± SD and analyzed using GraphPad Prism 9. Unpaired, two-tailed Student's *t*-test was conducted for comparisons between two groups. One- or two-way ANOVA with Bonferroni *post hoc*

test was used for comparisons among multiple groups. $P < 0.05$ was considered to be significant.

**Reporting summary**. Further information on research design is available in the Nature Research Reporting Summary linked to this article.

## Data availability

MiRNA array data are deposited at ArrayExpress under accession number E-MTAB-11417. All the other data are available within the article, Supplementary Information file, Source data file, or from the corresponding authors upon reasonable request. Source data are provided with this paper. A reporting summary for this article is available as a supplementary information file. ZEN 2.3 pro, microCT V6.1, FTC-3000 V1.0.3.44, Skanlt RE 6.0.2, and WPS Office 3.8.1 software are used to collect data. The R scripts are used to reproduce the analyses and plots for miRNA array. The raw data in this article are available from the corresponding authors upon reasonable request. Source data are provided with this paper.

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

## Acknowledgements

We thank the volunteers for their contributions to this study. This work was supported by the National Natural Science Foundation of China (**Grant Nos. 81801395** (Z.-X.W.), **82072504** (H.X.)**, 81871822** (H.X.), **81522012** (H.X.), **81670807** (H.X.), **81702237** (C.-Y.C.), **81600699** (J.C.), **81701383** (Z.-Z.L.), **81974127** (Z.-Z.L.)), the Innovation Driven Project of Central South University (**Grant Nos. 2019CX014** (H.X.), **2018CX029** (Z.-Z.L.)), the Non-profit Central Research Institute Fund of Chinese Academy of Medical Sciences (**Grant No. 2019-RC-HL-024** (H.X.)), the Science and Technology Plan Project of Hunan Province (**Grant Nos. 2017XK2039** (H.X.), **2018RS3029** (H.X.)), the Science and Technology Innovation Program of Hunan Province (**Grant No. 2020RC4008** (H.X.)), the Hunan Province Natural Science Foundation of China (**Grant Nos. 2020JJ4914** (Z.-X.W.), **2017JJ3501** (Z.-X.W.), **2020JJ5883** (C.-Y.C.), **2020JJ5900** (X.-K.H.)), the China Postdoctoral Science Foundation (**Grant Nos. 2019T120717** (Z.-X.W.), **2017M612596** (Z.-X.W)**, 2018M632998** (X.-K.H.), **2020T130142ZX** (X.-K.H.)), the Key Laboratory of Luminescence and Real-Time Analytical Chemistry (Southwest University) Ministry of Education Open Funding (**Grant No. 201813** (Z.-X.W.)), the Hunan Provincial Innovation Foundation for Postgraduate (**Grant Nos. CX2018B045** (Y.Z.), **CX20190148** (Z.-W.L.)), and the Free Exploration Program of Central South University (**Grant No. 502221901** (S.-S.R.)).

## Author contributions

H.X., Z.-X.W. and C.-Y.C. conceived and designed the experiments. Z.-W.L., F.-X.-Z.L., Z.-X.W., S.-S.R., Y.-W.L., Y.-Y.W., G,-Q.Z., J.-S.G, J.-T.Z., Y.-J.T., H.Y., Y.-Y.L., and Z.-H.H. performed the experiments. Z.-W.L., F.-X.-Z.L., Z.-X.W., J.C., Q.W., C.-Y.C., Y.Z., Y.H., X.-K.W., L.R., Z.-Z.L., X.-K.H., L.-Q.Y., R.X. and H.X. analyzed and interpreted the data. H.X., C.-Y.C., and Z.-X.W. wrote the manuscript.

## Competing interests

The authors declare no competing interests.
