## [Peer Review File · Nature Communications]

Aged bone matrix-derived extracellular vesicles as a messenger for calcification paradoxREVIEWER COMMENTS

Reviewer #1 (Remarks to the Author):

In their paper Wang et al., show that exosomes released from aged bone matrix (AB-Exo) during bone resorption favor adipogenesis rather than osteogenesis of BMSCs and via this way augment calcification of vascular smooth muscle cells (VSMCs). They further demonstrate that MiR-483-5p and miR-2861 are enriched in AB-Exo and are essential for AB-Exo-induced bone-fat imbalance and exacerbation of vascular calcification respectively.

This paper is of particular interest as it, in an elegant way and based on a series of relevant in vitro and in vivo experiments, uncovers the potential mechanism underlying the so-called calcification paradox; i.e. the disturbed physiological bone mineralization going along with pathological ectopic calcification in the vessels, as seen in osteoporotic patients. Findings present evidence for an alternative, though novel hypothesis which on the long run may pave the way for alternative therapeutic strategies allowing adequate treatment at the level of the bone without increasing the risk for vascular calcifications and vice versa.

Results are based on straightforward experimental set-ups applying a series of innovative techniques which are highly complementary and confirmative to each other which allow the authors to draw solid conclusions.

Appropriate statistical analysis is applied although the number of cell cultures/animals in each of the experiments is rather limited which given the extensiveness of the study is acceptable the more since findings from one experiment are strongly confirmed by the next.

The paper is well-organized and comprehensibly written. Some linguistic revision is recommended. The extensive 'Methods' sections and the appropriate references the authors provide to more detailed descriptions of the methodologies should be sufficient for the work to be reproduced.

As the calcification paradox not only is an issue in patients with osteoporosis but is well known to occur in other populations also, in particular in patients with chronic and end-stage kidney disease suffering from either a high or low bone turnover disease, it might be worth to deal herewith in the 'Discussion' section also, and put the findings of the present study in perspective to this population in which the pathological mechanism underlying the bone disease(s) differ from that in osteoporotic patients with normal renal function. This would be of particular value from a therapeutic point of view also as there is a continuous search for effective treatments of both bone and calcified vessels in these patients.

Throughout the text the wording sounds rather suggestive as the authors frequently use 'likely', 'suggest', 'might' They should be more confirmative.

Both calcium and phosphorus and in particular their ionic product are major players in the development of vascular calcification. Since osteoclastic resorption not only goes along with delivery of exosomes from the bone compartment but also with the release of calcium and phosphorus, the authors should provide information on circulating levels of both these compounds. What about the calcium-phosphorus content in the vesicles? This is worth being discussed also. As the authors compare young and aged rats to each other they should provide info on renal function also (e.g. BUN, or creatinine ...).

As the authors took bone samples of rats and mice for immunohistochemistry, it might have been of interest to besides cellular staining also perform a bone histomorphometric analysis in terms of mineralized area, osteoid area/thickness/perimeter and even dynamic parameters such as bone formation rate, mineralization lag time etc

Page 11, first para: Was there any reason for the authors not to compare the AB-Exo with solvent group and not the YB-Exo group? This could have been relevant also. Same comment for the experiment in in the next paragraph

Page 11, line 6 from bottom: '... osteoclasts undergoing bone resorption ...'. Needs rewording as osteoclasts do not undergo resorption.

It is not clear from the text whether ALE either inhibits osteoclast formation or activity or induces osteoclast death or bone resorption as such.

Is there any reason why the authors opted to use ALE and not any of the other available (more recent) bisphosphonates?

Findings on the miRs are of particular interest. 46 miR's were differentially expressed. Only miR 485 (bone fat balance) and miR 2861 (on VC) were tested yielding nice results. Given the number of differentially expressed miR's one might reasonably expect some redundancy effect which

seemingly was not the case. Should be dealt with in the discussion session.

The authors used the VitD3 model of vascular calcification. This is an acute model in which vascular calcifications develop very abrupt. The question remains as to whether the observed effects would also be present in chronic models of vascular calcification such as e.g. the warfarin or adenine models, in which mechanisms underlying the calcifications may be different and perhaps more relevant from a clinical point of view.

Page 19, para 1, lines 5-8: '...showing that other cell types such as hypertrophic chondrocytes are also able to produce SOST protein. Thus, more in-depth studies are required to decipher the cell origin of B-Exo.' This might be relevant when comparing young to aged people as one could expect differences in the features of the exosomes when originating from different cell types.

Page 20, line 5: '... MiR-2861 is a miRNA that can promote osteoblast differentiation '. Is this the case in the present study. Did the authors check whether there was an effect on osteoblast differentiation at the level of the bone. This might be important as an improved bone formation may lead to a better incorporation of calcium (and phosphate) in bone leading to lower circulating levels and reduced risk for vascular calcification.

Page 26, Section 'Animals and treatment': '3-month-old male mice were intraperitoneally injected with VD3 (500 U/g/d) for continuous 4 days, followed by treatment with 200 or 500 µg of AB-Exo, YB-Exo, antagomiR-2861- or antagomiR-NC-pre-treated AB-Exo, or an equal volume of PBS by intravenous injection (100 µL per mice) at days 1 and 3, or intramedullary injection (50 µL per mice) at day 1 only'. It is not clear from this text to the reader whether the compounds were administered during the time of VitD3 treatment or thereafter. This is important since in the first case the observed effect is preventive whilst in the second case it is curative.

Vascular calcification in the present study in general is assessed by Alazarin staining which provides info on the localization of the calcifications, however, is less quantitative. Only in a limited number of experiments bulk calcium is measured (which from a quantitative point of view is more relevant). Is there any reason why bulk calcium was not assessed in all experiments?

Reviewer #2 (Remarks to the Author):

In this manuscript, the authors demonstrated that exosomes derived from the aged bone matrix (AB-Exo) during bone resorption induced adipogenesis rather than osteogenesis of BMSCs and calcification of VSMCs. They succeeded in solving some of the mystery of the "calcification paradox" from the viewpoint of exosomes. Their finding would contribute to senescence research and bone research, however, there are some comments to improve this study.

Major comments

1.

One of the main aims of this study is to show that AB-Exo positively modulates osteogenic transdifferentiation of VSMCs. The authors should more directly demonstrate that AB-Exo promotes the phenotypical change of VSMCs into osteoblast-like cells. Many papers describe that this phenotypical change is accompanied by a gain of osteogenic markers and loss of SMC markers, such as SM22 α and SM α -actin. In *in vitro* experiments, the authors only showed up-regulation in RUNX2 expression and ALP activity. The authors should more directly demonstrate osteogenic transdifferentiation of VSMCs by showing down-regulation in SM22 α and SM α -actin mRNA expression. (Cardiovascular Research, 2018 March 15; Volume 114, Issue 4, :590–600 / Aging, 2019 Aug 15; 11(15): 5445–5462.)

2.

Furthermore, the authors estimated Runx2 expression and ALP activity to show osteogenic transdifferentiation of VSMCs without enough explanation. Kurimoto et al. (Am J Physiol Heart Circ Physiol 2009 Nov; 297(5): H1673-84.) identified that ALP and type I collagen, in addition to Runx2, were significantly upregulated in the senescent VSMCs, suggesting their osteoblastic transition during the senescence. In this study, knockdown of either ALP or type I collagen significantly reduced the calcification in the senescent VSMCs. The knockdown of Runx2 significantly reduced the ALP expression and calcification; however, it did not reduce the type I collagen expression. This result suggested that there might be a Runx2-independent pathway involved in the osteoblastic transition of VSMCs. The osteoblastic transition mechanism of VSMCs has not been

fully elucidated, and it may be slightly different depending on the body condition; senescence, CKD, diabetes, hypertension, or osteoporosis. Indeed, it is well known that RUNX2 and ALP are the critical factors of osteoblast-like change of VSMCs, but the authors should describe the reason for selecting RUNX2 and ALP as the most significant factors of osteogenic differentiation of VSMCs, citing some more references in the manuscript.

3.

Many osteoclast-derived exosomes are contained in the culture medium in OC (YB and AB)-CM. The properties of exosomes derived from OC cultured with bone slices may differ from those derived from simple-cultured OC; it is possible that OC resorbing YB and OC resorbing AB release different exosomes. If the amounts of exosomes released by OC is small compared to those released by bone cells, that does not matter. However, in our experience, osteoclasts differentiated from 264.7 cells release considerable amounts of exosomes. The exosomes released by AB-resorbing OC do not contain molecules promoting osteogenesis of VSMCs?

4.

According to the "Methods" section, isolated exosomes were used at a 50 µg/ml concentration. This is a very high concentration. Tremendous amounts of exosomes would be needed to achieve this concentration. The concentration may be too far off from that in vivo. It would be desirable to observe the same phenomena at the lower concentration (at most 20 µg/ml) if they could.

5.

Most serious problem of this article lies on the separation of exosomes. The authors used ExoQuick system. Unfortunately, this commercialized kit have a problematic for exosome or other membrane particles separation since it is well recognized that this kit may allow huge amount of contamination with proteins and other materials. The authors need to clarify by alternative methods like ultracentrifugation and sucrose purification.

Minor comments

1.

Figure1 is very small and difficult to see. In Figure 1 (A), I could not see the morphology of the exosomes. In Figure 1 (E), I could not see the dots of the fluorescent dye.

2.

In the "Preparation of bone-resection OC-CM" section (page 22, line 538 - page 23, line 549), the authors should describe more detail about the method. Did they seed the RAW264.7 cells on the bone slices? How did they know that osteoclasts began to resorb bone? What medium did they use? Did they take any pictures of osteoclasts as supplemental data? Didn't they perform medium change? In our experience, osteoclasts derived from RAW264.7 cells 8-10 days after induction usually undergo apoptosis, and it is quite challenging to maintain mature osteoclasts without changing to fresh medium even if RANKL concentration is 100 ng/ml. Please describe in detail as much as possible.

Reviewer #3 (Remarks to the Author):

In this manuscript, Wang et al report potential paracrine/endocrine effects of bone-derived exosomal vesicles on osteoblast differentiation and vascular calcification. They propose an interesting model in which bone-derived exosomal content varies with aging. Increased levels of exosomal miRNA-483-5p may promote increased marrow adipogenesis at the expense of reduced bone formation. In addition, increased levels of exosomal miRNA-2861 may promote vascular calcification. While this is an interesting model with implications for common aging-associated changes, enthusiasm is limited due to largely indirect experimental evidence to support this hypothesis as detailed below.

Major concerns:

1. It is very difficult to evaluate the physiologic significance of the mouse exosome transfer experiments performed in Figures 2, 3 and 6. In the absence of a 'dose response', it is completely unclear if the amounts of exosomes transferred by intramedullary or IV injection reflect anything

close to normal biology. Along these lines, while the comparison of effects of exosomes from young versus aged rat bones is interesting, I am slightly concerned about cross species immunogenicity related to this experimental design. Moreover, the effects of exosomes isolated from other (non-bone) tissues (liver or blood, for example) should be assessed in this highly artificial model.

2. Definitive proof that bone-derived exosomes really do enter the circulation and reach vascular tissue remains lacking. The study in Figure 3F is interesting, but it remains possible that some of the intramedullar-injected exosomes directly enter the circulation in a non-physiologic manner. The authors really should take advantage of their novel Cd63 knockin reporter model here to study trafficking of bone-derived exosomes in a physiologic manner. For example, since Dmp1-Cre is not expressed in blood vessels, it would be very important to demonstrate eGFP+ signal in vessels in this model. Without this kind of information, it's really hard to know whether or not bone-derived exosomes actually enter the circulation under normal circumstances.

3. While it is interesting that distinct exosome-derived miRNAs might regulate BMSC differentiation and vascular calcification, information about the target genes for these distinct miRNAs is lacking.

4. The OVX experiment in figure 4 where mice are treated plus/minus alendronate confirms the expected effects of this bisphosphonate on bone mass. The relationship of this experiment to exosomes is completely speculative. Again, it would be powerful to use the Cd63 knockin model to show that alendronate blocks trafficking of exosomes from bone to blood vessels. Obviously there are multiple ways that alendronate can effect bone biology, and a relationship to exosome trafficking in vivo remains purely speculative at this point.

Minor concerns:

1. Additional clinical data is needed about the patients (young and old) from whom exosomal vesicles were isolated
2. Regarding the CD63 GFP knockin model, the authors should acknowledge that Dmp1-Cre is active in osteoblasts and other places (see PMID 28163952). For this reason, perhaps Dmp1-Cre isn't the most ideal Cre driver to use to try to label osteocyte-derived vesicles. The authors might consider Sost-Cre as an alternative. In addition, description of this model (Fig 1E) should include all appropriate negative controls due to problems with auto-fluorescence in bone sections.
3. For all the in vivo animal studies, the authors should present data with each mouse as an individual data point rather than using "box and plunger" graphs as is currently done. In addition, the sample size used for most of the in vivo studies (n=5) seems somewhat small.
4. Along these lines, the bone changes assessed throughout focus exclusively on trabecular bone. Whether these mild changes are physiologically-important remains unclear. The authors should also report whether exosomes have effects on cortical bone and/or bone strength.
5. For assessment of marrow adipocytes, the authors would ideally confirm effects using perilipin staining with osmium-based microCT.
6. For assessment of vascular calcification, using a complementary method to ARS staining would be ideal given challenges associated with this method.
7. The authors should comment on the increase in miR-2861 and 483 in liver with aging. Even modest liver-derived increases clearly could impact circulating exosome levels.

Marc Wein

Dear Editor,

On behalf of my co-authors, we thank you for giving us an opportunity to revise our manuscript entitled “**Aged bone matrix-derived exosomes as a messenger for calcification paradox**” (NCOMMS-20-48849A). We appreciate the editor and reviewers very much for their comments on this paper. These comments are all valuable and helpful for revising and improving our manuscript.

We have studied the reviewers’ and editor’s comments carefully and tried our best to revise our manuscript. We have submitted new version of our manuscript and the amendments are highlighted in yellow in the revised manuscript. Here we make the following revision in response to the reviewers’ and editor’s critiques. We hope that the revised manuscript will be accepted for publication in *Nature Communications*. Thank you very much for your time and your consideration.

We are looking forward to hearing from you soon. Thank you and best regards.

Yours truly,

Hui Xie and Chun-Yuan Chen

Reviewer #1 (Remarks to the Author):

In their paper Wang et al., show that exosomes released from aged bone matrix (AB-Exo) during bone resorption favor adipogenesis rather than osteogenesis of BMSCs and via this way augment calcification of vascular smooth muscle cells (VSMCs). They further demonstrate that MiR-483-5p and miR-2861 are enriched in AB-Exo and are essential for AB-Exo-induced bone-fat imbalance and exacerbation of vascular calcification respectively.

This is paper is of particular interest as it, in an elegant way and based on a series of relevant in vitro and in vivo experiments, uncovers the potential mechanism underlying the so-called calcification paradox; i.e. the disturbed physiological bone mineralization going along with pathological ectopic calcification in the vessels, as seen in osteoporotic patients. Findings present evidence for an alternative, though novel hypothesis which on the long run may pave the way for alternative therapeutic

strategies allowing adequate treatment at the level of the bone without increasing the risk for vascular calcifications and vice versa.

Results are based on straightforward experimental set-ups applying a series of innovative techniques which are highly complementary and confirmative to each other which allow the authors to draw solid conclusions.

Appropriate statistical analysis is applied although the number of cell cultures/animals in each of the experiments is rather limited which given the extensiveness of the study is acceptable the more since findings from one experiment are strongly confirmed by the next.

The paper is well-organized and comprehensibly written. Some linguistic revision is recommended.

Response: Thanks for the reviewer's valuable comments. We have tried our best to revise and improve our manuscript. The amendments are highlighted in yellow in the revised manuscript.

The extensive 'Methods' sections and the appropriate references the authors provide to more detailed descriptions of the methodologies should be sufficient for the work to be reproduced.

Response: Thanks for the reviewer's valuable suggestion. The methods section has been revised and more detailed information has been provided in the method section of the revised manuscript.

As the calcification paradox not only is an issue in patients with osteoporosis but is well known to occur in other populations also, in particular in patients with chronic and end-stage kidney disease suffering from either a high or low bone turnover disease, it might be worth to deal herewith in the 'Discussion' section also, and put the findings of the present study in perspective to this population in which the pathological mechanism underlying the bone disease(s) differ from that in osteoporotic patients with normal renal function. This would be of particular value from a therapeutic point of view also as there is a continuous search for effective treatments of both bone and calcified vessels in these patients.

Response: Thanks for the reviewer's valuable suggestion. The calcification paradox is not only observed in osteoporosis patients, but also frequently occurs in patients with other age-related disorders, such as chronic kidney disease (CKD) [1]. The decrease of renal phosphorus excretion leads to increased serum phosphorus (hyperphosphatemia), which can combine with blood calcium to generate calcium phosphates and induce the osteogenic transition of VSMCs, resulting in CKD-mineral bone disorder (CKD-MBD) and vascular calcification [1, 2]. The adenine-induced CKD model is widely utilized for studying chronic vascular calcification [2-5]. In our revised study, according to the reviewer's suggestion, we established chronic vascular calcification experiment models with CKD in 3-month-old female mice by freely feeding the mice with diet containing 0.25% adenine for 4 weeks (new **Figure 4j**). At the first day of week 1 and 3 during the feeding period, the mice were intravenously injected with YB-Exo, AB-Exo, or an equal volume of solvent (new **Figure 4j**). qRT-PCR analysis revealed that treatment with AB-Exo, but not YB-Exo, induced prominent reductions of mRNA levels of smooth muscle cell (SMC) markers including *Sm22 α* and *α Sma* in abdominal aortas of these mice (new **Figure 4k**), indicating the loss of vascular smooth muscle phenotype after AB-Exo administration. Von Kossa and ARS staining showed that AB-Exo profoundly increased calcium deposition lesion areas in the mouse abdominal aortas, whereas the changes were not observed in the YB-Exo-treated groups (new **Figure 4l-n**). The increase of calcium deposition in abdominal aortas of the AB-Exo-treated mice was further confirmed by vascular calcium content analysis (new **Figure 4o**). Immunofluorescence staining for the osteogenic factor RUNX2 and qRT-PCR analysis for *Alpl*, respectively, revealed that AB-Exo, but not YB-Exo, resulted in significant upregulations of RUNX2 protein and *Alpl* mRNA levels in the mouse abdominal aortas (new **Figure 4p-r**), indicating that AB-Exo induce the transition of the cells within the vessels into an osteogenic phenotype. These findings demonstrate that AB-Exo can exacerbate vascular calcification in mouse models of CKD-related chronic vascular calcification, suggesting that the strategy targeting AB-Exo may be promising in treating vascular calcification in patients with CKD.

In our revised study, according to the reviewer's suggestion, we also assessed the serum levels of renal function indicators including blood urea nitrogen (BUN) and creatinine (CREA) in 18-month-old aged rats and 2-month-old young rats, which were respectively used for the harvest of AB-Exo and YB-Exo for many experiments.

The results showed that the aged rats exhibited much higher serum levels of BUN and CREA compared with the young rats (new **Figure S3a-b**), consistent with previous evidence that these two parameters increase with age in humans [6]. Aging contributes the development of CKD, which in turn increases the risks of both osteoporosis and vascular calcification [7]. CKD can lead to increased bone resorption due to a state of inflammation [7]. Considering that AB-Exo could be released during bone resorption and exert positive effects on bone-fat imbalance and vascular calcification, the impaired renal function during aging may be a factor leading to the increase of AB-Exo release to the bone marrow and blood, which finally induces bone-fat imbalance and vascular calcification. We apologize that we did not further assess the effects of AB-Exo on bone phenotypes in animal models of CKD and test whether the reduction of AB-Exo release or the administration of YB-Exo could induce bone protective effects in animal models of CKD. Given their effects in aged mice and the association between aging and CKD, we hypothesized that the inhibition of release or function of AB-Exo or the supplementation of YB-Exo may provide bone benefits in individuals with CKD, which requires future investigation.

The results regarding the effects of AB-Exo and YB-Exo in CKD-related chronic vascular calcification YB mouse models and the serum levels of BUN and CREA in the donor rats of AB-Exo and YB-Exo have been added to new **Figure 4j-r** and **S3a-b** in the revised version. The discussion on the association between the calcification paradox and CKD, the relation of this issue with our findings, and our limitation has been added to **line 1-25, paragraph 4** in the discussion section of the revised manuscript. Thanks!

new **Figure 4j-r**

j, Experimental design of adenine-induced chronic vascular calcification mouse models treated with solvent, YB-Exo, or AB-Exo by intravenous injection. **k**, qRT-PCR analysis of *Sm22 α* and *α Sma* expression in abdominal aortas of mice in (**j**). $n = 9$ per group. **l-n**, Von Kossa staining images (**l**) and quantification of the percentages of Von Kossa⁺ (**m**) and ARS⁺ (**n**) areas. Scale bar: 200 μ m. $n = 10$ per group. **o**, Vascular calcium content measurement. $n = 10$ per group. **p-q**, RUNX2 immunofluorescence staining images (**p**) and quantification of the percentage of RUNX2⁺ areas (**q**). Scale bar: 200 μ m. $n = 10$ per group. **r**, qRT-PCR for *Alpl* expression. $n = 9$ per group. ** $P < 0.01$, *** $P < 0.001$, **** $P < 0.0001$.

new **Figure S3a-b**

a-b, Serum levels of BUN (**a**) and CREA (**b**) in young and aged donor rats. $n = 10$ per group. ** $P < 0.01$, *** $P < 0.001$

Throughout the text the wording sounds rather suggestive as the authors frequently use ‘likely’, ‘suggest’, ‘might’ ... They should be more confirmative.

Response: Thanks for the reviewer’s valuable suggestion. We have carefully checked the related statements and made correction depending on the actual conditions in the revised manuscript.

Both calcium and phosphorus and in particular their ionic product are major players in the development of vascular calcification. Since osteoclastic resorption not only goes along with delivery of exosomes from the bone compartment but also with the release of calcium and phosphorus, the authors should provide information on circulating levels of both these compounds.

Response: According to the reviewer’s valuable suggestion, in the revised study, we assessed the circulating levels of calcium ions and inorganic phosphate in the VD3-induced acute vascular calcification and adenine-induced chronic vascular calcification mouse models receiving different treatments. The results showed that YB-Exo induced a significant increase of serum calcium ion in the mice with VD3-induced acute vascular calcification and a trend of increase of serum inorganic phosphate in both the acute and chronic mouse models of vascular calcification. However, treatment with AB-Exo resulted in marked increases of both serum calcium

ions and inorganic phosphate in these two models of vascular calcification, and the effects were much higher than that of YB-Exo. After pre-treatment with antagomiR-2861, but not antagomiR-NC, the ability of AB-Exo to increase serum calcium ions and inorganic phosphate was significantly decreased, but did not entirely abolish, in the mice with VD3-induced acute vascular calcification, indicating that miR-2861 partially contributes to the AB-Exo-induced increase of serum calcium ions and inorganic phosphate. We also tested the levels of serum calcium ions and inorganic phosphate in the VD3-treated aged Sham mice and OVX mice receiving solvent or ALE treatment. The results showed that OVX induced remarkable increases in the levels of serum calcium ions and inorganic phosphate. The resorption inhibitor ALE did not notably affect the levels of these parameters in the VD3-treated aged Sham mice. In the VD3-treated aged OVX mice, ALE reduced the circulating levels of these two parameters, but the differences did not reach statistical significance. These findings, along with evidences showing the occurrence of notable vascular calcification in VD3-treated OVX mice and the inhibition of vascular calcification in OVX + ALE mice in new **Figure 5m-r**, suggest that the reductions of serum calcium ions and inorganic phosphate are not the primary factors that mediate the ALE-induced inhibitory effects on vascular calcification in the VD3-treated aged OVX mice. The above results have been added to new **Figure S6a-b**, **S13a-b**, and **S16a-b**, and the description of the results has been added to **line 1-9, paragraph 14, line 1-8, paragraph 19**, and **line 29-33, paragraph 24** in the results section of the revised version. Thanks!

new **Figure S6a-b**

a-b, Serum levels of calcium ions and inorganic phosphate in VD3-induced acute (**a**) and adenine-induced chronic (**b**) vascular calcification. $n = 7$ per group. * $P < 0.05$, ** $P < 0.01$, *** $P < 0.001$, **** $P < 0.0001$.

new **Figure S13a-b**

a-b, Serum levels of calcium ions (**a**) and inorganic phosphorus (**b**) in the VD3-administrated aged Sham and OVX mice treated with solvent or ALE. $n = 5$ per group. * $P < 0.05$, ** $P < 0.01$.

new **Figure S16a-b**

a-b, Serum levels of calcium ions (**a**) and inorganic phosphorus (**b**) in the VD3-induced vascular calcification mouse models receiving solvent or AB-Exo pre-treated with antagomiR-NC or antagomiR-2861. $n = 7$ per group. * $P < 0.05$, ** $P < 0.01$, *** $P < 0.001$, **** $P < 0.0001$.

What about the calcium-phosphorus content in the vesicles? This is worth being discussed also.

Response: Thanks for the reviewer's valuable question and suggestion. We assessed the levels of calcium and phosphorus in AB-Exo and YB-Exo using the commercial kits. The results showed that both calcium and phosphorus could be detected in AB-Exo and YB-Exo, but the level of calcium in AB-Exo was much higher than that in YB-Exo. We have added the results to new **Figure S6c** and the description of the results to **line 9-13, paragraph 14** in the results section of the revised version. In our revised study, we also found that treatment with AB-Exo resulted in marked increases of both serum calcium ions and inorganic phosphorus in both the acute and chronic mouse models of vascular calcification, and the effects were much higher than that of YB-Exo (new **Figure S6a-b**). Adequate calcium and phosphorus supply is a prerequisite for both the occurrence of bone mineralization and the development vascular calcification [1, 8]. Besides their direct stimulatory effect on VSMC mineralization, the direct transport of large amounts of calcium to circulation and the increase of phosphate in the blood may be another important mechanism by which AB-Exo promote vascular calcification. The discussion on this issue has been added to **line 18-24, paragraph 1** in the discussion section of the revised manuscript.

Thanks!

new Figure S6

a-b, Serum levels of calcium ions and inorganic phosphate in VD3-induced acute (**a**) and adenine-induced chronic (**b**) vascular calcification. $n = 7$ per group. **c**, Calcium ion and inorganic phosphate contents in YB-Exo and AB-Exo. $n = 4$ per group. * $P < 0.05$, ** $P < 0.01$, *** $P < 0.001$, **** $P < 0.0001$.

As the authors compare young and aged rats to each other they should provide info on renal function also (e.g. BUN, or creatinine ...).

Response: In our study, besides the human and mouse bone specimens, we also used bone specimens from 18-month-old aged rats and 2-month-old young rats, respectively, to obtain AB-Exo and YB-Exo for various experiments *in vivo* and *in vitro*, because animal samples are easily attainable than human specimens and a larger quantity of B-Exo can be obtained from rat bone than that from an equal number of mouse bone. According to the reviewer's valuable suggestion, we assessed the serum levels of renal function indicators including blood urea nitrogen (BUN) and creatinine (CREA) in 18-month-old aged rats and 2-month-old young rats, which were respectively used for the harvest of AB-Exo and YB-Exo in our revised study. The results showed that the aged rats exhibited much higher serum levels of BUN and CREA compared with the young rats, consistent with previous evidence that these two parameters increase with age in humans [6]. We have added the results to new **Figure S3a-b** and the description of the results to **line 8-10, paragraph 7** in the results section of the revised version. Aging contributes the development of chronic kidney disease (CKD), which in turn increases the risks of both osteoporosis and vascular

calcification [7]. CKD can lead to increased bone resorption due to a state of inflammation [7]. Considering that AB-Exo could be released during bone resorption and exert positive effects on bone-fat imbalance and vascular calcification, the impairment of renal function during aging may be a factor leading to the increase of AB-Exo release to the bone marrow and blood, which finally induces bone-fat imbalance and vascular calcification. The discussion on this issue has been added to **line 11-19, paragraph 4** in the discussion section of the revised manuscript. Thanks!

new **Figure S3a-b**

a-b, Serum levels of BUN (**a**) and CREA (**b**) in young and aged donor rats. $n = 10$ per group. ** $P < 0.01$, *** $P < 0.001$

As the authors took bone samples of rats and mice for immunohistochemistry, it might have been of interest to besides cellular staining also perform a bone histomorphometric analysis in terms of mineralized area, osteoid area/thickness/perimeter and even dynamic parameters such as bone formation rate, mineralization lag time etc

Response: According to the reviewer's valuable suggestion, we assessed whether the intramedullary injection of AB-Exo or YB-Exo one time per two weeks for one month could affect bone dynamic parameters in young (3-month-old) mice and aged (15-month-old) mice by bone histomorphometric analysis after calcein double-labeling. The representative images of calcein double labeling and quantitative data of bone formation rate per bone surface (BFR/BS) and mineral apposition rate (MAR) have been added to new **Figure 3e-f** in the revised version. The results revealed that YB-Exo treatment induced much higher levels of new bone formation and mineralization in both young and aged mice, whereas the AB-Exo-treated young and aged mice exhibited trend of or significant lower levels of these indicators compared with the age-matched mice treated with solvent or YB-Exo. The results were consistent with the changes of osteoblast number and activity showing in new **Figure 3j-l**. The description of the results on calcein double-labeling has been added

to **line 1-5, paragraph 8** in the results section of the revised manuscript. Thanks!

new **Figure 3e-f**

e-f, Calcein double labeling of trabecular bones (e) and quantification of BFR/BS and MAR (f) of femurs from young or aged mice treated with solvent, YB-Exo, or AB-Exo. Scale bar: 25 μm . n = 5 per group. * $P < 0.05$, ** $P < 0.01$, *** $P < 0.001$, **** $P < 0.0001$.

Page 11, first para: Was there any reason for the authors not to compare the AB-Exo with solvent group and not the YB-Exo group? This could have been relevant also. Same comment for the experiment in in the next paragraph

Response: Thanks for the reviewer's valuable question. In **Figure 3a-e** in our original study and **Figure 4a-i** in our revised study, we determined that the intravenous injection of AB-Exo significantly exacerbated vascular calcification in the mice with VD3-induced acute vascular calcification. In **Figure 4j-r** in our revised study, we further demonstrated that AB-Exo had the ability to aggravate vascular calcification in mouse models of adenine-induced chronic vascular calcification after intravenous injection. However, treatment with YB-Exo did not induce notable effects in both the acute and chronic mouse models of vascular calcification. Thus, we did not set the YB-Exo group in the following experiments for investigating whether AB-Exo could be transported from bone to blood and exacerbate VD3-induced vascular calcification in mice after intramedullary injection (**Figure 3f-k** in our original study; **Figure 4s-v** and **S7a-d** in our revised study).

For experiments in **Figure 4a-d** in our original study and **Figure 5a-d** in our revised study, we aimed to investigate the bone resorption inhibitor ALE could inhibit the release of AB-Exo from aged bone, and thereby suppress the ability of conditioned media from osteoclasts cultured with the aged bone slices (OC^{AB} -CM) to augment BMSC adipogenesis and VSMC calcification. In the following study, we further

tested whether OVX could exacerbate VD3-induced vascular calcification in aged mice, and ALE could attenuate bone-fat imbalance and VD3-induced vascular calcification in the aged OVX mice (**Figure 4e-o** in our original study and **Figure 5e-r** in our revised study), in order to explore an association between the inhibitory effect of ALE on AB-Exo release during bone resorption and the protective effects of ALE against bone-fat imbalance and VD3-induced vascular calcification in aged OVX mice. Since we did not assess the effects of ALE on bone phenotypes and vascular calcification in young mice during these experiments, in our opinion, there seemed no need to set OC^{YB}-CM and OC^{YB+ALE}-CM groups in **Figure 4a-d** in our original study and **Figure 5a-d** in our revised study. Thanks!

Page 11, line 6 from bottom: ‘... osteoclasts undergoing bone resorption ...’. Needs rewording as osteoclasts do not undergo resorption.

Response: Thanks for the reviewer’s valuable suggestion. The sentence “ALE did not significantly affect the production of exosomes in osteoclasts cultured without aged bone slices, but notably reduced the concentration of exosomes in the culture supernatant of osteoclasts undergoing bone resorption” has been revised as follow: ALE did not significantly affect the production of exosomes in osteoclasts cultured without aged bone slices, but notably reduced the concentration of exosomes in the culture supernatant of osteoclasts in the presence of aged bone slices. The revised sentence is displayed in **line 8-11, paragraph 16** in the results section of the revised manuscript. Thanks!

It is not clear from the text whether ALE either inhibits osteoclast formation or activity or induces osteoclast death or bone resorption as such.

Response: Bisphosphonates are one of the classic anti-osteoporosis drugs well-known for their inhibitory effect on osteoclastic bone resorption, but they can also hinder osteoclast formation and induce osteoclast apoptosis [9, 10]. ALE is a bisphosphonate that possesses anti-osteoclastogenic, anti-resorptive, and pro-apoptosis effects [9, 10]. In our study, we found that ALE could reduce the release of AB-Exo from bone matrix and almost entirely abolished the positive effects of the culture supernatant from bone-resorbing osteoclasts on adipogenesis of BMSCs and calcification of VSMCs, suggesting that the blockade of AB-Exo release through the inhibition of bone resorption is an important mechanism by which ALE protects against bone-fat

imbalance and vascular calcification. Besides the direct inhibitory action on bone resorption, the suppression of osteoclast formation and induction of osteoclast apoptotic death by ALE could also cause the decline of AB-Exo release from the bone matrix due to the reduction of osteoclasts to resorb bones, thereby attenuating bone-fat imbalance and vascular calcification. The functional roles of ALE in osteoclast formation, activity, and apoptosis, as well as the related discussion on this issue have been stressed in **line 1-3, paragraph 5** and **line 15-25, paragraph 5** in the discussion section of the revised manuscript. Thanks!

Is there any reason why the authors opted to use ALE and not any of the other available (more recent) bisphosphonates?

Response: ALE is a second-generation bisphosphonate and commonly used as first-line therapy for osteoporosis by inhibiting bone resorption and increasing bone mass [11, 12]. Thus, in this study, we selected ALE to explore whether the inhibition of bone resorption could reduce the release of AB-Exo and subsequent protect against bone-fat imbalance and vascular calcification in the aged OVX mice. We apologize for that we did not use other available newer bisphosphonates (such as ibandronate and zoledronate [13]) for these experiments and explored whether they could induce similar effects as ALE on AB-Exo release, bone-fat imbalance, and vascular calcification. The reason why we selected ALE and our limitation have been stressed in **line 25-28, paragraph 5** in the discussion section of the revised manuscript. Thanks!

Findings on the miRs are of particular interest. 46 miR's were differentially expressed. Only miR 485 (bone fat balance) and miR 2861 (on VC) were tested yielding nice results. Given the number of differentially expressed miR's one might reasonably expect some redundancy effect which seemingly was not the case. Should be dealt with in the discussion session.

Response: Thanks for the reviewer's valuable suggestion. In this study, we identified that 46 miRNAs were differentially expressed (absolute fold change ≥ 1.5 ; $P < 0.05$) in AB-Exo and YB-Exo, among which 37 miRNAs were much higher and 9 miRNAs were much lower in AB-Exo compared with YB-Exo. Among these up-regulated miRNAs, miR-483-5p and miR-2861 were respectively the most and second most abundant miRNAs in AB-Exo compared with YB-Exo. In our revised study, we

re-performed *in vitro* and *in vivo* experiments to determine the roles of these two miRNAs in the AB-Exo-induced promotion of BMSC adipogenesis, VSMC osteogenic transition, bone-fat imbalance, and vascular calcification. Functional assays *in vitro* showed that miR-483-5p and miR-2861 were required for the AB-Exo-induced positive effects on adipogenesis of BMSCs and osteogenic transdifferentiation of VSMCs, respectively (**new Figure 6h-m**). The results *in vivo* revealed that the inhibition of these two miRNAs markedly impaired the ability of AB-Exo to induce marrow adiposity and calcification paradox (**new Figure 7a-n**). These findings were consistent with the results showing in **Figure 5h-k** and **6a-j** in our original study. We really agreed with the reviewer that some other miRNAs may also partially contribute to the regulatory roles of AB-Exo in these processes, because either the antagomiR-483-5p- or antagomiR-2861-pre-treated AB-Exo could still induce trend of positive effects on BMSC adipogenic differentiation and VSMC mineralization *in vitro* as well as on adipogenesis and vascular calcification *in vivo*. The discussion on this issue has been added to **line 17-21, paragraph 6** in the discussion section of the revised manuscript. Thanks!

The authors used the VitD3 model of vascular calcification. This is an acute model in which vascular calcifications develop very abrupt. The question remains as to whether the observed effects would also be present in chronic models of vascular calcification such as e.g. the warfarin or adenine models, in which mechanisms underlying the calcifications may be different and perhaps more relevant from a clinical point of view.

Response: Thanks for the reviewer's valuable suggestion. In the revised study, using a method described previously [2, 3], we established chronic vascular calcification experiment models with chronic kidney disease (CKD) in 3-month-old female mice by freely feeding the mice with diet containing 0.25% adenine for 4 weeks. At the first day of week 1 and 3 during the feeding period, the mice were intravenously injected with YB-Exo, AB-Exo, or an equal volume of solvent. The schematic diagram of the experimental design has been added to new **Figure 4j** in our revised version. qRT-PCR analysis revealed that treatment with AB-Exo, but not YB-Exo, induced prominent reductions of mRNA levels of smooth muscle cell markers including *Sm22 α* and *α Sma* in abdominal aortas of these mice, indicating the loss of vascular smooth muscle phenotype after AB-Exo administration. Von Kossa and ARS

staining revealed that AB-Exo profoundly increased calcium deposition lesion areas in the mouse abdominal aortas, whereas the changes were not observed in the YB-Exo-treated groups. The marked increase of calcium deposition in abdominal aortas of the AB-Exo-treated mice was further confirmed by vascular calcium content analysis. Immunofluorescence staining for the osteogenic factor RUNX2 and qRT-PCR analysis for *Alpl*, respectively, revealed that AB-Exo, but not YB-Exo, resulted in significant upregulations of RUNX2 protein and *Alpl* mRNA levels in the mouse abdominal aortas, indicating that AB-Exo induce the transition of the cells within the vessels into an osteogenic phenotype. All these findings demonstrate that AB-Exo can exacerbate vascular calcification in mouse models of chronic vascular calcification, consistent with that observed in the mice with VD3-induced acute vascular calcification. These results have been added to new **Figure 4k-r** and the description of the results has been added to **line 1-11, paragraph 13** in the results section of the revised manuscript. Thanks!

new **Figure 4j-r**

j, Experimental design of adenine-induced chronic vascular calcification mouse models treated with solvent, YB-Exo, or AB-Exo by intravenous injection. **k**, qRT-PCR analysis of *Sm22α* and *αSma* expression in abdominal aortas of mice in (**j**). $n = 9$ per group. **l-n**, Von Kossa staining images (**l**) and quantification of the percentages of Von Kossa⁺ (**m**) and ARS⁺ (**n**) areas. Scale bar: 200 μm . $n = 10$ per group. **o**, Vascular calcium content measurement. $n = 10$ per group. **p-q**, RUNX2 immunofluorescence staining images (**p**) and quantification of the percentage of RUNX2⁺ areas (**q**). Scale bar: 200 μm . $n = 10$ per group. **r**, qRT-PCR for *Alpl* expression. $n = 9$ per group. ** $P < 0.01$, *** $P < 0.001$, **** $P < 0.0001$.

Page 19, para 1, lines 5-8: ‘...showing that other cell types such as hypertrophic chondrocytes are also able to produce SOST protein. Thus, more in-depth studies are required to decipher the cell origin of B-Exo.’ This might be relevant when comparing

young to aged people as one could expect differences in the features of the exosomes when originating from different cell types.

Response: Thanks for the reviewer's valuable suggestion. Exosomes are specifically enriched with many molecules from their parent cells and also express marker proteins specific for their parent cells [14, 15]. For instance, exosomes released by endothelial progenitor cells express the characteristic endothelial marker protein CD31 [16] and exosomes from human CD34⁺ stem cells are positive for CD34 [17]. In **Figure 1d** in original study, flow cytometric analysis showed that the vast majority of YB-Exo, AB-Exo, and osteocytes-derived exosomes (OCY-Exo) were positive for SOST protein, a glycoprotein protein that is mainly produced by osteocytes [18, 19]. In our revised study, we obtained hypertrophic chondrocytes by treating the chondrogenic cell line ATDC5 with chondrogenic differentiation medium for 21 days [20] and isolated the primary osteocytes and osteoblasts from the marrow-depleted mouse femurs and tibias using the protocol provided by Stern AR *et al.* [21]. Then, we conducted flow cytometric analysis to assess the protein expression of SOST, type I collagen (COL I, a osteoblast marker protein [22]), and type X collagen (COL X, a phenotypic marker of hypertrophic chondrocytes [23]) in YB-Exo, AB-Exo, OCY-Exo, osteoblasts-derived exosomes (OB-Exo), and hypertrophic chondrocytes-derived exosomes (HYPC-Exo). The results showed that SOST protein was expressed in YB-Exo, AB-Exo, OCY-Exo, and HYPC-Exo, but not in OB-Exo. COL I and COL X were expressed in the vast majority of OB-Exo and HYPC-Exo, respectively, whereas OCY-Exo were negative for this protein. A very small proportion (<10%) of YB-Exo and AB-Exo were positive for COL I and COL X. These results indicate that the majority of B-Exo are released from osteocytes, but not from osteoblasts or hypertrophic chondrocytes. These results have been added to new **Figure 1d** and the description of the results has been added to **line 8-20, paragraph 1** in the results section of the revised manuscript. Thanks!

new **Figure 1d**

d, Flow cytometric analysis of the expression of SOST, COL I, and COL X in the indicated exosomes.

Page 20, line 5: ‘... MiR-2861 is a miRNA that can promote osteoblast differentiation’. Is this the case in the present study. Did the authors check whether there was an effect on osteoblast differentiation at the level of the bone. This might be important as an improved bone formation may lead to a better incorporation of calcium (and phosphate) in bone leading to lower circulating levels and reduced risk for vascular calcification.

Response: Thanks for the reviewer’s valuable question and comment. In our study, we found that B-Exo from young bone (YB-Exo) could augment BMSC osteogenesis and enhance bone formation. The aged bone-derived B-Exo (AB-Exo), however, lost the ability to exert pro-osteogenic effect on bone, but could favor adipogenesis of BMSCs and mineralization of VSMCs *in vitro* and increase bone-fat imbalance as well as VD3-induced vascular calcification *in vivo*. We identified that miR-483-5p and miR-2861 were highly enriched in AB-Exo compared with YB-Exo by miRNA array. Since miR-483-5p has been reported to facilitate adipogenic differentiation [24, 25] and miR-2861 can promote osteoblast differentiation and osteogenic transdifferentiation of VSMCs [26, 27], we then determined the role of miR-483-5p in the AB-Exo-induced promotion of BMSC adipogenesis and bone-fat imbalance, and the role of miR-2861 in the AB-Exo-induced positive effects on osteogenic transdifferentiation VSMCs and vascular calcification. Since AB-Exo had no notable effects on osteogenic differentiation of BMSCs *in vitro* and osteogenic responses *in vivo*, we did not assess whether the inhibition of the pro-osteogenic miR-2861 could

affect the effects of AB-Exo on osteoblast differentiation at the level of the bone. The miR-2861-abundant AB-Exo was not sufficient to augment osteogenic differentiation of BMSCs and bone formation like YB-Exo, which may be associated with the accumulation of bone-detrimental molecules, such as the pro-adipogenic miR-483-5p, in AB-Exo. The discussion on this issue has been stressed in **line 21-24, paragraph 6** in the discussion section of the revised manuscript. Thanks!

Page 26, Section ‘Animals and treatment’: ‘3-month-old male mice were intraperitoneally injected with VD3 (500 U/g/d) for continuous 4 days, followed by treatment with 200 or 500 µg of AB-Exo, YB-Exo, antagomiR-2861- or antagomiR-NC-pre-treated AB-Exo, or an equal volume of PBS by intravenous injection (100 µL per mice) at days 1 and 3, or intramedullary injection (50 µL per mice) at day 1 only’. It is not clear from this text to the reader whether the compounds were administered during the time of VitD3 treatment or thereafter. This is important since in the first case the observed effect is preventive whilst in the second case it is curative.

Response: Thanks for the reviewer’s valuable comments. In our study, AB-Exo, YB-Exo, antagomiR-2861- or antagomiR-NC-pre-treated AB-Exo, or an equal volume of solvent (PBS) were administered in the mouse models of vascular calcification during the period of VD3 treatment, but not after the establishment of vascular calcification model by VD3 administration. For AB-Exo, YB-Exo, or solvent treatment in VD3-induced acute vascular calcification mouse models, we presented schematic diagrams of the experimental design in **Figure 3a** and **3g** in our original version and new **Figure 4a** and **S7a** in our revised version. These interventions were actually preventive, but not curative.

new Figure 4a

a, Experimental design of the VD3-induced acute vascular calcification mouse models treated with solvent, YB-Exo, or AB-Exo by intravenous injection.

new **Figure S7a**

a, Experimental design of the VD3-induced acute vascular calcification mouse models treated with solvent, YB-Exo, or AB-Exo by intramedullary injection.

Vascular calcification in the present study in general is assessed by Alazarin staining which provides info on the localization of the calcifications, however, is less quantitative. Only in a limited number of experiments bulk calcium is measured (which from a quantitative point of view is more relevant). Is there any reason why bulk calcium was not assessed in all experiments?

Response: Thanks for the reviewer's valuable comment and question. We really agreed with the reviewer that we had better to assess vascular calcium content in all vascular calcification-related animal experiments. In our revised study, besides ARS staining, we also utilized vascular calcium content analysis and Von Kossa staining to assess the extent of calcium deposition in vascular calcification mouse models treated with solvent, YB-Exo, or AB-Exo. The results showed that the intravenous injection of AB-Exo, but not YB-Exo, resulted in profound increases of vascular calcium content and Von Kossa-stained calcium deposition lesion areas in abdominal aortas of the VD3-induced acute vascular calcification and adenine-induced chronic vascular calcification mouse models. Treatment with AB-Exo by intramedullary injection could also increase the content of vascular calcium and the percentage of Von Kossa-stained calcium deposition lesion areas in abdominal aortas of the mice with VD3-induced acute vascular calcification. After pre-treatment with antagomiR-2861, but not antagomiR-NC, the abilities of AB-Exo to augment the content of vascular calcium and the percentage of Von Kossa-stained calcium deposition lesion areas were markedly decreased in the mice with VD3-induced acute vascular calcification. We also tested the content of vascular calcium in the VD3-treated aged Sham mice and OVX mice receiving solvent or ALE treatment. The results showed that OVX induced a significant increase of vascular calcium content in the VD3-treated aged mice. The resorption inhibitor ALE did not notably affect the content of vascular calcium in the VD3-treated aged Sham mice, but significantly reduced vascular

calcium content in the VD3-treated aged OVX mice. The above findings were consistent with results of ARS staining. We have added these new results to new **Figure 4c-d, 4f, 4l-m, 4o, 4t-v, 5o, 7h-i, and 7k**. The description of the results has been added to the results section (**line 7-13, paragraph 12; line 4-11, paragraph 13; line 9-14, paragraph 15; line 3-9, paragraph 18; line 13-16, paragraph 18; line 21-26, paragraph 24**) of the revised manuscript. Thanks!

new **Figure 4**

a, Experimental design of the VD3-induced acute vascular calcification mouse models treated with solvent, YB-Exo, or AB-Exo by intravenous injection. **b**, qRT-PCR analysis of *Sm22 α* and *aSma* expression in abdominal aortas of mice in (**a**). n = 9 per group. **c-e**, Von Kossa and ARS staining images (**c**) and quantification of the percentages of Von Kossa⁺ (**d**) and ARS⁺ (**e**) areas. Scale bar: 200 μ m. n = 10 per group. **f**, Vascular calcium content measurement. n = 10 per group. **g-h**, RUNX2 immunofluorescence staining images (**g**) and quantification of the percentage of RUNX2⁺ areas (**h**). Scale bar: 200 μ m. n = 10 per group. **i**, qRT-PCR analysis of *Alpl* expression. n = 9 per group. **j**, Experimental design of the adenine-induced chronic vascular calcification mouse models treated with solvent, YB-Exo, or AB-Exo by intravenous injection. **k**, qRT-PCR analysis of *Sm22 α* and *aSma* expression in abdominal aortas of mice in (**j**). n = 9 per group. **l-n**, Von Kossa staining images (**l**) and quantification of the percentages of Von Kossa⁺ (**m**) and ARS⁺ (**n**) areas. Scale bar: 200 μ m. n = 10 per group. **o**, Vascular calcium content measurement. n = 10 per group. **p-q**, RUNX2 immunofluorescence staining images (**p**) and quantification of the percentage of RUNX2⁺ areas (**q**). Scale bar: 200 μ m. n = 10 per group. **r**, qRT-PCR for *Alpl* expression. n = 9 per group. **s**, Fluorescence microscopy analysis of femur and abdominal aorta sections from mice treated with solvent or DiO-labeled AB-Exo by intramedullary injection. CB: cortical bone; BM: bone marrow. Scale bar: 200 μ m (for bone) or 50 μ m (for vessel). **t-v**, Von Kossa staining images (**t**), quantification of the percentage of Von Kossa⁺ areas (**u**), and vascular calcium content measurement (**v**) in abdominal aortas from VD3-induced acute vascular calcification mouse models receiving solvent or AB-Exo treatment by intramedullary injection. Scale bar: 200 μ m. n = 5 per group. * $P < 0.05$, ** $P < 0.01$, *** $P < 0.001$, **** $P < 0.0001$.

new **Figure 5**

a, Total protein contents of exosomes isolated from the conditioned media of osteoclasts treated with solvent (OC-CM), ALE (OC^{ALE}-CM), AB (OC^{AB}-CM), or AB + ALE (OC^{AB+ALE}-CM). $n = 5$ per group. **b-d**, Quantification of the percentages of ARS⁺ (**b**) and ORO⁺ (**c**) areas in BMSCs with different treatments under osteogenic or adipogenic induction, or ARS⁺ areas (**d**) in VSMCs with different treatments under osteogenic induction. $n = 5$ per group. **e-f**, μ CT-reconstructed images of femurs from 16-month-old Sham or OVX mice in different groups (**e**) and quantification of Tb. BV/TV, Tb. N, Tb. Th, and Tb. Sp (**f**). Scale bars: 1 mm. $n = 5$ per group. **g-h**, PLIN immunofluorescence staining images of femur sections (**g**) and quantification of adipocyte number in bone marrow (**h**). Scale bar: 100 μ m. $n = 5$ per group. **i**, qRT-PCR for *Ppar γ* expression in femurs. $n = 5$ per group. **j-l**, OCN immunostaining images (**j**), quantification of osteoblast number on BS (**k**), and ELISA for OCN (**l**). Scale bar: 50 μ m. $n = 5$ per group. **m-n**, ARS staining images (**m**) and quantification of the percentage of ARS⁺ areas (**n**). Scale bar: 200 μ m. $n = 5$ per group. **o**, Vascular calcium content measurement. $n = 5$ per group. **p-q**, RUNX2 immunostaining images (**p**) and quantification of the percentage of RUNX2⁺ areas (**q**). Scale bar: 200 μ m. $n = 5$ per group. **r**, qRT-PCR for *Alpl* expression. $n = 5$ per group. * $P < 0.05$, ** $P < 0.01$, *** $P < 0.001$, **** $P < 0.0001$.

new **Figure 7**

a, μ CT-reconstructed images of femurs from 3-month-old young mice treated with solvent or AB-Exo pre-treated with antagomiR-NC or antagomiR-483-5p. Scale bars: 1 mm. **b**, Quantification of Tb. BV/TV, Tb. N, Tb. Th, and Tb. Sp. n = 10 per group. **c-d**, PLIN immunofluorescence staining images (c) and quantification of adipocyte number in bone marrow (d). Scale bar: 100 μ m. n = 8 per group. **e**, qRT-PCR analysis of *Ppar γ* expression. n = 9 per group. **f-g**, OCN immunostaining images (f) and quantification of osteoblast number on BS (g). Scale bar: 50 μ m. n = 8 per group. **h-j**, Von Kossa staining images (h), quantification of the percentage of Von Kossa⁺ (i) and ARS⁺ (j) areas in abdominal aortas from VD3-induced acute vascular calcification mouse models receiving solvent or AB-Exo pre-treated with antagomiR-NC or antagomiR-2861. Scale bar: 200 μ m. n = 10 per group. **k**, Vascular calcium content analysis. n = 10 per group. **l-m**, RUNX2 immunostaining images (l) and quantification of the percentage of RUNX2⁺ areas (m). Scale bar: 200 μ m. n = 10 per group. **n**, qRT-PCR for *Alpl* expression. n = 9 per group. **P* < 0.05, ***P* < 0.01, ****P* < 0.001, *****P* < 0.0001.

Reviewer #2 (Remarks to the Author):

In this manuscript, the authors demonstrated that exosomes derived from the aged bone matrix (AB-Exo) during bone resorption induced adipogenesis rather than osteogenesis of BMSCs and calcification of VSMCs. They succeeded in solving some of the mystery of the “calcification paradox” from the viewpoint of exosomes. Their finding would contribute to senescence research and bone research, however, there are some comments to improve this study.

Major comments

1.

One of the main aims of this study is to show that AB-Exo positively modulates osteogenic transdifferentiation of VSMCs. The authors should more directly demonstrate that AB-Exo promotes the phenotypical change of VSMCs into osteoblast-like cells. Many papers describe that this phenotypical change is accompanied by a gain of osteogenic markers and loss of SMC markers, such as SM22 α and SM α -actin. In *in vitro* experiments, the authors only showed up-regulation in RUNX2 expression and ALP activity. The authors should more directly demonstrate osteogenic transdifferentiation of VSMCs by showing down-regulation in SM22 α and SM α -actin mRNA expression. (Cardiovascular Research, 2018 March 15; Volume 114, Issue 4, :590–600 / Aging, 2019 Aug 15; 11(15): 5445–5462.)

Response: Thanks for the reviewer’s valuable comment and suggestion. It is really true as the reviewer indicated that many papers have described that VSMC calcification is accompanied by a gain of osteogenic markers and loss of SMC markers such as smooth muscle 22 α (SM22 α) and smooth muscle α -actin (SM α A/ α SMA) [28, 29]. According to the reviewer’s valuable suggestion, we performed qRT-PCR analysis to assess the expression of SM22 α and α SMA in human VSMCs undergoing osteogenic induction. The results showed that AB-Exo significantly reduced the mRNA levels of SM22 α and α SMA in VSMCs subjected to osteogenic transdifferentiation, whereas these changes were not observed in the group treated with YB-Exo. These findings, along with the positive effects of AB-Exo on vascular calcium deposition, RUNX2 expression, and ALP activity, indicate that AB-Exo have the ability to promote the switching of VSMC phenotype to osteogenic phenotype. For *in vivo* experiments in the revised study, we also evaluated the gene expression of SM22 α and α SMA in vascular calcification mouse models treated with

solvent, YB-Exo, or AB-Exo. Consistent with the data *in vitro*, the results showed that the intravenous injection of AB-Exo, but not YB-Exo, resulted in significant decreases of mRNA levels of *Sm22 α* and *α Sma* in abdominal aortas of the VD3-induced acute vascular calcification and adenine-induced chronic vascular calcification mouse models. The above results have been added to new **Figure 2e**, **4b**, and **4k** in the revised version. The description of the results has been added to the results section (**line 14-24, paragraph 4; line 4-7, paragraph 12; line 4-11, paragraph 13**) of the revised manuscript. Thanks!

new **Figure 2e**

e, qRT-PCR analysis of *SM22 α* and *α SMA* expression in VSMCs treated with solvent, YB-Exo, or AB-Exo under osteogenic induction. Scale bar: 50 μ m. n = 6 per group. * P < 0.05.

new **Figure 4b**

b, qRT-PCR analysis of *Sm22 α* and *α Sma* expression in abdominal aortas of the VD3-induced acute vascular calcification mouse models treated with solvent, YB-Exo, or AB-Exo by intravenous injection.

*** P < 0.001, **** P < 0.0001.

new **Figure 4k**

k, qRT-PCR analysis of *Sm22 α* and *α Sma* expression in abdominal aortas of the adenine-induced chronic vascular calcification mouse models treated with solvent, YB-Exo, or AB-Exo by intravenous injection. **** P < 0.0001.

2.

Furthermore, the authors estimated Runx2 expression and ALP activity to show osteogenic transdifferentiation of VSMCs without enough explanation. Kurimoto et al. (Am J Physiol Heart Circ Physiol 2009 Nov;297(5): H1673-84.) identified that ALP and type I collagen, in addition to Runx2, were significantly upregulated in the senescent VSMCs, suggesting their osteoblastic transition during the senescence. In this study, knockdown of either ALP or type I collagen significantly reduced the calcification in the senescent VSMCs. The knockdown of Runx2 significantly reduced the ALP expression and calcification; however, it did not reduce the type I collagen expression. This result suggested that there might be a Runx2-independent pathway involved in the osteoblastic transition of VSMCs. The osteoblastic transition mechanism of VSMCs has not been fully elucidated, and it may be slightly different depending on the body condition; senescence, CKD, diabetes, hypertension, or osteoporosis.

Indeed, it is well known that RUNX2 and ALP are the critical factors of osteoblast-like change of VSMCs, but the authors should describe the reason for selecting RUNX2 and ALP as the most significant factors of osteogenic differentiation of VSMCs, citing some more references in the manuscript.

Response: Thanks for the reviewer's valuable comment and suggestion. As the reviewer indicated, the study by Nakano-Kurimoto *et al.* has shown that the gene expression of *ALP* and type I collagen (*COL1A1*), and calcification in senescent human VSMCs are markedly enhanced compared with the control young cells [30]. RUNX2 mediates, but only partially, the osteoblastic transition of senescent VSMCs, because *RUNX2* knockdown just suppresses *ALP* expression and calcification, but has no effect on *COL1A1* expression, suggesting the existence of a RUNX2-independent pathway in the osteoblastic transition of senescent VSMCs [30]. Since RUNX2 and ALP are the critical factors of osteoblast-like change of VSMCs [30, 31], in our original study, we just selected these two factors for assessing the effects of AB-Exo and YB-Exo on osteogenic transdifferentiation of VSMCs. In our revised study, we re-performed ARS staining, qRT-PCR analysis, and ALP activity assay to compare the effects of AB-Exo and YB-Exo on VSMC osteoblastic transition. For qRT-PCR analysis, we did not only test the mRNA level of *RUNX2*, a key transcription factor essential for osteogenesis and VSMC calcification [30, 31], but also examined the expression of *COL1A1* gene, which encodes the major component of extracellular

matrix [30]. ARS staining, qRT-PCR analysis, and ALP activity assay, respectively, revealed that AB-Exo, but not YB-Exo, significantly increased calcium deposition, the expression of *RUNX2* and *COL1A1*, and ALP activity in human VSMCs undergoing osteogenic induction, indicating that AB-Exo augment osteogenic transdifferentiation of VSMCs. Based on the findings by Nakano-Kurimoto *et al.* and our evidences showing that both *RUNX2* and *COL1A1* were markedly increased upon AB-Exo treatment, we supposed that AB-Exo can function through both *RUNX2*-dependent and -independent pathways to augment VSMC osteoblastic transition, which warrants future investigation. The above results are displayed in new **Figure 2c-d** and **2f-g** in the revised version. The reasons why we selected *RUNX2*, ALP, and *COL1A1* as the significant factors for assessing VSMC osteoblastic transition and the description of the results have been added to **line 14-24, paragraph 4** in the results section of the revised manuscript. Thanks!

new **Figure 2c-d** and **2f-g**

c-d and **f-g**, ARS staining images (**c**), quantification of the percentage of ARS⁺ areas (**d**), qRT-PCR analysis of *RUNX2* and *COL1A1* (**f**) expression, and ALP activity (**g**) in VSMCs treated with solvent, YB-Exo, or AB-Exo under osteogenic induction. Scale bar: 50 μ m. n = 6 per group. ** $P < 0.01$, *** $P < 0.001$, **** $P < 0.0001$.

3.

Many osteoclast-derived exosomes are contained in the culture medium in OC (YB and AB)-CM. The properties of exosomes derived from OC cultured with bone slices may differ from those derived from simple-cultured OC; it is possible that OC resorbing YB and OC resorbing AB release different exosomes. If the amounts of exosomes released by OC is small compared to those released by bone cells, that does not matter. However, in our experience, osteoclasts differentiated from 264.7 cells release considerable amounts of exosomes. The exosomes released by AB-resorbing

OC do not contain molecules promoting osteogenesis of VSMCs?

Response: Thanks for the reviewer's valuable comment and question. We really agreed with the reviewer that the components in exosomes from simple-cultured osteoclasts (OC-Exo) may be different with exosomes from osteoclasts cultured with YB (OC^{YB}-Exo) or AB (OC^{AB}-Exo). There may exist some factors in OC^{YB}-Exo that can promote BMSC osteogenesis, and some molecules in OC^{AB}-Exo that can promote adipogenic differentiation of BMSCs and calcification of VSMCs, which may contribute to the pro-osteogenic effect of Exo in OC^{YB}-CM and the positive effects of Exo in OC^{AB}-CM on BMSC adipogenesis and VSMC mineralization showing in **Figure 1n-q** in our original study and **Figure 2j-m** in our revised study. In other words, the bone-resorbing osteoclasts may also secrete functional exosomes to affect bone and vessel phenotypes, which still warrants future investigation. We apologize that we did not utilize a great method that could efficiently separate OC-Exo and the released B-Exo from bone matrix in the culture supernatant of osteoclasts with bone slices. The discussion on this issue and our limitation have been stressed in **line 4-13, paragraph 7** in the discussion section of the revised version. Thanks!

4.

According to the "Methods" section, isolated exosomes were used at a 50 µg/ml concentration. This is a very high concentration. Tremendous amounts of exosomes would be needed to achieve this concentration. The concentration may be too far off from that *in vivo*. It would be desirable to observe the same phenomena at the lower concentration (at most 20 µg/ml) if they could.

Response: Thanks for the reviewer's valuable comment and suggestion. In our study, for assays *in vitro*, exosomes in different groups were used at the concentration of 50 µg/mL. For experiments *in vivo*, exosomes were used at 200 µg (dissolved in 100 µL PBS for intravenous injection; dissolved in 10 µL PBS for intramedullary injection) or 500 µg (dissolved in 10 µL PBS for intramedullary injection) per time for each mouse. 3-month-old adult male mice (weighing 21-24 g) and 15-month-old aged male mice (weighing 32-35 g) were used for exosomes administration. The extracellular fluid volume in mice has been estimated to be 20% of body weight, which means that 4.2-4.8 mL extracellular fluid exists in a 22-25 g mouse and 6.4-7.0 mL extracellular fluid exists in a 32-35 g mouse. Thus, when exosomes were injected by intravenous route, 200 µg exosomes distributed in 4.2-7.0 mL extracellular fluid would yield

concentrations of 47.62-28.57 $\mu\text{g}/\text{mL}$, which were lower than but not too far below that of *in vitro* (50 $\mu\text{g}/\text{mL}$). The volume of bone marrow in a mouse is usually lower than 60 μL . Once exosomes were treated through intramedullary route, 200 or 500 μg exosomes distributed in the bone marrow fluid would yields concentrations larger than 50 $\mu\text{g}/\text{mL}$. Nevertheless, the concentrations of the injected exosomes would probably not achieve 50 $\mu\text{g}/\text{mL}$ due to rapid distribution and metabolization. According to the reviewer's suggestion, in the revised study, we re-performed the related functional assays *in vitro* and selected the dose of 20 $\mu\text{g}/\text{mL}$ for exosomes treatments. Since the reviewer suggested that we should use alternative methods like ultra-centrifugation and sucrose purification to harvest exosomes in the major comments No.5, we utilized Optiprep™ density gradient ultracentrifugation, but not the commercial kit, to isolate exosomes in our revised study. The *in vitro* results regarding the effects of exosomes in different groups have been added to new **Figure 2a-m, 6h-m, S1a-c, S2a-b, and S3c-f** in the revised version, which were consistent with that observed in our original study and did not affect the conclusions in our study.

new Figure 2a-m

a-b, ARS or ORO staining of BMSCs treated with solvent, YB-Exo, or AB-Exo under osteogenic or adipogenic induction (**a**) and quantification of the percentages of ARS⁺ and ORO⁺ areas (**b**). Scale bar: 50 μ m. n = 6 per group. **c-g**, ARS staining (**c**), quantification of the percentage of ARS⁺ areas (**d**), qRT-PCR analysis of *SM22 α* and *α SMA* (**e**), *RUNX2* and *COL1A1* (**f**) expression, and ALP activity (**g**) in VSMCs treated with solvent, YB-Exo, or AB-Exo under osteogenic induction. Scale bar: 50 μ m. n = 6 per group. **h**, Quantification of the percentages of ARS⁺ and ORO⁺ areas in BMSCs treated with solvent, YB-OCY-Exo, or AB-OCY-Exo under osteogenic or adipogenic induction. n = 3 per group. **i**, Quantification of the percentages of ARS⁺ areas in VSMCs treated with solvent, YB-OCY-Exo, or AB-OCY-Exo under osteogenic induction. n = 3 per group. **j-k**, ARS and ORO staining (**j**) and quantification of the percentages of ARS⁺ and ORO⁺ areas (**k**) in BMSCs with different treatments under osteogenic or adipogenic induction. OC: osteoclasts; CM: conditioned media. Scale bar: 50 μ m. n = 6 per group. **l-m**, ARS staining (**l**) and quantification of the percentage of ARS⁺ areas (**m**) in VSMCs with different treatments under osteogenic induction. Scale bar: 50 μ m. n = 6 per group. **P* < 0.05, ***P* < 0.01, ****P* < 0.001, *****P* < 0.0001.

new Figure 6h-m

h-j, ORO staining images (**h**), quantification of the percentage of ORO⁺ areas (**i**), and qRT-PCR for *Pparγ* expression (**j**) in BMSCs with different treatments under adipogenic induction. Scale bar: 50 μ m. n = 6 per group. **k-m**, ARS staining images (**k**), quantification of the percentage of ARS⁺ areas (**l**), and qRT-PCR for *RUNX2* expression (**m**) in VSMCs with different treatments under osteogenic induction. Scale bar: 50 μ m. n = 6 per group. * $P < 0.05$, ** $P < 0.01$, *** $P < 0.001$, **** $P < 0.0001$.

new Figure S1a-c

a, Internalization of the red fluorescent dye DiI-labeled YB-Exo and AB-Exo by BMSCs. Scale bar: 25 μ m. **b**, qRT-PCR analysis of the expression of osteogenesis-related genes (*Runx2*, *Bglap*, and *Alpl*) in BMSCs with different treatments under osteogenic induction. n = 6 per group. **c**, qRT-PCR analysis of the expression of adipogenesis-related genes (*Pparγ*, *Cebpa*, and *Fabp4*) in BMSCs with different treatments under adipogenic induction. n = 6 per group. * $P < 0.05$, ** $P < 0.01$, *** $P < 0.001$, **** $P < 0.0001$.

0.0001.

new **Figure S2a-b**

a, Uptake of the DiI-labeled YB-Exo and AB-Exo by VSMCs. Scale bar: 25 μ m. **b**, CCK-8 analysis of the survival/growth of VSMCs treated with solvent, YB-Exo, or AB-Exo. n = 4 per group.

new **Figure S3c-f**

c, ARS or ORO staining of BMSCs treated with solvent or rats-derived YB-Exo or AB-Exo under osteogenic or adipogenic induction. Scale bar: 50 μ m. **d**, Quantification of the percentages of ARS⁺ and ORO⁺ areas. n = 6 per group. **e-f**, ARS staining of VSMCs with different treatments under osteogenic induction (**e**) and quantification of the percentage of ARS⁺ areas (**f**). Scale bar: 50 μ m. n = 6 per group.

**** $P < 0.0001$.

Most serious problem of this article lies on the separation of exosomes. The authors used ExoQuick system. Unfortunately, this commercialized kit have a problematic for exosome or other membrane particles separation since it is well recognized that this kit may allow huge amount of contamination with proteins and other materials. The authors need to clarify by alternative methods like ultracentrifugation and sucrose purification.

Response: Optiprep™ density gradient ultracentrifugation can be utilized to isolate exosomes and reduce the contamination of protein aggregates and other materials [32-34]. According to the reviewer's valuable comment and suggestion, in our revised study, we utilized Optiprep™ density gradient ultracentrifugation, but not the commercial kit, to isolate exosomes. Then, we re-performed the exosomes-related *in vitro* and *in vivo* experiments except for miRNA array for AB-Exo and YB-Exo, and qRT-PCR analysis for miR-483-5p and miR-2861 in serum exosomes. The procedures for the isolation of exosomes using Optiprep™ density gradient ultracentrifugation have been detailed in methods section (**line 1-17, paragraph 4** and **line 1-16, paragraph 5**) in the revised manuscript. The results were consistent with that observed in our original study and did not affect the conclusions in our study. Thanks!

Minor comments

1.

Figure 1 is very small and difficult to see. In Figure 1 (A), I could not see the morphology of the exosomes. In Figure 1 (E), I could not see the dots of the fluorescent dye.

Response: Thanks for the reviewer's valuable comments. We apologize for our negligence of the detail. We have provided new images for **Figure 1a** with higher magnification and better resolution in the revised version. As the reviewer 3 suggested that we should provide definitive proof to show that osteocytes-derived exosomes do enter the circulation and reach vascular tissue in the major comments NO. 2, we did not only assess the fluorescent signals of mCherry and eGFP proteins in the bone and vessel sections of *Dmp1^{iCre}; Cd63^{em(loxp-mCherry-loxp-eGFP)}3* mice, but also performed immunostaining to determine whether the eGFP-labeled exosomes could be stained by SOST protein, a glycoprotein protein that is mainly produced by osteocytes [18, 19]. Thus, the images in **Figure 1e** in our original version have been replaced with new images showing in new **Figure 1g** in our revised study. We have tried our best to

provide the images with high quality, from which we can clearly see the dot-like fluorescent signals. Thanks!

new Figure 1a

a, Morphology of YB-Exo and AB-Exo. Scale bar: 50 nm.

new Figure 1g

g, Localization of eGFP (green) and mCherry (red), and immunofluorescence staining for SOST (purple) in bone and vessel from *Dmp1^{iCre}* mice, *Cd63^{em(loxp-mCherry-loxp-eGFP)3}* mice, and *Dmp1^{iCre}; Cd63^{em(loxp-mCherry-loxp-eGFP)3}* mice. Scale bar: 20 μ m (for bone) or 50 μ m (for vessel).

2.

In the "Preparation of bone-resection OC-CM" section (page 22, line 538 - page 23, line 549), the authors should describe more detail about the method. Did they seed the RAW264.7 cells on the bone slices? How did they know that osteoclasts began to resorb bone? What medium did they use? Did they take any pictures of osteoclasts as supplemental data? Didn't they perform medium change? In our experience, osteoclasts derived from RAW264.7 cells 8-10 days after induction usually undergo apoptosis, and it is quite challenging to maintain mature osteoclasts without changing to fresh medium even if RANKL concentration is 100 ng/ml. Please describe in detail as much as possible.

Response: Thanks for the reviewer's valuable questions and suggestions. Studies have indicated that multiple growth factors, cytokines, and minerals deposited in the bone matrix can be released into bone marrow during osteoclastic bone resorption and then participate in the regulation of bone homeostasis [35-39]. To mimic this process *in vitro*, the researchers (including the corresponding author Hui Xie in this study) plate the macrophages/monocytes onto bone slices and culture them in osteoclastic induction medium for 6-8 days to generate mature osteoclasts [37-39]. Using a scanning electronic microscope, Tang *et al.* have demonstrated that osteoclasts begin to form and resorb bone at days 6-7 [37] (**images shown as below**). Then, they culture the osteoclasts with bone slices for additional 1-3 days, in order to harvest the bone resorption-conditioned media (CM) for analyzing the components of interest in the CM and evaluating the regulatory function of the CM [37-39]. In the present study, for preparation of the CM from young or aged bone-resorbing osteoclasts (OC^{YB}-CM or OC^{AB}-CM), the periosteum- and bone marrow-depleted young or aged bones were cut into small slices and equally distributed into several culture plates. The osteoclast progenitor RAW264.7 cells (1×10^5 per well) were seeded onto the bone slices and cultured in osteoclastic induction medium (high glucose DMEM + 10% FBS + 100 ng/mL RANKL). The cells cultured in the RANKL-containing osteoclastic induction medium without bone slices served as the control group. The medium was changed every other day. After 5 to 6 days of induction, many osteoclasts were formed and

located on the surface of the bone slices. According to the past experience of Prof. Hui Xie as a postdoctoral researcher in the research lab led by Prof. Xu Cao in Johns Hopkins University School of Medicine, the newly formed osteoclasts began to resorb bone during this period. At days 6, the medium was replaced with fresh osteoclastic induction medium (high glucose DMEM + exosomes-depleted 10% FBS + 100 ng/mL RANKL) and the cells were incubated for another 2 days. Then, the OC^{YB}-CM, OC^{AB}-CM, or CM from the simple-cultured osteoclasts (OC-CM) were collected. If the newly formed osteoclasts were less after 5 to 6 days of induction, the cells were subjected to induction for a total of 8 days, followed by incubation in fresh osteoclastic induction medium (high glucose DMEM + exosomes-depleted 10% FBS + 100 ng/mL RANKL) for additional 2 days to collect the CM. As the medium was regularly updated and the induction time was not very long, few osteoclasts suffered apoptosis during the induction period. Since the study by Tang *et al.* have demonstrated that osteoclasts begin to form and resorb bone after induction for 6-7 days [37], we did not further collect the bone slices for scanning electronic microscopy in this study. The information on the preparation of bone resorption CM has been detailed in **line 1-15, paragraph 8** in the methods section of the revised version. Thanks!

Supplementary Figure 1. Preparation of osteoclastic bone resorption conditioned medium [37].

(a) Macrophages/monocytes isolated from bone marrow of four to eight week-old mice, which were cultured in the presence of M-CSF (22 ng ml^{-1}) as osteoclastic precursors and negative for TRAP staining, or with both M-CSF and RANKL (100 ng ml^{-1}), which induced osteoclast formation as indicated by TRAP staining (left two panels). Scanning electronic microscopy revealed that the precursors did not resorb bovine bone slices after being cultured for seven days whereas the osteoclasts exhibited active bone resorption (right two panels).

Reviewer #3 (Remarks to the Author):

In this manuscript, Wang et al report potential paracrine/endocrine effects of bone-derived exosomal vesicles on osteoblast differentiation and vascular calcification. They propose an interesting model in which bone-derived exosomal content varies with aging. Increased levels of exosomal miRNA-483-5p may promote increased marrow adipogenesis at the expense of reduced bone formation. In addition, increased levels of exosomal miRNA-2861 may promote vascular calcification. While this is an interesting model with implications for common aging-associated changes, enthusiasm is limited due to largely indirect experimental evidence to support this hypothesis as detailed below.

Major concerns:

1. It is very difficult to evaluate the physiologic significance of the mouse exosome transfer experiments performed in Figures 2, 3 and 6. In the absence of a 'dose response', it is completely unclear if the amounts of exosomes transferred by intramedullary or IV injection reflect anything close to normal biology. Along these lines, while the comparison of effects of exosomes from young versus aged rat bones is interesting, I am slightly concerned about cross species immunogenicity related to this experimental design. Moreover, the effects of exosomes isolated from other (non-bone) tissues (liver or blood, for example) should be assessed in this highly artificial model.

Response: Thanks for these valuable comments and suggestions. Our responses are displayed as follows:

1) In the revised study, since the reviewer 2 suggested that we should use an alternative method to harvest exosomes in the major comments No.5, we utilized Optiprep™ density gradient ultracentrifugation, but not the commercial kit, to isolate exosomes and re-performed all the exosomes-related functional experiments *in vivo*. The results (new **Figure 3, 4, and 7**) were consistent with that observed in our original study (**Figure 2, 3, and 6**), which confirmed the benefits of YB-Exo on bone, the positive effects of AB-Exo on bone-fat imbalance and vascular calcification, and the essential roles of miR-483-5p and miR-2861 in the AB-Exo-induced promotion of bone-fat imbalance and vascular calcification. We apologize for that we did not compare the effects of different concentrations of

AB-Exo and YB-Exo on bone and vessel phenotypes, and assessed whether there existed a dose-dependent responses in the treated mice. We also apologize for that we have not found out evidence showing the physiological concentrations of AB-Exo and YB-Exo in the bone and vessel tissues. In this study, we did not perform accurate assays to determine the physiological concentrations of AB-Exo and YB-Exo. These limitations have been stressed in the discussion section (**line 14-21, paragraph 7**) of our revised manuscript. Thanks!

- 2) In our revised study, according to the reviewer's valuable suggestion, we obtained and photographed the spleen samples from 3-month-old young mice intravenously injected with solvent, YB-Exo, or AB-Exo one time per two weeks for 4 weeks. The result showed that the YB-Exo- or AB-Exo-treated mice showed comparable spleen sizes and weights compared to the solvent-treated control mice. Hematoxylin and eosin (H&E) staining revealed that treatment with YB-Exo or AB-Exo did not induce obvious histopathological changes such as inflammatory cell infiltration and lymph node hyperplasia in the mouse spleen tissues. There were also no significant differences in the percentages of lymphocytes and neutrophils in white blood cells among the solvent-, YB-Exo-, or AB-Exo-treated mice. Together, these findings indicate that the rats-derived YB-Exo and AB-Exo do not induce notable immune and inflammatory responses in mice after intravenous injection. These results have been added to new **Figure S5** in the revised version and the description of the results has been added to **line 1-10, paragraph 11** in the results section of the revised manuscript. Thanks!
- 3) In our study, we found that the much higher levels of miR-483-5p and miR-2861 were not only detected in B-Exo, bone tissues, and abdominal aortas from aged mice, but also in liver tissues from aged mice and serum exosomes (Ser-Exo) from old people, as compared with the same type of samples from young mice or young people (**Figure 5a-e** in the original study and **Figure 6a-e** in the revised study). According to the reviewer's valuable suggestion, we obtained the liver tissues-derived exosomes (Liver-Exo) and Ser-Exo from young (2-month-old) and aged (18-month-old) mice for further analyses. qRT-PCR analysis revealed that Liver-Exo and Ser-Exo from aged mice (A-Liver-Exo and A-Ser-Exo) had higher levels of these two miRNAs than those from young mice (Y-Liver-Exo and Y-Ser-Exo), but the extents of up-regulation were higher in A-Ser-Exo than that in A-Liver-Exo. We then assessed the effects of these exosomes on BMSC

differentiation fate and VSMC mineralization at the same dose with YB-Exo and AB-Exo. ARS and ORO staining showed that Y-Ser-Exo, but not other exosomes, induced a statistically significant increase of calcium nodule formation of BMSCs, whereas only A-Ser-Exo markedly increased BMSC adipogenesis and VSMC mineralization. Although A-Liver-Exo had increased levels of miR-483-5p and miR-2861 compared with Y-Liver-Exo, treatment with A-Liver-Exo at the current dose could not induce marked effects on BMSC adipogenesis and VSMC mineralization, which may be associated with the insufficient levels of these miRNAs or/and the enrichment of other miRNAs that have different regulatory effects on these processes in A-Liver-Exo. These findings, along with the high extent of miR-483-5p and miR-2861 accumulation in AB-Exo and the remarkable positive effects of AB-Exo on BMSC adipogenesis and VSMC osteogenic transition, suggest that AB-Exo, but not A-Liver-Exo, are the primary source of these two miRNAs in A-Ser-Exo and contribute to the A-Ser-Exo-induced promotion of BMSC adipogenesis and VSMC mineralization. Nevertheless, we could not rule out the contribution of A-Liver-Exo to the increase of miR-483-5p and miR-2861 in A-Ser-Exo. We also could not rule out that exosomes from other non-bone tissues such as Liver-Exo may be involved in the development of aging-associated bone-fat imbalance and vascular calcification. We apologize for that we did not further perform animal experiments to test whether other non-bone tissues-derived exosomes play roles in bone-fat imbalance and vascular calcification during aging. The results on the levels of miR-483-5p and miR-2861 in Liver-Exo and Ser-Exo from young or aged mice and their effects on BMSC differentiation fate and VSMC mineralization have been added to new **Figure S14a-e**. The description of the results, the related discussion, and our limitation have been added to the revised manuscript (**line 12-16, paragraph 22** and **line 20-26, paragraph 22** in the results section; **line 24-33, paragraph 6** in the discussion section). Thanks!

new **Figure S5**

a-b, Gross view (**a**) and organ weight (**b**) of spleens from the mice treated with solvent or rats-derived YB-Exo or AB-Exo. Scale bar: 1 cm. $n = 10$ per group. **c**, H&E staining of spleen sections. Scale bar: 100 μm . **d**, The percentages of lymphocytes and neutrophils in white blood cells. $n = 10$ per group.

new **Figure S14a-e**

a, qRT-PCR for miR-483-5p and miR-2861 expression in Liver-Exo or Ser-Exo from 3-month-old or 18-month-old mice. $n = 6$ per group. **b-c**, ARS or ORO staining of BMSCs with different treatments

under osteogenic or adipogenic induction (**b**) and quantification of the percentages of ARS⁺ and ORO⁺ areas (**c**). Scale bar: 50 μ m. n = 3 per group. **d-e**, ARS staining of VSMCs with different treatments under osteogenic induction (**d**) and quantification of the percentage of ARS⁺ areas. Scale bar: 50 μ m. n = 3 per group. **P* < 0.05, ***P* < 0.01, ****P* < 0.001, *****P* < 0.0001.

2. Definitive proof that bone-derived exosomes really do enter the circulation and reach vascular tissue remains lacking. The study in Figure 3F is interesting, but it remains possible that some of the intramedullar-injected exosomes directly enter the circulation in a non-physiologic manner. The authors really should take advantage of their novel Cd63 knockin reporter model here to study trafficking of bone-derived exosomes in a physiologic manner. For example, since *Dmp1-Cre* is not expressed in blood vessels, it would be very important to demonstrate eGFP⁺ signal in vessels in this model. Without this kind of information, it's really hard to know whether or not bone-derived exosomes actually enter the circulation under normal circumstances.

Response: Thanks for the reviewer's valuable comment and suggestion. In our study, we generated *Cd63*^{em(loxp-mCherry-loxp-eGFP)3} mice harboring conditional *Cd63* alleles in which the stop codon in exon 8 at the 3' UTR of *Cd63* gene was replaced with a knock-in *mCherry* reporter gene flanked by two LoxP sites, and followed by an *eGFP* reporter gene, which would be activated if the *loxp-mCherry-loxp* sequence was excised (new **Figure 1e**). Then, we crossed the reporter mice with the transgenic mice expressing improved Cre recombinase under the control of the *Dmp1* promoter to generate *Dmp1*^{iCre}; *Cd63*^{em(loxp-mCherry-loxp-eGFP)3} mice, whose eGFP could be transcribed in osteocytes due to the deletion of the floxed *mCherry* by iCre recombinase (new **Figure 1e-f**). In our revised study, we assessed the fluorescent signals of mCherry and eGFP proteins in the bone and vessel sections of *Dmp1*^{iCre}; *Cd63*^{em(loxp-mCherry-loxp-eGFP)3} mice, *Cd63*^{em(loxp-mCherry-loxp-eGFP)3} mice, and *Dmp1*^{iCre} mice. Meanwhile, we performed immunostaining to determine whether the eGFP-labeled exosomes could be stained by SOST protein, a glycoprotein protein that is mainly produced by osteocytes [18, 19]. The results showed the presence of abundant mCherry red fluorescence in cells (osteocytes) within the bone matrix of cortical bone of *Cd63*^{em(loxp-mCherry-loxp-eGFP)3} mice, and in cells of vascular tissues from *Cd63*^{em(loxp-mCherry-loxp-eGFP)3} mice and *Dmp1*^{iCre}; *Cd63*^{em(loxp-mCherry-loxp-eGFP)3} mice. However, there were only a few signals of mCherry protein in osteocytes of *Dmp1*^{iCre}; *Cd63*^{em(loxp-mCherry-loxp-eGFP)3} mice. Abundant dot-like eGFP green signals were at the

perinuclear region of osteocytes, within the bone matrix, or detected in the vascular tissues of these mice, and most of these green dots were positive for SOST protein, suggesting that most of them are the released exosomes by osteocytes (OCY-Exo) and can enter into the bone matrix and vascular tissues under physiologic condition. Neither mCherry or eGFP signals were observed in the bone and vascular tissues of *Dmp1*^{iCre} mice, indicating that these signals in *Cd63*^{em(loxp-mCherry-loxp-eGFP)3} mice and *Dmp1*^{iCre}; *Cd63*^{em(loxp-mCherry-loxp-eGFP)3} mice are not non-specific-fluorescence. There were dot-like SOST-positive signals in the vascular tissues of *Dmp1*^{iCre} mice and *Cd63*^{em(loxp-mCherry-loxp-eGFP)3} mice, which also suggest that OCY-Exo can be transported from bone to blood vessels under normal circumstance. Since we have demonstrated that osteocytes are the primary cells that release B-Exo in **Figure 1d**, these results suggest that B-Exo can enter from bone matrix to the circulation under physiologic condition. The above new results have been added to new **Figure 1g** in the revised version and the description of the results has been added to **line 11-29, paragraph 2** in the results section of the revised version. Thanks!

new **Figure 1**

a-d, Morphology (**a**), diameter distribution (**b**), exosomal marker analysis (**c**), and expression of SOST,

COL I, and COL X (**d**) in the indicated exosomes. Scale bar: 50 nm. **e**, Schematic diagram of the gene targeting strategy for the generation of $Cd63^{em(loxP-mCherry-loxP-eGFP)3}$ mice by inserting a *mCherry* reporter gene flanked by two *loxP* sites and an *eGFP* reporter gene in the stop codon in exon 8 at the 3' UTR of *Cd63* gene. **f**, PCR genotyping of wild-type mice, $Dmp1^{iCre}$ mice, $Cd63^{em(loxP-mCherry-loxP-eGFP)3}$ mice, and $Dmp1^{iCre}; Cd63^{em(loxP-mCherry-loxP-eGFP)3}$ mice using primers for determining the insertion of *iCre* (up) and *eGFP* (down). **g**, Localization of eGFP (green) and mCherry (red), and immunofluorescence staining for SOST (purple) in bone and vessel from $Dmp1^{iCre}$ mice, $Cd63^{em(loxP-mCherry-loxP-eGFP)3}$ mice, and $Dmp1^{iCre}; Cd63^{em(loxP-mCherry-loxP-eGFP)3}$ mice. Scale bar: 20 μ m (for bone) or 50 μ m (for vessel).

3. While it is interesting that distinct exosome-derived miRNAs might regulate BMSC differentiation and vascular calcification, information about the target genes for these distinct miRNAs is lacking.

Response: Thanks for the reviewer's valuable comments. Zhang *et al.* has reported that miR-483-5p facilitates adipogenesis of mouse pre-adipocyte 3T3-L1 cells by positively regulating the expression of *Ppar γ* [24]. Chen *et al.* has demonstrated that miR-483-5p promotes adipogenic differentiation of human adipose-derived MSCs by directly inhibiting *ERK1* gene and subsequently increasing *PPAR γ* expression [25]. MiR-2861 is a miRNA that can target histone deacetylase 5 (HDAC5) to promote the expression of *Runx2*, thus stimulating osteoblast differentiation and osteogenic transdifferentiation of VSMCs [26, 27]. The above information has been stressed in **line 2-7, paragraph 6** in the discussion section of our revised manuscript. In our revised study, we conducted qRT-PCR analysis to assess whether miR-483-5p and miR-2861 were respectively responsible for the regulatory effects of AB-Exo on *Ppar γ* expression in mouse BMSCs and *RUNX2* expression in human VSMCs. The results showed that the overexpression of miR-483-5p by agomiR-483-5p further augmented the ability of AB-Exo to stimulate *Ppar γ* expression in mouse BMSCs under adipogenesis, but the down-regulation of this miRNA by antagomiR-483-5p significantly suppressed the positive effect of AB-Exo on *Ppar γ* expression in these differentiated BMSCs. As compared with the agomiR-NC-pre-treated AB-Exo, the agomiR-2861-pre-treated AB-Exo induced a much higher level of *RUNX2* expression in human VSMCs under osteogenic induction, but the antagomiR-2861-pre-treated AB-Exo failed to increase *RUNX2* expression in the differentiated VSMCs. We also tested whether the suppression of miR-483-5p and miR-2861 in AB-Exo by the respective antagomiRs could affect their effects of AB-Exo on the gene or protein expression of *Ppar γ* and *Runx2* in the mouse bone and vascular tissues, respectively.

qRT-PCR analysis showed a marked increase of *Pparγ* expression in the bone tissues of 3-month-old young mice treated with the antagomiR-NC-pre-treated AB-Exo, while the antagomiR-483-5p-pre-treated AB-Exo failed to stimulate the expression of *Pparγ* in the bone tissues of mice. In VD3-treated 3-month-old young mice, immunostaining confirmed a markedly increased expression of RUNX2 protein in the vascular tissues of the mice treated with the antagomiR-NC-pre-treated AB-Exo, but the antagomiR-2861-pre-treated AB-Exo did not notably influence vascular RUNX2 expression. These findings demonstrate that PPAR γ and RUNX2 are the downstream factors positively modulated by miR-483-5p and miR-2861, respectively. These results have been added to new **Figure 6j, 6m, 7e, and 7l-m** in the revised version and the description of the results has been added to **line 5-14, paragraph 23 and line 26-29, paragraph 24** in the results section of the revised version. Thanks!

new **Figure 6h-m**

h-j, ORO staining images (**h**), quantification of the percentage of ORO⁺ areas (**i**), and qRT-PCR for *Pparγ* expression (**j**) in BMSCs with different treatments under adipogenic induction. Scale bar: 50 μ m. n = 6 per group. **k-m**, ARS staining images (**k**), quantification of the percentage of ARS⁺ areas (**l**), and qRT-PCR for *RUNX2* expression (**m**) in VSMCs with different treatments under osteogenic induction. Scale bar: 50 μ m. n = 6 per group. * $P < 0.05$, ** $P < 0.01$, *** $P < 0.001$, **** $P < 0.0001$.

new **Figure 7e**

e, qRT-PCR analysis of *Pparγ* expression in femurs from 3-month-old young mice treated with solvent or AB-Exo pre-treated with antagomiR-NC or antagomiR-483-5p. n = 9 per group. **** $P < 0.0001$.

new **Figure 7l-m**

l-m, RUNX2 immunostaining images (**l**) and quantification of the percentage of RUNX2⁺ areas (**m**) in abdominal aortas from VD3-induced acute vascular calcification mouse models receiving solvent or AB-Exo pre-treated with antagomiR-NC or antagomiR-2861. Scale bar: 200 μ m. n = 10 per group.

**** $P < 0.0001$.

4. The OVX experiment in figure 4 where mice are treated plus/minus alendronate confirms the expected effects of this bisphosphonate on bone mass. The relationship of this experiment to exosomes is completely speculative.

Again, it would be powerful to use the Cd63 knockin model to show that alendronate blocks trafficking of exosomes from bone to blood vessels. Obviously there are multiple ways that alendronate can effect bone biology, and a relationship to exosome trafficking *in vivo* remains purely speculative at this point.

Response: Thanks for the reviewer's valuable comment and suggestion. In **Figure 4a-d** in our original study and **Figure 5a-d** in our revised study, we determined that the bone resorption inhibitor alendronate (ALE) could inhibit AB-Exo release from aged bone, and suppress the ability of conditioned media from osteoclasts cultured with the aged bone slices (OC^{AB}-CM) to augment BMSC adipogenesis and VSMC calcification. In the following *in vivo* experiments, we further determined that OVX could exacerbate VD3-induced vascular calcification in aged mice, and ALE could attenuate bone-fat imbalance and VD3-induced vascular calcification in the aged OVX mice (**Figure 4e-o** in our original study and **Figure 5e-r** in our revised study). These findings, together with the stimulatory effects of AB-Exo on BMSC adipogenesis and VSMC calcification *in vitro* as well as on bone-fat imbalance and vascular calcification *in vivo*, suggest that the blockade of AB-Exo release because of the inhibition of bone resorption is another important mechanism by which ALE protect against bone-fat imbalance and vascular calcification. There are multiple ways that ALE affects bone phenotypes. We really agreed with the reviewer that the association between the inhibition of AB-Exo release by ALE and the protective effects of ALE against bone-fat imbalance and vascular calcification remains

speculative based on current evidences. We apologize for that we did not utilize a method that could selectively block the release of AB-Exo without affecting other processes in this study. According to the reviewer's valuable suggestion, we tried to obtain $Dmp1^{iCre}; Cd63^{em(loxp-mCherry-loxp-eGFP)3}$ mice for testing whether ALE could block or reduce the trafficking of AB-Exo from bone to blood vessels, but we have not yet obtained enough mice for the following experiments due to various reasons. From the first half of this year, the reproductive capacities of some transgenic mice including $Dmp1^{iCre}$ mice in our group seemed to be decreased. In May, five sex- and age-matched $Dmp1^{iCre}; Cd63^{em(loxp-mCherry-loxp-eGFP)3}$ mice were obtained, but these mice died due to unknown reasons. In July, four offspring of $Dmp1^{iCre}$ mice and $Cd63^{em(loxp-mCherry-loxp-eGFP)3}$ mice were born, but unfortunately none of them were identified to be $Dmp1^{iCre}; Cd63^{em(loxp-mCherry-loxp-eGFP)3}$ mice. Recently, six $Dmp1^{iCre}; Cd63^{em(loxp-mCherry-loxp-eGFP)3}$ mice were successfully obtained, but very unfortunately and sadly, they failed to grow up due to infanticide by their mothers or some unknown reasons. In our opinion, to interpret the results in **Figure 4e-o** in our original study and **Figure 5e-r** in our revised study by evidence *in vivo*, we should use the aged (16-month-old) Sham or OVX $Dmp1^{iCre}; Cd63^{em(loxp-mCherry-loxp-eGFP)3}$ mice for ALE treatment. If all goes well, this will take more than one year and a half. For this revision, we have requested the editor to extend the revision deadline for two times, but now we cannot apply for another extension according to the editor's claim. We apologize for that we did not use an animal model to trace AB-Exo *in vivo* after ALE administration in this study. In our revised manuscript, our limitations on this issue have been stressed in **line 28-34, paragraph 5** in the discussion section. Thanks!

Minor concerns:

1. Additional clinical data is needed about the patients (young and old) from whom exosomal vesicles were isolated.

Response: Thanks for the reviewer's valuable suggestion. Since the reviewer 2 suggested us to use a better method to harvest exosomes in the major comments No.5, in our revised study, we utilized Optiprep™ density gradient ultracentrifugation, but not the commercial kit, to isolate exosomes. Thus, the human bone and serum samples used in the revised study were collected from new patients. The human bone tissues were collected during surgical resection in patients (three 27–31 years old young women and three 67–73 years old women) who underwent open reduction and

internal fixation of tibial plateau fracture, or joint replacement due to osteoarthritis, in order to obtain YB-Exo and AB-Exo. Serum samples from these donors were collected to isolate exosomes. The young donors had no other diseases. Besides osteoarthritis, the aged donors suffered from other diseases such as osteoporosis, thyroid nodule, pulmonary nodule, pneumonia, hyperlipidemia, hyperuricemia, or/and thyroid polypectomy. The clinical information on these patients has been detailed in new **Table S2** in our revised version. Thanks!

Table S2 Clinical information for the human donors

Patient no.	Group	Gender	Age	Pathogeny	Surgery	Other diseases
01	Young	Female	27	Tibial plateau fracture	Fracture open reduction and internal fixation	None
02	Young	Female	29	Tibial plateau fracture	Fracture open reduction and internal fixation	None
03	Young	Female	31	Tibial plateau fracture	Fracture open reduction and internal fixation	None
04	Aged	Female	67	Osteoarthritis	Joint replacement	Osteoporosis Thyroid nodule Pulmonary nodule Pneumonia
05	Aged	Female	68	Osteoarthritis	Joint replacement	Osteoporosis Hyperlipidemia Hyperuricemia Pulmonary nodule
06	Aged	Female	73	Osteoarthritis	Joint replacement	Osteoporosis Right thyroid polypectomy

2. Regarding the CD63 GFP knockin model, the authors should acknowledge that Dmp1-Cre is active in osteoblasts and other places (see PMID 28163952). For this reason, perhaps Dmp1-Cre isn't the most ideal Cre driver to use to try to label osteocyte-derived vesicles. The authors might consider Sost-Cre as an alternative. In addition, description of this model (Fig 1E) should include all appropriate negative controls due to problems with auto-fluorescence in bone sections.

Response: According to the reviewer's valuable suggestions, in our revised study, we did not only assess the fluorescent signals of mCherry and eGFP proteins in the bone and vessel sections of *Dmp1^{iCre}; Cd63^{em(loxp-mCherry-loxp-eGFP)3}* mice, but also tested these

signals in $Cd63^{em(loxp-mCherry-loxp-eGFP)3}$ mice and $Dmp1^{iCre}$ mice. Meanwhile, we performed immunostaining to determine whether the eGFP-labeled exosomes could be stained by SOST protein, a glycoprotein protein that is mainly produced by osteocytes [18, 19]. The results showed the presence of abundant mCherry red fluorescence in cells (osteocytes) within the bone matrix of cortical bone of $Cd63^{em(loxp-mCherry-loxp-eGFP)3}$ mice, and in cells of vascular tissues from $Cd63^{em(loxp-mCherry-loxp-eGFP)3}$ mice and $Dmp1^{iCre}; Cd63^{em(loxp-mCherry-loxp-eGFP)3}$ mice. However, there were only a few signals of mCherry protein in osteocytes of $Dmp1^{iCre}; Cd63^{em(loxp-mCherry-loxp-eGFP)3}$ mice. Abundant dot-like eGFP green signals were at the perinuclear region of osteocytes, within the bone matrix, or detected in the vascular tissues of these mice, and most of these green dots were positive for SOST protein, suggesting that most of them are the released exosomes by osteocytes (OCY-Exo) and can enter into the bone matrix and vascular tissues under physiologic condition. Neither mCherry or eGFP signals were observed in the bone and vascular tissues of $Dmp1^{iCre}$ mice, indicating that these signals in $Cd63^{em(loxp-mCherry-loxp-eGFP)3}$ mice and $Dmp1^{iCre}; Cd63^{em(loxp-mCherry-loxp-eGFP)3}$ mice are not non-specific-fluorescence. There were dot-like SOST-positive signals in the vascular tissues of $Dmp1^{iCre}$ mice and $Cd63^{em(loxp-mCherry-loxp-eGFP)3}$ mice, which also suggest that OCY-Exo can be transported from bone to blood vessels under normal circumstance. The above results have been added to new **Figure 1g** in the revised version and the description of the results has been added to **line 11-29, paragraph 2** in the results section of the revised version.

Dmp1-Cre mice are widely used for gene deletion in osteocytes [40-43], but *Dmp1-Cre* has been reported to be able to inevitably target osteoblasts and some cells in other places [44]. DMP1 expression is also found in hypertrophic chondrocytes [45]. Thus, *Dmp1-Cre* mice is not an ideal model to label and trace OCY-Exo. SOST protein is mainly produced by osteocytes [18, 19]. However, there are evidences that other cell types such as hypertrophic chondrocytes are also able to produce SOST protein [46, 47], which suggests that the *Sost-Cre* mouse is also not a perfect model to label and trace OCY-Exo. We apologize for that we did not find out and utilize an ideal Cre mouse model that targets only osteocytes. Since exosomes usually express marker proteins specific for their parent cells [14-17], in our revised study, we conducted flow cytometric analysis to assess the protein expression of SOST, type I collagen (COL I, a osteoblast marker protein [22]), and type X collagen (COL X, a phenotypic marker of hypertrophic chondrocytes [23]) in YB-Exo, AB-Exo, OCY-Exo,

osteoblasts-derived exosomes (OB-Exo), and hypertrophic chondrocytes-derived exosomes (HYPC-Exo). The results (new **Figure 1d**) showed that SOST protein was expressed in YB-Exo, AB-Exo, OCY-Exo, and HYPC-Exo, but not in OB-Exo. COL I and COL X were expressed in the vast majority of OB-Exo and HYPC-Exo, respectively, whereas OCY-Exo were negative for this protein. A very small proportion (<10%) of YB-Exo and AB-Exo were positive for COL I and COL X. These findings, along with the evidence that OCY-Exo could trigger a B-Exo-like age-dependent regulation of BMSC differentiation and VSMC calcification (new **Figure 2h-i**), indicate that osteocytes, the most abundant bone cells (>90%) [48], are the major parent cells that release B-Exo. Nevertheless, we could not rule out that a minority of osteoblasts and hypertrophic chondrocytes may also contribute to the generation of B-Exo and play roles in the calcification paradox. The discussion on this issue and our limitation have been stressed in **line 1-23, paragraph 3** in the discussion section of the revised version. Thanks!

new **Figure 1**

a-d, Morphology (a), diameter distribution (b), exosomal marker analysis (c), and expression of SOST,

COL I, and COL X **(d)** in the indicated exosomes. Scale bar: 50 nm. **e**, Schematic diagram of the gene targeting strategy for the generation of $Cd63^{em(loxp-mCherry-loxp-eGFP)3}$ mice by inserting a *mCherry* reporter gene flanked by two *loxP* sites and an *eGFP* reporter gene in the stop codon in exon 8 at the 3' UTR of *Cd63* gene. **f**, PCR genotyping of wild-type mice, $Dmp1^{iCre}$ mice, $Cd63^{em(loxp-mCherry-loxp-eGFP)3}$ mice, and $Dmp1^{iCre}; Cd63^{em(loxp-mCherry-loxp-eGFP)3}$ mice using primers for determining the insertion of *iCre* (up) and *eGFP* (down). **g**, Localization of eGFP (green) and mCherry (red), and immunofluorescence staining for SOST (purple) in bone and vessel from $Dmp1^{iCre}$ mice, $Cd63^{em(loxp-mCherry-loxp-eGFP)3}$ mice, and $Dmp1^{iCre}; Cd63^{em(loxp-mCherry-loxp-eGFP)3}$ mice. Scale bar: 20 μ m (for bone) or 50 μ m (for vessel).

new **Figure 2h-i**

h, Quantification of the percentages of ARS⁺ and ORO⁺ areas in BMSCs treated with solvent, YB-OCY-Exo, or AB-OCY-Exo under osteogenic or adipogenic induction. n = 3 per group. **i**, Quantification of the percentages of ARS⁺ areas in VSMCs treated with solvent, YB-OCY-Exo, or AB-OCY-Exo under osteogenic induction. n = 3 per group. **** $P < 0.0001$.

3. For all the *in vivo* animal studies, the authors should present data with each mouse as an individual data point rather than using "box and plunger" graphs as is currently done. In addition, the sample size used for most of the *in vivo* studies (n=5) seems somewhat small.

Response: Thanks for the reviewer's valuable suggestions. In the revised version, all bar graphs have been revised to bar graphs overlaid with dot plots, from which we can see the individual data points in each group. Moreover, since the reviewer 2 suggested that we should use an alternative method to harvest exosomes in the major comments No.5, we utilized Optiprep™ density gradient ultracentrifugation to isolate exosomes and re-performed the exosomes-related functional experiments *in vivo*. The sample sizes were increased (>5) for most of these experiments. The information on the sample size for each assay has been detailed in the figure legends of the revised manuscript. Thanks!

4. Along these lines, the bone changes assessed throughout focus exclusively on trabecular bone. Whether these mild changes are physiologically-important remains unclear. The authors should also report whether exosomes have effects on cortical

bone and/or bone strength.

Response: According to the reviewer's valuable suggestion, in the revised study, we analyzed the cortical bone parameters including cortical bone area fraction (Ct. Ar/Tt. Ar) and cortical thickness (Ct. Th). The results showed that both YB-Exo and AB-Exo did not cause statistically significant differences of these cortical bone parameters in both young (3-month-old) and aged (15-month-old) mice after intramedullary injection one time per two weeks for one month. After pre-treatment with antagomiR-NC or antagomiR-483-5p, AB-Exo also did not notably affect the levels of these cortical bone parameters in young mice after intramedullary injection. These results indicate that treatment with YB-Exo, AB-Exo, antagomiR-NC-pretreated AB-Exo, or antagomiR-483-5p-pretreated AB-Exo at the current administration regime is not sufficient to induce obvious effect on cortical bone. Treatment with the bone resorption inhibitor ALE also had no significant effects on Ct. Ar/Tt. Ar and Ct. Th in both aged Sham and OVX mice. Following the reviewer's valuable suggestion, we also performed three-point bending test to assess the femur strength in young and aged mice receiving solvent, YB-Exo, or AB-Exo treatment. The data showed that YB-Exo induced a trend of increase of femur ultimate load value in both young and aged mice, whereas AB-Exo caused a significant and tend of decrease of this parameter in young and aged mice, respectively, indicating the potential positive effect of YB-Exo and definite negative effect of AB-Exo on bone strength. These results have been added to new **Figure S4a-b, 3d, S8c, and S15a-b** in the revised version. The description of the results has been added to the results section (**line 21-27, paragraph 7; line 11-13, paragraph 17; line 9-11, paragraph 24**) of the revised version. Thanks!

new **Figure S4a-b**

a-b, Quantification of Ct. Ar/Tt. Ar (**a**) and Ct. Th. (**b**) of femurs from young or aged mice treated with solvent, YB-Exo, or AB-Exo. n = 10 per group. **P* < 0.05.

new **Figure 3d**

d, Ultimate load values of femurs from young or aged mice treated with solvent, YB-Exo, or AB-Exo. n = 10 for young mice. n = 6 for aged mice. **P* < 0.05, ****P* < 0.001.

new **Figure S8c**

c, Quantification of Ct. Ar/Tt. Ar and Ct. Th of femurs from 16-month-old aged Sham and OVX mice treated with solvent or ALE. n = 5 per group.

new **Figure S15a-b**

a-b, Quantification of Ct. Ar/Tt. Ar (**a**) and Ct. Th. (**b**) of femurs from 3-month-old young mice treated with solvent or AB-Exo pre-treated with antagomiR-NC or antagomiR-483-5p. n = 10 per group.

5. For assessment of marrow adipocytes, the authors would ideally confirm effects using perilipin staining with osmium-based microCT.

Response: Thanks for the reviewer's valuable suggestion. For technical reasons, we

apologize for that we just performed immunofluorescence staining for perilipin (PLIN), but did not use osmium tetroxide staining combined with μ CT, to assess the changes of marrow adipocytes. The limitation has been stressed in **line 1-3, paragraph 7** in the discussion section of the revised manuscript. Thanks!

6. For assessment of vascular calcification, using a complementary method to ARS staining would be ideal given challenges associated with this method.

Response: Thanks for the reviewer's valuable suggestion. In our revised study, besides ARS staining and immunofluorescence staining for the osteogenic factor RUNX2, we also utilized qRT-PCR analysis of smooth muscle cell markers (*Sm22 α* and *α Sma*) and osteogenic marker *Alpl*, Von Kossa staining, and vascular calcium content analysis, to assess the extent of vascular calcification in vascular calcification mouse models treated with solvent, YB-Exo, or AB-Exo. The results showed that the intravenous injection of AB-Exo, but not YB-Exo, resulted in significant decreases of mRNA levels of *Sm22 α* and *α Sma*, and profound increases of Von Kossa-stained calcium deposition lesion areas, vascular calcium content, and mRNA level of *Alpl* in abdominal aortas of the VD3-induced acute vascular calcification and adenine-induced chronic vascular calcification mouse models. Treatment with AB-Exo by intramedullary injection could also increase the Von Kossa-stained calcium deposition lesion areas, vascular calcium content, and *Alpl* expression in abdominal aortas of the mice with VD3-induced acute vascular calcification. After pre-treatment with antagomiR-2861, but not antagomiR-NC, the abilities of AB-Exo to augment the Von Kossa-stained calcium deposition lesion areas, vascular calcium content, and *Alpl* expression were all markedly decreased in the mice with VD3-induced acute vascular calcification. We also tested the content of vascular calcium in the VD3-treated aged Sham mice and OVX mice receiving solvent or ALE treatment. The results showed that OVX induced a significant increase of vascular calcium content in the VD3-treated aged mice. The resorption inhibitor ALE did not notably affect the content of vascular calcium in the VD3-treated aged Sham mice, but significantly reduced vascular calcium content in the VD3-treated aged OVX mice. The above findings were consistent with results of ARS staining and immunofluorescence staining for RUNX2. We have added these new results to new **Figure 4b-d, 4f, 4i, 4k-m, 4o, 4r, 5o, 5r, 7i-j, and 7l** in the revised version. The description of the results has been added to the results section (**line 4-17, paragraph**

12; line 4-11, paragraph 13; line 3-9, paragraph 18; line 13-16, paragraph 18; line 21-29, paragraph 24) of the revised manuscript. Thanks!

new **Figure 4**

a, Experimental design of the VD3-induced acute vascular calcification mouse models treated with solvent, YB-Exo, or AB-Exo by intravenous injection. **b**, qRT-PCR analysis of *Sm22α* and *αSma* expression in abdominal aortas of mice in (**a**). n = 9 per group. **c-e**, Von Kossa and ARS staining

images (c) and quantification of the percentages of Von Kossa⁺ (d) and ARS⁺ (e) areas. Scale bar: 200 μm. n = 10 per group. f, Vascular calcium content measurement. n = 10 per group. g-h, RUNX2 immunofluorescence staining images (g) and quantification of the percentage of RUNX2⁺ areas (h). Scale bar: 200 μm. n = 10 per group. i, qRT-PCR analysis of *Alpl* expression. n = 9 per group. j, Experimental design of the adenine-induced chronic vascular calcification mouse models treated with solvent, YB-Exo, or AB-Exo by intravenous injection. k, qRT-PCR analysis of *Sm22α* and *αSma* expression in abdominal aortas of mice in (j). n = 9 per group. l-n, Von Kossa staining images (l) and quantification of the percentages of Von Kossa⁺ (m) and ARS⁺ (n) areas. Scale bar: 200 μm. n = 10 per group. o, Vascular calcium content measurement. n = 10 per group. p-q, RUNX2 immunofluorescence staining images (p) and quantification of the percentage of RUNX2⁺ areas (q). Scale bar: 200 μm. n = 10 per group. r, qRT-PCR for *Alpl* expression. n = 9 per group. s, Fluorescence microscopy analysis of femur and abdominal aorta sections from mice treated with solvent or DiO-labeled AB-Exo by intramedullary injection. CB: cortical bone; BM: bone marrow. Scale bar: 200 μm (for bone) or 50 μm (for vessel). t-v, Von Kossa staining images (t), quantification of the percentage of Von Kossa⁺ areas (u), and vascular calcium content measurement (v) in abdominal aortas from VD3-induced acute vascular calcification mouse models receiving solvent or AB-Exo treatment by intramedullary injection. Scale bar: 200 μm. n = 5 per group. **P* < 0.05, ***P* < 0.01, ****P* < 0.001, *****P* < 0.0001.

new **Figure 5**

a, Total protein contents of exosomes isolated from the conditioned media of osteoclasts treated with solvent (OC-CM), ALE (OC^{ALE}-CM), AB (OC^{AB}-CM), or AB + ALE (OC^{AB+ALE}-CM). $n = 5$ per group. **b-d**, Quantification of the percentages of ARS⁺ (**b**) and ORO⁺ (**c**) areas in BMSCs with different treatments under osteogenic or adipogenic induction, or ARS⁺ areas (**d**) in VSMCs with different treatments under osteogenic induction. $n = 5$ per group. **e-f**, μ CT-reconstructed images of femurs from 16-month-old Sham or OVX mice in different groups (**e**) and quantification of Tb. BV/TV, Tb. N, Tb. Th, and Tb. Sp (**f**). Scale bars: 1 mm. $n = 5$ per group. **g-h**, PLIN immunofluorescence staining images of femur sections (**g**) and quantification of adipocyte number in bone marrow (**h**). Scale bar: 100 μ m. $n = 5$ per group. **i**, qRT-PCR for *Ppar γ* expression in femurs. $n = 5$ per group. **j-l**, OCN immunostaining images (**j**), quantification of osteoblast number on BS (**k**), and ELISA for OCN (**l**). Scale bar: 50 μ m. $n = 5$ per group. **m-n**, ARS staining images (**m**) and quantification of the percentage of ARS⁺ areas (**n**). Scale bar: 200 μ m. $n = 5$ per group. **o**, Vascular calcium content measurement. $n = 5$ per group. **p-q**, RUNX2 immunostaining images (**p**) and quantification of the percentage of RUNX2⁺ areas (**q**). Scale bar: 200 μ m. $n = 5$ per group. **r**, qRT-PCR for *Alpl* expression. $n = 5$ per group. * $P < 0.05$, ** $P < 0.01$, *** $P < 0.001$, **** $P < 0.0001$.

new **Figure 7**

a, μ CT-reconstructed images of femurs from 3-month-old young mice treated with solvent or AB-Exo pre-treated with antagomiR-NC or antagomiR-483-5p. Scale bars: 1 mm. **b**, Quantification of Tb. BV/TV, Tb. N, Tb. Th, and Tb. Sp. $n = 10$ per group. **c-d**, PLIN immunofluorescence staining images (**c**) and quantification of adipocyte number in bone marrow (**d**). Scale bar: 100 μ m. $n = 8$ per group. **e**, qRT-PCR analysis of *Ppar γ* expression. $n = 9$ per group. **f-g**, OCN immunostaining images (**f**) and quantification of osteoblast number on BS (**g**). Scale bar: 50 μ m. $n = 8$ per group. **h-j**, Von Kossa staining images (**h**), quantification of the percentage of Von Kossa⁺ (**i**) and ARS⁺ (**j**) areas in abdominal aortas from VD3-induced acute vascular calcification mouse models receiving solvent or AB-Exo pre-treated with antagomiR-NC or antagomiR-2861. Scale bar: 200 μ m. $n = 10$ per group. **k**, Vascular calcium content analysis. $n = 10$ per group. **l-m**, RUNX2 immunostaining images (**l**) and quantification of the percentage of RUNX2⁺ areas (**m**). Scale bar: 200 μ m. $n = 10$ per group. **n**, qRT-PCR for *Alpl* expression. $n = 9$ per group. * $P < 0.05$, ** $P < 0.01$, *** $P < 0.001$, **** $P < 0.0001$.

7. The authors should comment on the increase in miR-2861 and 483 in liver with aging. Even modest liver-derived increases clearly could impact circulating exosome

levels.

Response: In our study, we found that the much higher levels of miR-483-5p and miR-2861 were not only detected in B-Exo, bone tissues, and abdominal aortas from aged mice, but also in liver tissues from aged mice and serum exosomes (Ser-Exo) from old people, as compared with the same type of samples from young mice or young people (**Figure 5a-e** in the original study and **Figure 6a-e** in the revised study). In our revised study, we obtained the liver tissues-derived exosomes (Liver-Exo) and Ser-Exo from young (2-month-old) and aged (18-month-old) mice for further analyses. qRT-PCR analysis revealed that Liver-Exo and Ser-Exo from aged mice (A-Liver-Exo and A-Ser-Exo) had higher levels of these two miRNAs than those from young mice (Y-Liver-Exo and Y-Ser-Exo), but the extents of up-regulation were higher in A-Ser-Exo than that in A-Liver-Exo. We then assessed the effects of these exosomes on BMSC differentiation fate and VSMC mineralization at the same dose with YB-Exo and AB-Exo. ARS and ORO staining showed that Y-Ser-Exo, but not other exosomes, induced a statistically significant increase of calcium nodule formation of BMSCs, whereas only A-Ser-Exo markedly increased BMSC adipogenesis and VSMC mineralization. Although A-Liver-Exo had increased levels of miR-483-5p and miR-2861 compared with Y-Liver-Exo, treatment with A-Liver-Exo at the current dose could not induce marked effects on BMSC adipogenesis and VSMC mineralization, which may be associated with the insufficient levels of these miRNAs or/and the enrichment of other miRNAs that have different regulatory effects on these processes in A-Liver-Exo. These findings, along with the high extent of miR-483-5p and miR-2861 accumulation in AB-Exo and the remarkable positive effects of AB-Exo on BMSC adipogenesis and VSMC osteogenic transition, suggest that AB-Exo, but not A-Liver-Exo, are the primary source of these two miRNAs in A-Ser-Exo and contribute to the A-Ser-Exo-induced promotion of BMSC adipogenesis and VSMC mineralization. Nevertheless, we could not rule out the contribution of A-Liver-Exo to the increase of miR-483-5p and miR-2861 in A-Ser-Exo. We also could not rule out that exosomes from other non-bone tissues such as Liver-Exo may be involved in the development of aging-associated bone-fat imbalance and vascular calcification. In our study, we did not further perform animal experiments to test whether other non-bone tissues-derived exosomes play roles in bone-fat imbalance and vascular calcification during aging. The results on the levels of miR-483-5p and miR-2861 in Liver-Exo and Ser-Exo from young or aged mice

and their effects on BMSC differentiation fate and VSMC mineralization have been added to new **Figure S14a-e**. The description of the results, the related discussion, and our limitation have been added to the revised manuscript (**line 12-16, paragraph 22** and **line 20-26, paragraph 22** in the results section; **line 24-33, paragraph 6** in the discussion section). Thanks!

new **Figure S14a-e**

a, qRT-PCR for miR-483-5p and miR-2861 expression in Liver-Exo or Ser-Exo from 3-month-old or 18-month-old mice. $n = 6$ per group. **b-c**, ARS or ORO staining of BMSCs with different treatments under osteogenic or adipogenic induction (**b**) and quantification of the percentages of ARS⁺ and ORO⁺ areas (**c**). Scale bar: 50 μm . $n = 3$ per group. **d-e**, ARS staining of VSMCs with different treatments under osteogenic induction (**d**) and quantification of the percentage of ARS⁺ areas. Scale bar: 50 μm . $n = 3$ per group. * $P < 0.05$, ** $P < 0.01$, *** $P < 0.001$, **** $P < 0.0001$.

References

- Persy V, D'Haese P. Vascular calcification and bone disease: the calcification paradox. *Trends Mol Med*. 2009; 15: 405-16.
- Yoshida T, Yamashita M, Horimai C, Hayashi M. Smooth Muscle-Selective Nuclear Factor- κ B Inhibition Reduces Phosphate-Induced Arterial Medial Calcification in Mice With Chronic Kidney Disease. *J Am Heart Assoc*. 2017; 6.
- Clinkenbeard EL, Noonan ML, Thomas JC, Ni P, Hum JM, Aref M, et al. Increased FGF23 protects against detrimental cardio-renal consequences during elevated blood phosphate in CKD. *JCI Insight*. 2019; 4.
- Kukida M, Mogi M, Kan-No H, Tsukuda K, Bai HY, Shan BS, et al. AT2 receptor stimulation inhibits phosphate-induced vascular calcification. *Kidney Int*. 2019; 95: 138-48.
- Schantl AE, Verhulst A, Neven E, Behets GJ, D'Haese PC, Maillard M, et al. Inhibition of vascular calcification by inositol phosphates derivatized with ethylene glycol oligomers. *Nat Commun*. 2020; 11: 721.

-
6. He YH, Pu SY, Xiao FH, Chen XQ, Yan DJ, Liu YW, et al. Improved lipids, diastolic pressure and kidney function are potential contributors to familial longevity: a study on 60 Chinese centenarian families. *Sci Rep.* 2016; 6: 21962.
 7. Evenepoel P, Opdebeeck B, David K, D'Haese PC. Bone-Vascular Axis in Chronic Kidney Disease. *Adv Chronic Kidney Dis.* 2019; 26: 472-83.
 8. Jing L, Li L, Sun Z, Bao Z, Shao C, Yan J, et al. Role of Matrix Vesicles in Bone-Vascular Cross-Talk. *J Cardiovasc Pharmacol.* 2019; 74: 372-8.
 9. Fisher JE, Rogers MJ, Halasy JM, Luckman SP, Hughes DE, Masarachia PJ, et al. Alendronate mechanism of action: geranylgeraniol, an intermediate in the mevalonate pathway, prevents inhibition of osteoclast formation, bone resorption, and kinase activation in vitro. *Proc Natl Acad Sci U S A.* 1999; 96: 133-8.
 10. Reszka AA, Halasy-Nagy JM, Masarachia PJ, Rodan GA. Bisphosphonates act directly on the osteoclast to induce caspase cleavage of mst1 kinase during apoptosis. A link between inhibition of the mevalonate pathway and regulation of an apoptosis-promoting kinase. *J Biol Chem.* 1999; 274: 34967-73.
 11. Rennert G, Pinchev M, Gronich N, Saliba W, Flugelman A, Lavi I, et al. Oral Bisphosphonates and Improved Survival of Breast Cancer. *Clin Cancer Res.* 2017; 23: 1684-9.
 12. Saag KG, Petersen J, Brandi ML, Karaplis AC, Lorentzon M, Thomas T, et al. Romosozumab or Alendronate for Fracture Prevention in Women with Osteoporosis. *N Engl J Med.* 2017; 377: 1417-27.
 13. Lambrinoudaki I, Christodoulakos G, Botsis D. Bisphosphonates. *Ann N Y Acad Sci.* 2006; 1092: 397-402.
 14. Coakley G, Maizels RM, Buck AH. Exosomes and Other Extracellular Vesicles: The New Communicators in Parasite Infections. *Trends Parasitol.* 2015; 31: 477-89.
 15. Guo SC, Tao SC, Dawn H. Microfluidics-based on-a-chip systems for isolating and analysing extracellular vesicles. *J Extracell Vesicles.* 2018; 7: 1508271.
 16. Zhang J, Chen C, Hu B, Niu X, Liu X, Zhang G, et al. Exosomes Derived from Human Endothelial Progenitor Cells Accelerate Cutaneous Wound Healing by Promoting Angiogenesis Through Erk1/2 Signaling. *Int J Biol Sci.* 2016; 12: 1472-87.
 17. Sahoo S, Klychko E, Thorne T, Misener S, Schultz KM, Millay M, et al. Exosomes from human CD34(+) stem cells mediate their proangiogenic paracrine activity. *Circ Res.* 2011; 109: 724-8.
 18. Appelman-Dijkstra NM, Papapoulos SE. Clinical advantages and disadvantages of anabolic bone therapies targeting the WNT pathway. *Nat Rev Endocrinol.* 2018; 14: 605-23.
 19. van Bezooijen RL, ten Dijke P, Papapoulos SE, Löwik CW. SOST/sclerostin, an osteocyte-derived negative regulator of bone formation. *Cytokine Growth Factor Rev.* 2005; 16: 319-27.
 20. Yamaura K, Akiyama S, Ueno K. Increased expression of the histamine H4 receptor subtype in hypertrophic differentiation of chondrogenic ATDC5 cells. *J Cell Biochem.* 2012; 113: 1054-60.
 21. Stern AR, Bonewald LF. Isolation of osteocytes from mature and aged murine bone. *Methods Mol Biol.* 2015; 1226: 3-10.
 22. Bolarin DM, Swerdlow P, Wallace AM, Littsey L. Type I collagen as a marker of bone metabolism in sickle cell hemoglobinopathies. *J Natl Med Assoc.* 1998; 90: 41-5.
 23. Wilson R, Freddi S, Chan D, Cheah KS, Bateman JF. Misfolding of collagen X chains harboring Schmid metaphyseal chondrodysplasia mutations results in aberrant disulfide bond formation, intracellular retention, and activation of the unfolded protein response. *J Biol Chem.* 2005; 280: 15544-52.

-
24. Zhang J, Huang Y, Shao H, Bi Q, Chen J, Ye Z. Grape seed procyanidin B2 inhibits adipogenesis of 3T3-L1 cells by targeting peroxisome proliferator-activated receptor γ with miR-483-5p involved mechanism. *Biomed Pharmacother.* 2017; 86: 292-6.
 25. Chen K, He H, Xie Y, Zhao L, Zhao S, Wan X, et al. miR-125a-3p and miR-483-5p promote adipogenesis via suppressing the RhoA/ROCK1/ERK1/2 pathway in multiple symmetric lipomatosis. *Sci Rep.* 2015; 5: 11909.
 26. Hu R, Liu W, Li H, Yang L, Chen C, Xia ZY, et al. A Runx2/miR-3960/miR-2861 regulatory feedback loop during mouse osteoblast differentiation. *J Biol Chem.* 2011; 286: 12328-39.
 27. Xia ZY, Hu Y, Xie PL, Tang SY, Luo XH, Liao EY, et al. Runx2/miR-3960/miR-2861 Positive Feedback Loop Is Responsible for Osteogenic Transdifferentiation of Vascular Smooth Muscle Cells. *Biomed Res Int.* 2015; 2015: 624037.
 28. Durham AL, Speer MY, Scatena M, Giachelli CM, Shanahan CM. Role of smooth muscle cells in vascular calcification: implications in atherosclerosis and arterial stiffness. *Cardiovasc Res.* 2018; 114: 590-600.
 29. Henze LA, Luong TTD, Boehme B, Masyout J, Schneider MP, Brachs S, et al. Impact of C-reactive protein on osteo-/chondrogenic transdifferentiation and calcification of vascular smooth muscle cells. *Aging.* 2019; 11: 5445-62.
 30. Nakano-Kurimoto R, Ikeda K, Uraoka M, Nakagawa Y, Yutaka K, Koide M, et al. Replicative senescence of vascular smooth muscle cells enhances the calcification through initiating the osteoblastic transition. *Am J Physiol Heart Circ Physiol.* 2009; 297: H1673-84.
 31. Sun Y, Byon CH, Yuan K, Chen J, Mao X, Heath JM, et al. Smooth muscle cell-specific runx2 deficiency inhibits vascular calcification. *Circ Res.* 2012; 111: 543-52.
 32. Ji H, Greening DW, Barnes TW, Lim JW, Tauro BJ, Rai A, et al. Proteome profiling of exosomes derived from human primary and metastatic colorectal cancer cells reveal differential expression of key metastatic factors and signal transduction components. *Proteomics.* 2013; 13: 1672-86.
 33. Arab T, Raffo-Romero A, Van Camp C, Lemaire Q, Le Marrec-Croq F, Drago F, et al. Proteomic characterisation of leech microglia extracellular vesicles (EVs): comparison between differential ultracentrifugation and Optiprep density gradient isolation. *J Extracell Vesicles.* 2019; 8: 1603048.
 34. Xu R, Greening DW, Rai A, Ji H, Simpson RJ. Highly-purified exosomes and shed microvesicles isolated from the human colon cancer cell line LIM1863 by sequential centrifugal ultrafiltration are biochemically and functionally distinct. *Methods.* 2015; 87: 11-25.
 35. Crane JL, Cao X. Function of matrix IGF-1 in coupling bone resorption and formation. *J Mol Med.* 2014; 92: 107-15.
 36. Crane JL, Cao X. Bone marrow mesenchymal stem cells and TGF-beta signaling in bone remodeling. *J Clin Invest.* 2014; 124: 466-72.
 37. Tang Y, Wu X, Lei W, Pang L, Wan C, Shi Z, et al. TGF-beta1-induced migration of bone mesenchymal stem cells couples bone resorption with formation. *Nat Med.* 2009; 15: 757-65.
 38. Xian L, Wu X, Pang L, Lou M, Rosen CJ, Qiu T, et al. Matrix IGF-1 maintains bone mass by activation of mTOR in mesenchymal stem cells. *Nat Med.* 2012; 18: 1095-101.
 39. Xie H, Cui Z, Wang L, Xia Z, Hu Y, Xian L, et al. PDGF-BB secreted by preosteoclasts induces angiogenesis during coupling with osteogenesis. *Nat Med.* 2014; 20: 1270-8.
 40. Lu Y, Xie Y, Zhang S, Dusevich V, Bonewald LF, Feng JQ. DMP1-targeted Cre expression in odontoblasts and osteocytes. *J Dent Res.* 2007; 86: 320-5.
 41. Xiong J, Onal M, Jilka RL, Weinstein RS, Manolagas SC, O'Brien CA. Matrix-embedded cells

control osteoclast formation. *Nat Med.* 2011; 17: 1235-41.

42. Joeng KS, Lee YC, Lim J, Chen Y, Jiang MM, Munivez E, et al. Osteocyte-specific WNT1 regulates osteoblast function during bone homeostasis. *The Journal of clinical investigation.* 2017; 127: 2678-88.

43. Asada N, Katayama Y, Sato M, Minagawa K, Wakahashi K, Kawano H, et al. Matrix-embedded osteocytes regulate mobilization of hematopoietic stem/progenitor cells. *Cell Stem Cell.* 2013; 12: 737-47.

44. Lim J, Burclaff J, He G, Mills JC, Long F. Unintended targeting of Dmp1-Cre reveals a critical role for Bmpr1a signaling in the gastrointestinal mesenchyme of adult mice. *Bone Res.* 2017; 5: 16049.

45. Zhang Q, Lin S, Liu Y, Yuan B, Harris SE, Feng JQ. Dmp1 Null Mice Develop a Unique Osteoarthritis-like Phenotype. *Int J Biol Sci.* 2016; 12: 1203-12.

46. Weivoda MM, Youssef SJ, Oursler MJ. Sclerostin expression and functions beyond the osteocyte. *Bone.* 2017; 96: 45-50.

47. van Bezooijen RL, Bronckers AL, Gortzak RA, Hogendoorn PC, van der Wee-Pals L, Balemans W, et al. Sclerostin in mineralized matrices and van Buchem disease. *J Dent Res.* 2009; 88: 569-74.

48. Tatsumi S, Ishii K, Amizuka N, Li M, Kobayashi T, Kohno K, et al. Targeted ablation of osteocytes induces osteoporosis with defective mechanotransduction. *Cell Metab.* 2007; 5: 464-75.

REVIEWERS' COMMENTS

Reviewer #1 (Remarks to the Author):

The revised version of the manuscript has markedly improved as the authors performed various additional analyses and experiments by which their conclusions now are very well supported by the additional results. The methodology is innovative, sound and described in sufficient detail. Overall this is an original study which is of particular interest shedding new light on the mechanisms underlying the so-called 'calcification paradox' which in the long run might be highly relevant in the search for new therapeutics to prevent/treat vascular calcification without affecting bone mineralization.

Reviewer #2 (Remarks to the Author):

The authors revised their manuscript as according to the comments. The usage of more accurate EV extraction method rather than commercialized kit reevaluate the accuracy of the results.

Reviewer #3 (Remarks to the Author):

In this revised manuscript, the authors addressed the majority of the points that were previously raised. That being said, it is unfortunate that they did not more carefully evaluate the relationship of their exosome transfer studies to normal physiology by performing 'dose response' experiments. Despite this limitation, this represents a large and interesting body of work that will open new and exciting directions in the study of bone-derived exosomes in skeletal and vascular aging.

Dear Editor:

Thank you very much for your letter and for the editor's and reviewers' comments concerning our manuscript currently entitled "**Aged bone matrix-derived extracellular vesicles as a messenger for calcification paradox**" (NCOMMS-20-48849A). These comments are valuable and helpful for revising and improving our manuscript. We have resubmitted new version of our manuscript and the amendments are marked with tracked changes in the revised manuscript. The manuscript with tracked changes accepted has been also submitted to the system. We have also finished the Author Checklist according to the editorial requests in the attached documents. The updated documents have been submitted to the manuscript system. Here we make the following revise in response to the editor's and reviewers' critiques.

We hope that the revised manuscript will be accepted for publication in *Nature Communications*. Thank you very much for your time and your consideration.

Yours truly,

Hui Xie and Chun-Yuan Chen

REVIEWERS' COMMENTS

Reviewer #1 (Remarks to the Author):

The revised version of the manuscript has markedly improved as the authors performed various additional analyses and experiments by which their conclusions now are very well supported by the additional results. The methodology is innovative, sound and described in sufficient detail.

Overall this an original study which is of particular interest shedding new light on the mechanisms underlying the so-called 'calcification paradox' which on the long run might be highly relevant in the search for new therapeutics to prevent/treat vascular calcification without affecting bone mineralization.

Response: Thank you very much for your time and positive comments on our

manuscript.

Reviewer #2 (Remarks to the Author):

The authors revised their manuscript as according to the comments.

The usage of more accurate EV extraction method rather than commercialized kit reevaluate the accuracy of the results.

Response: Thank you very much for your time and positive comments on our manuscript.

Reviewer #3 (Remarks to the Author):

In this revised manuscript, the authors addressed the majority of the points that were previously raised. That being said, it is unfortunate that they did not more carefully evaluate the relationship of their exosome transfer studies to normal physiology by performing 'dose response' experiments. Despite this limitation, this represents a large and interesting body of work that will open new and exciting directions in the study of bone-derived exosomes in skeletal and vascular aging.

Response: Thanks for the reviewer's valuable comments. We apologize for that we did not perform "dose response" experiments to more carefully evaluate the effects of AB-Exo and YB-Exo on bone and vessel phenotypes in the normal physiology and the pathology of osteoporosis and vascular calcification. Currently, there is no evidence showing the physiological concentrations of AB-Exo and YB-Exo in the bone and vessel tissues. Future studies are required to determine the physiological concentrations of AB-Exo and YB-Exo using accurate assays and investigate whether there exist dose-dependent responses in the AB-Exo- or YB-Exo-treated mice, which will be beneficial for more deeply deciphering the functional roles of bone-derived exosomes in skeletal and vascular aging and for developing strategies to inhibit AB-Exo or utilize YB-Exo for therapeutic uses. The above limitations and the related discussion have been stressed in the discussion section (**line 13-22, paragraph 7**) of our revised manuscript with tracked changes accepted. However, it should be noted that "AB-Exo", "YB-Exo", and "bone-derived exosomes" have been revised to "AB-EVs", "YB-EVs", and "bone-derived EVs" in the revised manuscript according

to the editor's comment in Author Checklist. Thanks!